# Sustainable conversion of alkaline nitrate to ammonia at activities greater than 2 A cm$^{-2}$

Wanru Liao [1,9], Jun Wang [1,9], Ganghai Ni[1], Kang Liu[1], Changxu Liu [2], Shanyong Chen[3], Qiyou Wang[1], Yingkang Chen[1], Tao Luo[1], Xiqing Wang[1], Yanqiu Wang[3], Wenzhang Li[3], Ting-Shan Chan [4], Chao Ma [5], Hongmei Li[1], Ying Liang [6], Weizhen Liu[7], Junwei Fu[1] ✉, Beidou Xi [8] ✉ & Min Liu [1] ✉

Nitrate (NO$_3^-$) pollution poses significant threats to water quality and global nitrogen cycles. Alkaline electrocatalytic NO$_3^-$ reduction reaction (NO$_3$RR) emerges as an attractive route for enabling NO$_3^-$ removal and sustainable ammonia (NH$_3$) synthesis. However, it suffers from insufficient proton (H$^+$) supply in high pH conditions, restricting NO$_3^-$-to-NH$_3$ activity. Herein, we propose a halogen-mediated H$^+$ feeding strategy to enhance the alkaline NO$_3$RR performance. Our platform achieves near-100% NH$_3$ Faradaic efficiency (pH = 14) with a current density of 2 A cm$^{-2}$ and enables an over 99% NO$_3^-$-to-NH$_3$ conversion efficiency. We also convert NO$_3^-$ to high-purity NH$_4$Cl with near-unity efficiency, suggesting a practical approach to valorizing pollutants into valuable ammonia products. Theoretical simulations and in situ experiments reveal that Cl-coordination endows a shifted $d$-band center of Pd atoms to construct local H$^+$-abundant environments, through arousing dangling O-H water dissociation and fast *H desorption, for *NO intermediate hydrogenation and finally effective NO$_3^-$-to-NH$_3$ conversion.

Nitrate (NO$_3^-$) is one of the most widespread water contaminants, sourced from agricultural runoff and industrial sewage discharges, that disharmonizes the global nitrogen cycle[1-4]. The presence of NO$_3^-$ pollution in water bodies is a matter of significant environmental concern due to its detrimental impacts on both aquatic ecosystems and human health. Excessive NO$_3^-$ concentrations can lead to eutrophication[5], the depletion of dissolved oxygen, and the production of harmful algal blooms. Ingesting water with high NO$_3^-$ levels can also have adverse health effects, inducing problems such as cancer[6] and the commonly known blue-baby disease[7].

To address the pressing challenges posed by NO$_3^-$ pollution, tremendous efforts have been made to develop cost-effective methods for NO$_3^-$ removal, ranging from reverse osmosis, ion exchange, electrocatalysis to electrodialysis and biological denitrification[8,9]. Among them, electrocatalysis is the most promising solution[10-12], performed under mild conditions with high selectivity, versatility and environmental sustainability[1,13-16]. Within this scope, electrocatalytic NO$_3^-$ reduction reaction (NO$_3$RR), which converts NO$_3^-$ to ammonia (NH$_3$) with renewable electricity inputs, offers an operational avenue for restoring the disturbed nitrogen cycle and facilitating the

[1]Hunan Joint International Research Center for Carbon Dioxide Resource Utilization, State Key Laboratory of Powder Metallurgy, School of Physics, Central South University, Changsha 410083, PR China. [2]Centre for Metamaterial Research & Innovation, Department of Engineering, University of Exeter, Exeter EX4 4QF, UK. [3]School of Chemistry and Chemical Engineering, Central South University, Changsha 410083, PR China. [4]National Synchrotron Radiation Research Center, Hsinchu 300092, Taiwan. [5]College of Materials Science and Engineering, Hunan University, Changsha 410082, PR China. [6]College of Food Science and Engineering, Central South University of Forestry and Technology, Changsha 410004, PR China. [7]School of Environment and Energy, Guangdong Provincial Key Laboratory of Solid Wastes Pollution Control and Recycling, South China University of Technology, Guangzhou 510006, PR China. [8]State Key Laboratory of Environmental Criteria and Risk Assessment, Chinese Research Academy of Environmental Sciences, 100012 Beijing, PR China. [9]These authors contributed equally: Wanru Liao, Jun Wang. ✉e-mail: fujunwei@csu.edu.cn; xibd@craes.org.cn; minliu@csu.edu.cn

denitrification of wastewater, as well as a sustainable alternative to the energy-intensive Haber-Bosch techniques that consume 2% of the world's energy and release 1.4% of global carbon dioxide emissions[17-21]. Tackling $NO_3^-$ in high pH systems is of particular interest owing to its practical applications toward industrial and agricultural wastewater[22] and less formation of toxic nitrogen oxide byproducts[23-25].

In the past five years, significant advancements have been made in the field of alkaline $NO_3RR$ to $NH_3$ synthesis, utilizing a repertoire of catalysis, including Cu encapsulated in a porous carbon framework[26], Fe-based single-atom catalysts (SACs)[27], CuPd nanocubes[24], strained Ru nanoclusters[28], reduced-graphene-oxide-supported RuCu alloy[29], and $Cu_{50}Ni_{50}$ alloy[30]. In spite of continuous improvements in Faradaic efficiency (FE), a close-to-100% value was only reported under moderate $NH_3$ yield rates (<10 mg h$^{-1}$ cm$^{-2}$) and limited current density (<300 mA cm$^{-2}$)[28-31]. Further increasing the current density and production rate have been accompanied by a degradation of the FE[18,23]. Given the pressing challenge of wastewater containing $NO_3^-$ resulting from urbanization and population growth, alkaline electrocatalytic system for $NO_3^-$ reduction with both high $NH_3$ FE (>95%) and fast reaction rates (>1 A cm$^{-2}$) is of pivotal importance for a sustainable future, but remains elusive.

In this study, we design and realize a halogen-mediated alkaline electrocatalytic platform to overcome the limitation and achieve high-speed conversion of $NO_3^-$ to $NH_3$ while maintaining an ideal FE. Modifying Pd species on $Cu_2O$ platform with excellent $NO_3^-$ adsorption and conversion ability[32,33] could favor $H_2O$ dissociation under high pH conditions[34-36]. But the intense interaction between the $d$ orbitals of Pd and $s$ orbitals of *H (* denotes the adsorbed state) brings strong Pd-H binding[37], which affects the desorption of *H. Halogen elements with high first-electron-affinity[38] can tailor the $3d$ orbital electron structure of Pd atom to regulate *H release, thereby breaking the bottleneck of FE due to the scarce proton (H$^+$) feeding in high pH conditions. Here we develop Cl-coordinated Pd SACs-dispersed $Cu_2O$ matrix (Pd-Cl/$Cu_2O$) nanocrystal to carry out alkaline $NO_3RR$. As a result, we simultaneously achieve a $NH_3$ FE of ~100% with a current density of ~2 A cm$^{-2}$ for 1 h and $NH_3$ yield rate of ~330 mg h$^{-1}$ cm$^{-2}$ at 1 M $NO_3^-$ concentration (pH = 14), outperforming previous results with large current densities[23,39,40]. Impressively, our platform can reduce the $NO_3^-$ concentration from an industrial wastewater level of 56 mM to a drinkable water level (<0.8 mM), with an over 99.1% $NO_3^-$-to-$NH_3$ conversion efficiency. Further, we demonstrate a successful conversion of $NO_3^-$ into practical ammonia products with near-unity efficiency via coupling the $NO_3RR$ with an air stripping process. The combined results of in situ Raman spectroscopy, in situ attenuated total reflection infrared spectroscopy (ATR-IR), kinetic isotope effect (KIE) experiments, and theoretical simulations reveal that Cl-coordination induces a shifted $d$-band center

of Pd atoms to construct local H$^+$-abundant environments, through triggering the dissociation of dangling O-H water and fast *H desorption for *NO intermediate hydrogenation and finally efficient $NO_3^-$-to-$NH_3$ conversion (Fig. 1). This tactic can be extended to other halogen element Pd-(F, Br, I)/$Cu_2O$ for alkaline $NO_3RR$ to $NH_3$, demonstrating the wider applicability of the halogen-mediate strategy.

## Results
### Theoretical prediction
To test the halogen mediating effect, density functional theory (DFT) calculations were conducted on the models of $Cu_2O$, Pd-dispersed $Cu_2O$ (Pd/$Cu_2O$) and Pd-Cl/$Cu_2O$ with different Cl numbers (Supplementary Figs. 1, 2). $Cu_2O$ showed a high Gibbs free energy change for $H_2O$ dissociation ($\Delta G_{*H_2O}$ of 1.19 eV, Fig. 2a), while the corresponding value on Pd/$Cu_2O$ (0.70 eV) decreased, indicating the accelerated $H_2O$ dissociation under the assistance of Pd SACs. The introduction of Cl coordination endowed Pd-Cl/$Cu_2O$ model (optimized model of Pd atoms coordinated with two Cl, Supplementary Fig. 3) with a further decrease of $\Delta G_{*H_2O}$ (0.68 eV) and more favorable to generate H$^+$ ($\Delta G_{H^+}$ of −1.34 eV).

To investigate the influence of Cl coordination on the H$^+$ formation, differential charge distribution and the local density of states (LDOS) were calculated. From the charge density difference (Fig. 2b), the two Cl atoms on Pd-Cl/$Cu_2O$ both obtained 0.47 $e$, which was supplied by the coordinated Pd atom (0.13 $e$) in addition to the contribution from the $Cu_2O$ substrate. The Cl ligand with strong first-electron-affinity seized the electrons of Pd[38] and induced a downward shifted $3d_{xy}$ and $3d_{z^2}$-band center of Pd atom in Pd-Cl/$Cu_2O$ compared to that of Pd/$Cu_2O$, as well as a upward shifted $3d_{xz}$-band center of Pd (Fig. 2c). Thus, under the regulation of Cl ligand, the shifted $d$-band center of Pd enabled *H on the catalyst to obtain more electrons (0.13 $e$) and make *H more unstable to promote the H$^+$ release (Fig. 2b).

Next, after considering the effects of potential and pH on $NO_3RR$ pathway with multiple possible branches (Supplementary Figs. 4−6), the optimal pathway on $Cu_2O$, Pd/$Cu_2O$ and Pd-Cl/$Cu_2O$ models at the potential of −0.6 V vs. RHE for pH = 14 was proposed and the corresponding $\Delta G$ of each intermediate was calculated (Fig. 2d and Supplementary Figs. 7, 8). In such a sequential electron–proton transfer process (Supplementary Fig. 9)[41,42], the hydrogenation of *$NO_2$ into *$NO_2H$ (*$NO_2$ + $H_2O$ + $e^-$ → *$NO_2H$ + $OH^-$) was the potential-determining step (PDS), which involved a $\Delta G$ of −0.57 eV over pure $Cu_2O$. Pd/$Cu_2O$ also presented a relatively lower $\Delta G$ of PDS (−0.65 eV). Pd-Cl/$Cu_2O$ showed the lowest $\Delta G$ of PDS (−0.76 eV) and correspondingly advanced the progress of $NO_3RR$. Thus, Pd-Cl/$Cu_2O$ was anticipated as a promising candidate for alkaline $NO_3RR$ towards $NH_3$ synthesis.

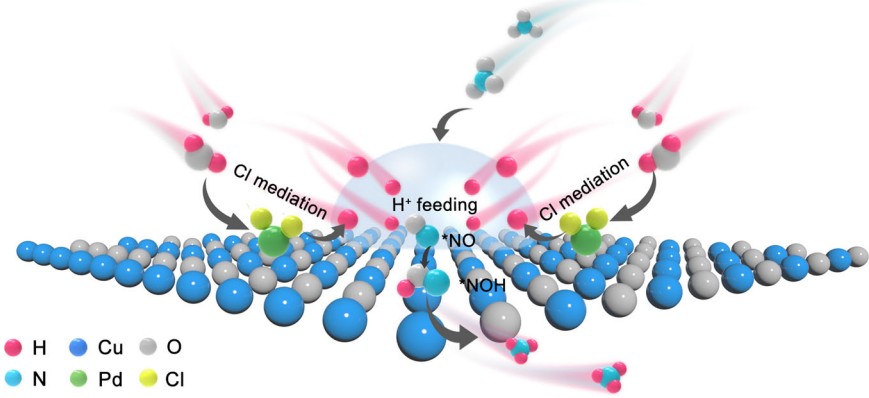

**Fig. 1 | Cl mediated H$^+$ feeding.** Schematic diagram of Cl mediated H$^+$ feeding to boost *NO intermediate hydrogenation and finally achieve efficient $NO_3^-$-to-$NH_3$ conversion in alkaline $NO_3RR$ over Pd-Cl/$Cu_2O$.

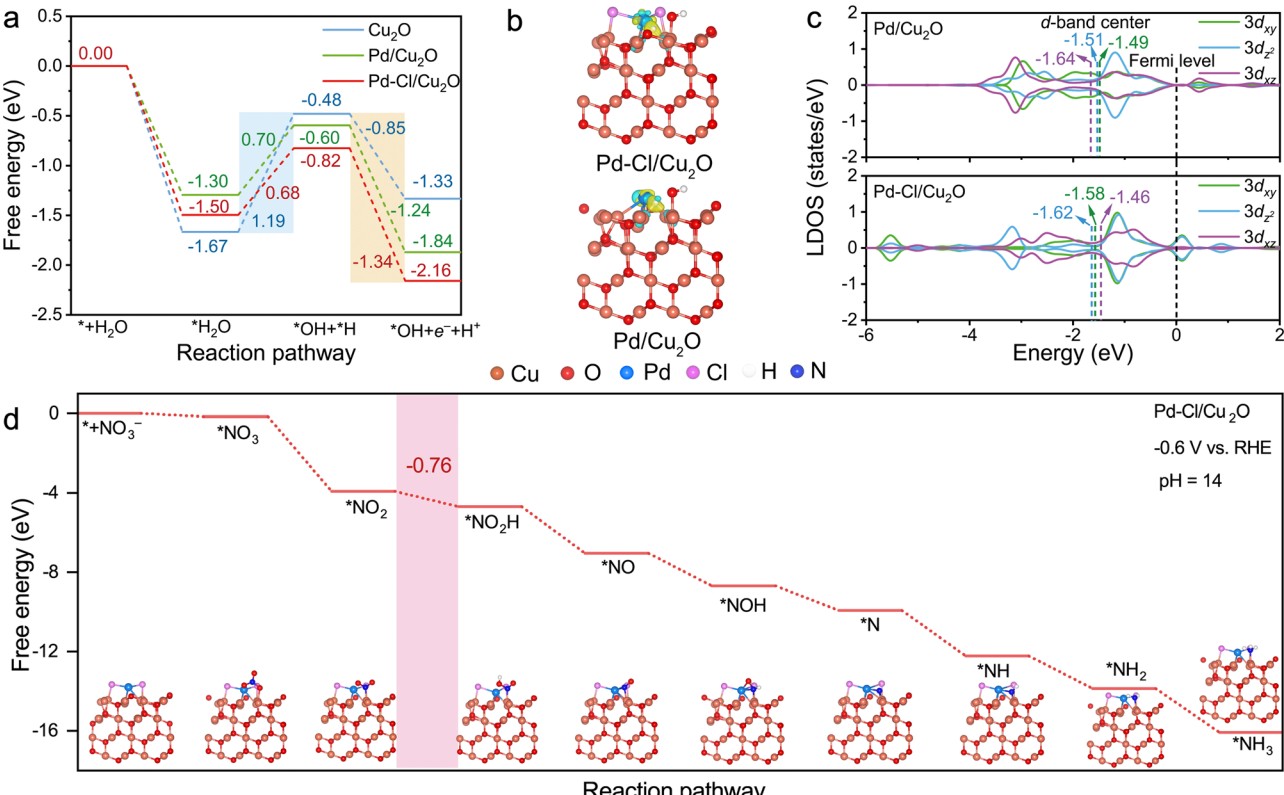

**Fig. 2 | Theoretical prediction. a** Gibbs free energy change of H$^+$ supply over catalysts. **b** The charge density difference between *H intermediate and catalysts. The isosurface level was 0.005 $e$·bohr$^{-3}$. The yellow and cyan colors represented positive and negative charge regions, respectively. 0.13 $e$ was the electrons transferred from Pd-Cl/Cu$_2$O to *H, and 0.09 $e$ was the electrons transferred from Pd/Cu$_2$O to *H. **c** The local density of states of Pd 3$d$ orbital for the Pd/Cu$_2$O and Pd-Cl/Cu$_2$O catalysts. **d** Gibbs free energy diagram of various intermediates generated during NO$_3$RR over Pd-Cl/Cu$_2$O at the potential of −0.6 V vs. RHE for pH = 14.

## Synthesis and structural characterizations of catalysts

Inspired by the theoretical results, we synthesized Cu$_2$O, Pd/Cu$_2$O and Pd-Cl/Cu$_2$O catalysts (see details in Methods). Scanning electron microscopy (SEM), transmission electron microscopy (TEM) and high-resolution TEM (HRTEM) showed Cu$_2$O had a lotus-like morphology on Cu foam with the orientation of (111) crystal planes (Supplementary Figs. 10–15). Pd-Cl/Cu$_2$O and Pd/Cu$_2$O maintained similar morphology to the initial Cu$_2$O (Fig. 3a and Supplementary Figs. 16, 17). X-ray diffraction (XRD) patterns only presented the peaks of Cu$_2$O and no Pd signals in these catalysts (Supplementary Fig. 18). The corresponding energy-dispersive X-ray (EDX) elemental mapping indicated the uniform distribution of Pd species on the Cu$_2$O matrix (Fig. 3b and Supplementary Fig. 19). The Pd appearing as the bright and isolated atoms was observed by aberration-corrected high-angle annular dark-field scanning transmission electron microscopy (AC-HAADF-STEM) images (Fig. 3c and Supplementary Fig. 20), confirming the successful preparation of Pd SACs in Pd-Cl/Cu$_2$O and Pd/Cu$_2$O.

To explore the electronic structure of Pd SACs, high-resolution X-ray photoelectron spectroscopy (XPS) measurements were performed. The Cu 2$p$ XPS and Cu LMM Auger spectra (Supplementary Fig. 21a, b) identified that Cu$^{1+}$ existed in all samples[43,44]. Pd 3$d$ spectra displayed the binding energies of Pd$^{δ+}$ (0 < δ < 2) species in Pd-Cl/Cu$_2$O positively shifted compared with Pd/Cu$_2$O (Supplementary Fig. 21c)[45], indicating the electron overflow on Pd in Pd-Cl/Cu$_2$O. In the Cl 2$p$ spectra, obvious Cl signals of Pd-Cl/Cu$_2$O proved the presence of Cl, while not observed in Pd/Cu$_2$O (Supplementary Fig. 21d). The binding energies of Cl peaks negatively shifted 0.3 eV compared to those of commercial PdCl$_2$ (Supplementary Fig. 22), revealing the electron-enriched Cl species in Pd-Cl/Cu$_2$O[46,47]. Combined with the results of Cl 2$p$ and Pd 3$d$ spectra, it can be concluded that the electron transferred

from Pd to Cl in Pd-Cl/Cu$_2$O under the Cl mediation, consistent with DFT calculation analysis.

To ascertain the coordination structure of catalysts, X-ray absorption fine structure (XAFS) was investigated. The Pd K-edge X-ray absorption near edge structure (XANES) spectra exhibited the pre-edge absorption energy of Pd-Cl/Cu$_2$O and Pd/Cu$_2$O located between those of Pd foil and PdCl$_2$ references (Fig. 3d and Supplementary Fig. 23a), implying the valence state of Pd within 0 to 2. The Fourier transformed (FT) extended X-ray absorption fine structure (EXAFS) suggested the absence of Pd-Pd scattering (2.5 Å) in both catalysts, verifying the single-atom dispersion of Pd. The main peak at 2.1 Å for the Pd-Cl/Cu$_2$O can be deconvoluted into 1.8 Å and 2.2 Å scattering (Fig. 3e), ascribing to Pd-Cl and Pd-Cu coordination structures, respectively. The quantitative least-squares best-fitting of EXAFS spectra (Fig. 3f and Supplementary Table 1) confirmed that Pd center was coordinated with ~2 Cl atoms and ~3 Cu atoms (Fig. 3f, inset). By comparison, Pd atom in Pd/Cu$_2$O only coordinated with ~3 Cu atoms (Supplementary Figs. 23b, c and Table 2). The high-resolution wavelet transform (WT) EXAFS plots in K spaces further demonstrate the existence of Pd-Cl coordination in Pd-Cl/Cu$_2$O, but absence in Pd/Cu$_2$O (Supplementary Fig. 24). These results suggested the successful synthesis of the Pd-Cl/Cu$_2$O and Pd/Cu$_2$O catalysts, as proposed in DFT simulation.

## NO$_3^-$ intermediates hydrogenation.

To probe H$^+$ feeding on halogen-mediated samples, in situ Raman spectra were carried out (Supplementary Fig. 25). For the Pd-Cl/Cu$_2$O (Fig. 4a), the stretching vibration peak of H$_2$O at 1615 cm$^{-1}$ appeared at 0.1 V vs. RHE and then shifted to 1605 cm$^{-1}$ as the potential decreased to −0.8 V vs. RHE, indicating a weaker H-bond of interfacial H$_2$O to enable H$_2$O dissociation[48-50]. Notably, the emerging H$_3$O$^+$ peak (1770 cm$^{-1}$) proved that the *H

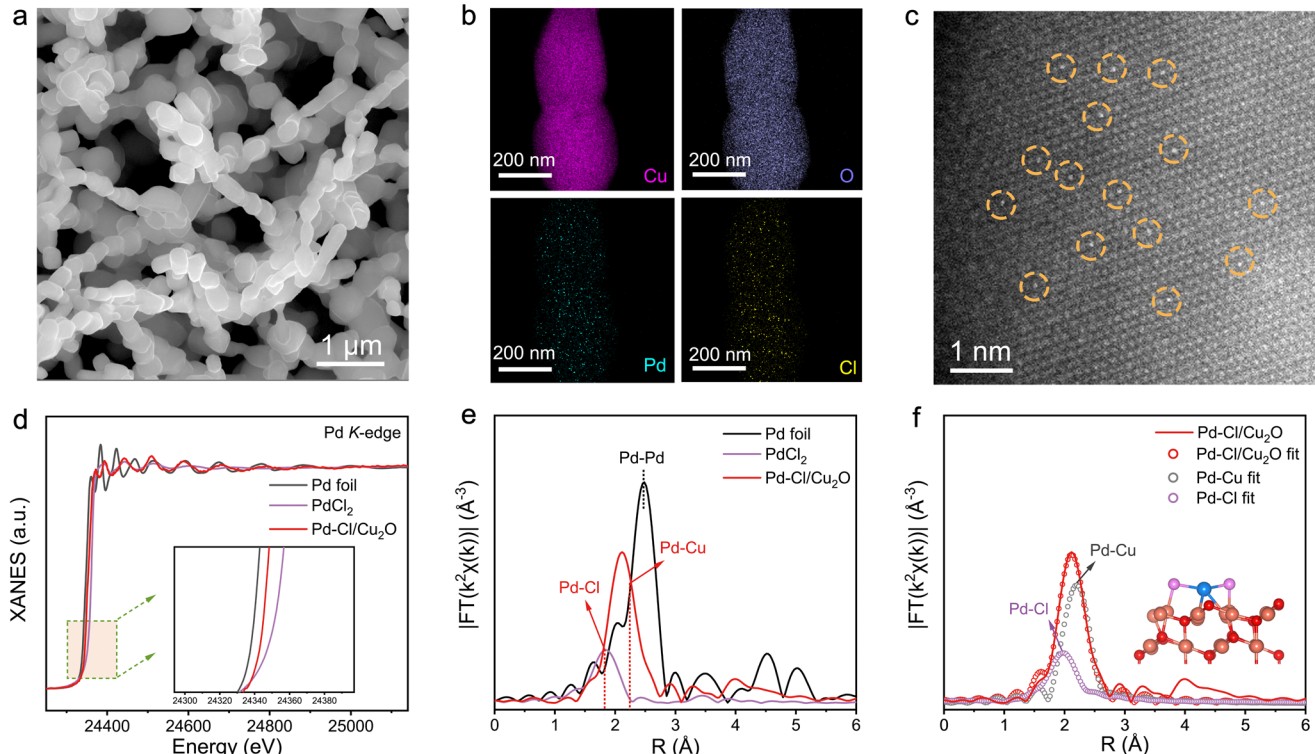

**Fig. 3 | Synthesis and structural characterizations of Pd-Cl/Cu₂O.** SEM (**a**), EDX mapping (**b**), and AC-HAADF-STEM images (**c**) of Pd-Cl/Cu₂O. Pd K-edge XANES spectra (**d**), and FT $k^2$-weighted EXAFS spectra (**e**) of Pd-Cl/Cu₂O and reference samples. **f** The fitting EXAFS spectra of Pd-Cl/Cu₂O. Inset: fitting model.

generated by $H_2O$ dissociation was immediately desorbed from Pd-Cl/ Cu₂O surface, to construct local $H^+$-abundant environments in high-pH conditions (Fig. 4a, b)[51,52]. The formed $H^+$ could stably accumulate on the local cathode surface. To distinguish the type of dissociated $H_2O$, we analyzed the envelope peaks at 3000-3700 cm$^{-1}$ which can be deconvoluted into three types of interfacial $H_2O$, including tetra-hedrally coordinated water (tetra-$H_2O$, 3230 cm$^{-1}$), trihedrally coordinated water (tri-$H_2O$, 3450 cm$^{-1}$) and dangling O-H bonds of water (dangling O-H, 3600 cm$^{-1}$), respectively[36,48,50]. The three types of interfacial water can vary as a function of electrode potential due to the Stark effect. At more negative potentials, the changes in peak intensity and shift of these interfacial water would accordingly become obvious. Compared to the tetra-$H_2O$ and tri-$H_2O$, dangling O-H water exhibited a smoother area change and a steeper shift slope (−23.0 cm$^{-1}$ V$^{-1}$) as the potential decreases (Fig. 4c-e), certifying the preferential dissociation of dangling O-H water on Pd-Cl/Cu₂O. Although rapid dissociation of dangling O-H water also occurred over Pd/Cu₂O (Supplementary Figs. 26, 27), the non-detected $H_3O^+$ peaks suggested the difficulty in donating $H^+$ (Supplementary Fig. 26a), due to the strong binding of *H on Pd without Cl mediation. In addition, the inapparent shift of $H_2O$ peak (1617 cm$^{-1}$) and the negligible $H_3O^+$ peak manifested the poor $H_2O$ dissociation with little $H^+$ coverage on Cu₂O (Supplementary Fig. 28). Therefore, under the mediation of Cl, Pd-Cl/Cu₂O catalysts could promote dangling O-H water dissociation to construct the local $H^+$-abundant environments in alkaline conditions.

To validate the effective $H^+$ for *NO intermediates hydrogenation in alkaline NO₃RR over catalysts, in situ ATR-IR was carried out (Supplementary Figs. 29-31). Under the driven of applied potential from 0.2 to −0.7 V vs. RHE, the detected N-O peaks (at 1540 cm$^{-1}$)[53] in the spectra of Pd-Cl/Cu₂O demonstrated the deoxygenation of NO₃$^-$ to the intermediate *NO. The conspicuous peaks of hydro-nitrogen intermediates (-NH) at 3200-3380 cm$^{-1}$ indicated the effective hydrogenation of *NO intermediates on Pd-Cl/Cu₂O[54-57]. In comparison, Pd/Cu₂O displayed the stronger *NO intermediate peaks and the weaker

-NH signals (Fig. 4f, g). These results evidenced that Cl-mediated $H^+$ feeding could boost *NO intermediates hydrogenation in alkaline NO₃RR for promising $NH_3$ synthesis over Pd-Cl/Cu₂O.

## Alkaline NO₃RR performance

The electrocatalytic NO₃RR performance was conducted under ambient temperature and pressure in a standard three-electrode H-type cell. $NH_4^+$, NO₃$^-$, and NO₂$^-$ in the reaction system were detected and quantified by colouration and $^1$H nuclear magnetic resonance (NMR) experiments (Supplementary Figs. 32-36). Given the common industrial and agricultural wastewater-relevant NO₃$^-$ concentration ranging from 40 to 80 mM[58-61], we reasonably selected 56 mM NO₃$^-$ in the electrolyte (pH = 14) for the standard electrochemical tests. Linear sweep voltammetry (Supplementary Fig. 37) curves of Pd-Cl/Cu₂O presented the distinct cathodic reduction peak between 0 and −0.5 V vs. RHE in the NO₃$^-$-containing electrolyte relative to NO₃$^-$-free solutions, expressing the underlying NO₃RR process. The NO₃RR performance was then determined by chronoamperometry (Supplementary Fig. 38). Pd-Cl/Cu₂O delivered an excellent $NH_3$ yield rate of 30.1 mg h$^{-1}$ cm$^{-2}$ with a corresponding $NH_3$ FE of 99.2% and a current density of 350 mA cm$^{-2}$ at −0.4 V vs. RHE (Fig. 5a and Supplementary Fig. 38c), in which the $NH_3$ yield rate value was 5-folds and 2.1-folds than that of Cu₂O and Pd/Cu₂O. The electrochemical surface area-normalized $NH_3$ yield also verified the best internal activity of Pd-Cl/Cu₂O (Supplementary Fig. 39, 40).

The KIE experiment which used $D_2O$ solvent for replacing the $H_2O$ in the electrolyte was tested to further investigate the NO₃RR performance (Fig. 5b and Supplementary Fig. 41)[62,63]. The Cu₂O presented a higher KIE value (ratio of $NH_3$ yield rate in $H_2O$ to $D_2O$) of 2.3, elucidating that the sluggish $H_2O$ dissociation limited the $H^+$ offer to hinder $NH_3$ activity. After introducing Pd SACs, the sharply decreased KIE over Pd/Cu₂O (1.23) and Pd-Cl/Cu₂O (1.05) corroborated the accelerated $H_2O$ dissociation, matching with the in situ Raman results. While, the lower $NH_3$ yield rate of Pd/Cu₂O than that of Pd-Cl/Cu₂O resulted from

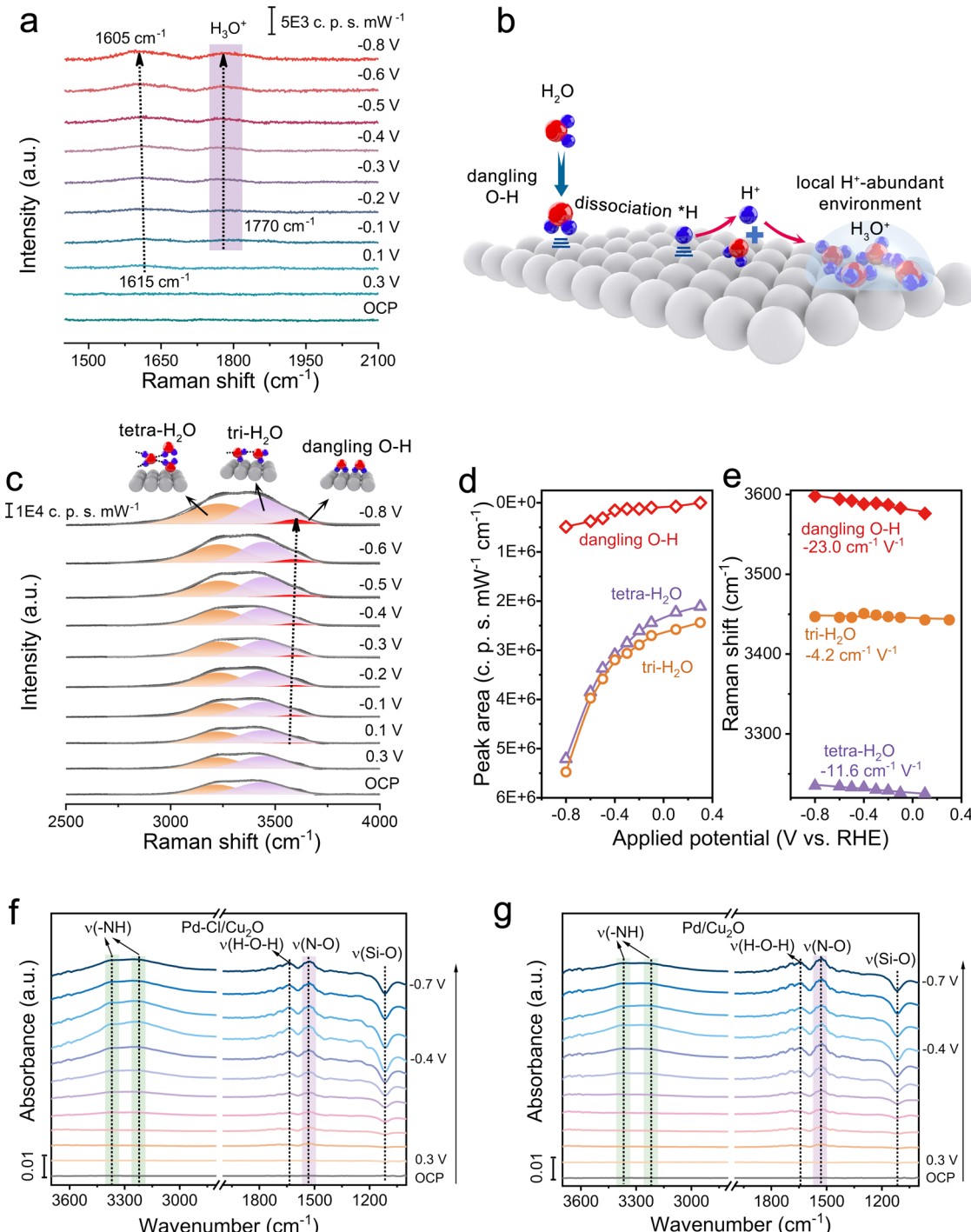

**Fig. 4 | NO$_3^-$ intermediates hydrogenation. a** In situ Raman spectra of Pd-Cl/Cu$_2$O. **b** Schematic diagram of the local H$^+$-abundant environment construction over Pd-Cl/Cu$_2$O. In situ Raman spectra of Pd-Cl/Cu$_2$O (**c**), corresponding peak area (**d**) and

Raman shift (**e**) of various interfacial H$_2$O structures. In situ ATR-IR spectra of Pd-Cl/ Cu$_2$O (**f**) and Pd/Cu$_2$O (**g**) catalysts. Si-O signal was derived from the reduction of surface SiO$_2$ on the Si semi-cylindrical prism substrate under the applied potentials.

the strong binding of *H on Pd, which induced the side reaction of hydrogen-hydrogen dimerization (Supplementary Figs. 41, 42). Under the Cl-mediated H$^+$ feeding effect, Pd-Cl/Cu$_2$O obtained superior alkaline NO$_3$RR performance. Control experiments further demonstrated that the mediated effect originated from the Cl ligand of Pd-Cl/ Cu$_2$O rather than the free Cl ions in the system (Supplementary Figs. 43, 44).

The NO$_3^-$ removal ability over the catalysts was examined by carrying out conversion tests under 56 mM NO$_3^-$ at −0.4 V vs. RHE. The three catalysts, Pd-Cl/Cu$_2$O, Pd/Cu$_2$O and Cu$_2$O, all showed a high NO$_3^-$

conversion rate of ~99%, indicating that Cu$_2$O matrix has a strong NO$_3^-$ removal ability. Yet, the NO$_3^-$ to NH$_3$ conversion rates of Cu$_2$O and Pd/ Cu$_2$O within 1 h electrolysis were as low as ~20% and ~45%, respectively, accompanied by producing 26% and 21% of NO$_2^-$ (Supplementary Fig. 45). Electrochemical online differential electrochemical mass spectrometry (DEMS) showed that Cu$_2$O and Pd/Cu$_2$O also generated the gas products during the potentiostatic process at −0.4 V vs. RHE (Supplementary Fig. 46a, b), including the $m/z$ signals of NH$_3$ (17), H$_2$ (2), N$_2$ (28), NO (30), NH$_2$OH (33), and N$_2$O (44). Notably, nearly all the NO$_3^-$ was converted into NH$_3$ on Pd-Cl/Cu$_2$O within 1 h electrolysis, and

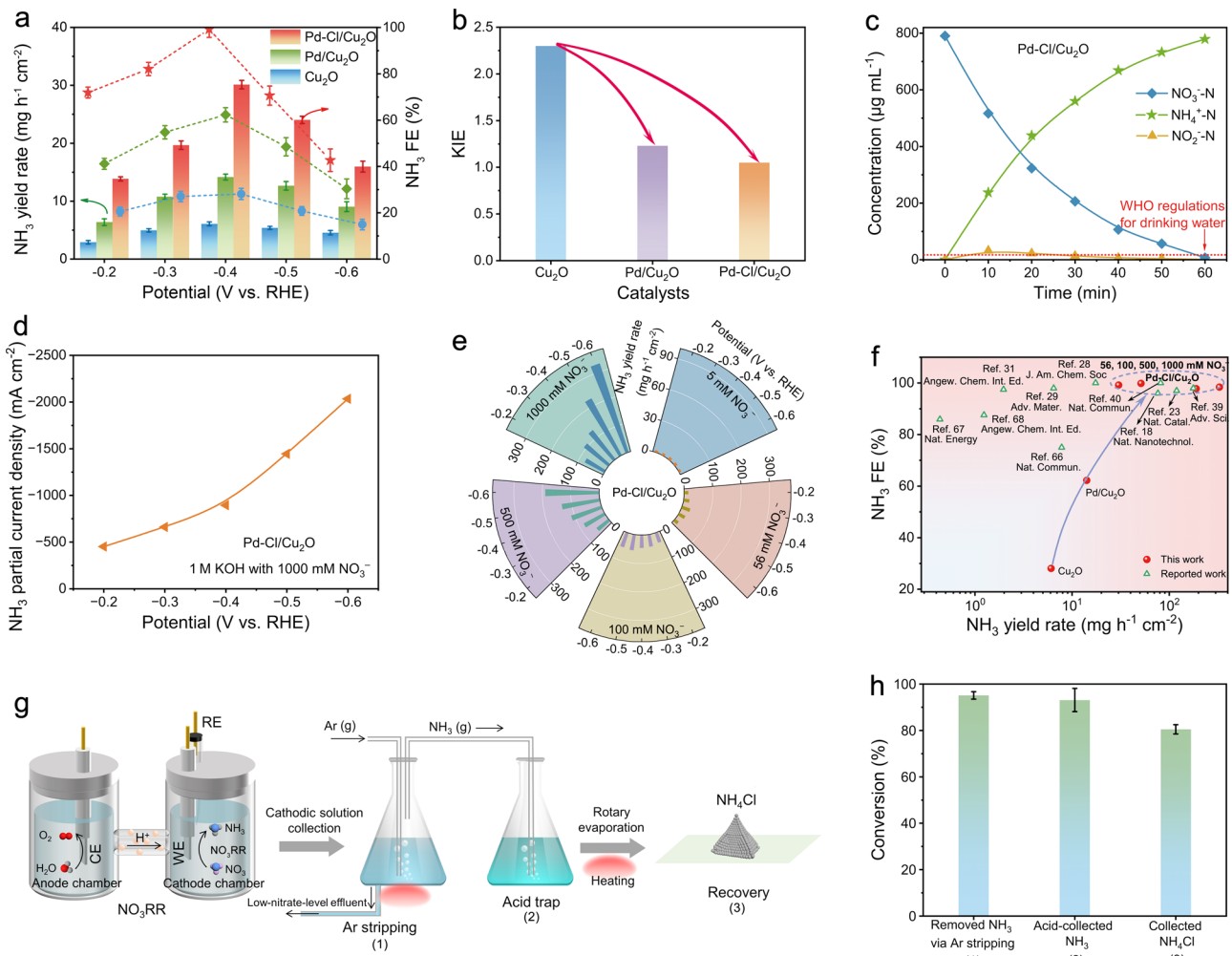

**Fig. 5 | Alkaline NO$_3$RR performance. a** NH$_3$ yield rate and NH$_3$ FE of catalysts in a 1 M KOH with 56 mM NO$_3^-$ electrolyte (pH = 14) for 1 h electrolysis. Catalyst mass loading: 3 mg cm$^{-2}$. Resistance of catalyst: 0.156 Ω cm$^{-2}$. Resistance of electrolyte: 1.45 Ω. **b** Kinetic isotopic effect (KIE) diagram for the ratio of NH$_3$ yield rate in H$_2$O to D$_2$O solvent in a 1 M KOH with 56 mM NO$_3^-$ electrolyte at −0.4 V vs. RHE. **c** NO$_3^-$ removal of catalysts measured in a 1 M KOH with 56 mM NO$_3^-$ electrolyte (equals 790.3 µg mL$^{-1}$ NO$_3^-$–N) at −0.4 V vs. RHE. After 1 h electrolysis, only 7.1 µg mL$^{-1}$ of NO$_3^-$–N and 0.85 µg mL$^{-1}$ of NO$_2^-$–N remained, both below the WHO regulations for drinking water (NO$_3^-$–N < 11.3 µg mL$^{-1}$ and NO$_2^-$–N < 0.91 µg mL$^{-1}$). **d** NH$_3$ partial

current densities of Pd-Cl/Cu$_2$O in a 1 M KOH electrolyte with 1000 mM NO$_3^-$ under the potential range from −0.2 to −0.6 V vs. RHE. **e** NH$_3$ yield rate of Pd-Cl/Cu$_2$O in a 1 M KOH electrolyte with different NO$_3^-$ concentrations for 1 h of electrolysis. **f** NO$_3$RR performance comparison of reported electrocatalysts. **g** Schematic of the ammonia product synthesis process from 1000 mM NO$_3^-$ electrolyte to NH$_4$Cl for 5 h electrolysis at −0.6 V vs. RHE. **h** The conversion efficiency of different steps for the ammonia product synthesis process. Numbers on the *x*-axis indicated the corresponding conversion steps in panel **g**. Error bars indicate the relative standard deviations of the mean (*n* = 3). See "Methods" for experimental details.

the corresponding selectivity reached ~99.1% with negligible NO$_2^-$ and gas products (Fig. 5c and Supplementary Fig. 46c). The residual NO$_3^-$ and NO$_2^-$ concentrations were both significantly lower than the World Health Organization (WHO) regulations for drinking water (Supplementary Fig. 47)[64,65]. These results corroborated the excellent NO$_3^-$ removal rate and high NH$_3$ selectivity of Pd-Cl/Cu$_2$O. In addition, Pd-Cl/Cu$_2$O behaved with favorable NO$_3$RR stability during ten-cycling tests (Supplementary Fig. 48). The morphology, phase structure, chemical valence state, atomic coordination environment and electrochemical properties remained steady after the electrolysis, further indicating the robust structure of catalysts (Supplementary Figs. 49–52 and Supplementary Tables 3, 4).

Furthermore, the optimal Pd-Cl/Cu$_2$O (Supplementary Table 5 and Supplementary Figs. 53, 54) was utilized to explore the NO$_3$RR performance at different NO$_3^-$ concentrations, due to the diverse pollutant sources with a broad scope of NO$_3^-$ concentrations[1,66]. Aside from 56 mM NO$_3^-$, 5, 100, 500, and 1000 mM NO$_3^-$ were also chosen to cover the concentration range expected in household and

heavy-industry wastewater (Supplementary Fig. 55, 56). The catalyst not only exhibited an excellent NH$_3$ FE (~97.8%) in 5 mM NO$_3^-$ system (Supplementary Fig. 57), but also preserved the high NH$_3$ selectivity of >95% under higher NO$_3^-$ concentration (100-1000 mM). Impressively, Pd-Cl/Cu$_2$O achieved an industrial-relevant NH$_3$ partial current density of ~2 A cm$^{-2}$ (Fig. 5d) while maintaining 99.1% NH$_3$ FE, with a splendid NH$_3$ yield rate of ~330 mg h$^{-1}$ cm$^{-2}$ at 1000 mM NO$_3^-$ concentration systems (Fig. 5e), which outperformed almost all state-of-the-art NO$_3$RR performance ever reported (Fig. 5f and Supplementary Table 6)[18,23,28,29,31,39,40,66–68].

With the impressive NO$_3$RR performance of Pd-Cl/Cu$_2$O catalyst, the high-purity ammonia products were collected to demonstrate their practical application potential (Fig. 5g). After conducting NO$_3$RR tests in a sealed reactor, the cathodic electrolyte was transferred into a conical flask. The generated NH$_3$ was striped at 70 °C by an Ar stripping method owing to the high NH$_3$ vapor pressure in the alkaline envir-onment (see details in Methods)[69,70]. As a result, ~95.2% of the NH$_3$ vapor was successfully stripped out from the electrolyte (Fig. 5h),

indicating water source denitrification with high efficiency and simultaneous production of high value-added $NH_3$. The outflowing $NH_3$ gas was trapped in a HCl solution (~93.1%), subsequently performed rotary evaporation, and finally collected ~80.5% of $NH_4Cl$ powder. While the high-purity $NH_4Cl$ powder (confirmed by XRD measurement in Supplementary Fig. 58) sheds light on the potential as a fertilizer for agricultural production, the limitations induced by additional cost of nitride concentration, interference (such as heavy metals and CODs) removal and product recovery[17] needs to be considered and overcome for practical ammonia products.

In addition to Cl mediated strategy, this performing principle with high $NH_3$ performance was available for other halogen ligand systems, such as Pd-F/$Cu_2O$, Pd-Br/$Cu_2O$ and Pd-I/$Cu_2O$. These catalysts were successfully synthesized using a similar wet-immersion and $H_2$ calcination approach, as proved by XRD and XPS characterization (Supplementary Figs. 59–62). Besides, Pd-(F, Br, I)/$Cu_2O$ catalyst exhibited a higher $NH_3$ activity and selectivity for alkaline $NO_3RR$ than those of $Cu_2O$ matrix and Pd/$Cu_2O$. Their $NH_3$ yield rates in a 1 M KOH with 56 mM $NO_3^-$ electrolyte at −0.4 V vs. RHE were ~3.7, ~3.1, ~2.6-fold of $Cu_2O$, and ~1.7, ~1.4, ~1.2-fold of Pd/$Cu_2O$, respectively (Supplementary Fig. 63a). The corresponding $NH_3$ FE value were ~2.9, ~2.3, ~2.1-fold of $Cu_2O$, and ~1.3, ~1.2, ~1.1-fold of Pd/$Cu_2O$, respectively (Supplementary Fig. 63b). Therefore, halogen-mediate strategy has expansive universality in enhancing alkaline $NO_3RR$ performance.

In summary, we have proposed and realized a highly efficient halogen-mediated $H^+$ feeding strategy to boost the $NO_3RR$ to $NH_3$ synthesis in alkaline conditions. Nitrate reduction to $NH_3$, instead of $N_2$, significantly enhances the product value of the electrochemical process. $NH_3$ can serve as a valuable nitrogen source in agricultural fertilizers, a chemical feedstock for various industrial processes, and a high-energy-density carrier for renewable hydrogen. The optimal Pd-Cl/$Cu_2O$ nanocrystals achieved a $NH_3$ partial current density of ~2 A $cm^{-2}$ while maintaining a nearly 100% $NH_3$ FE with a high $NH_3$ yield rate of ~330 mg $h^{-1}$ $cm^{-2}$ at 1 M $NO_3^-$ concentration (pH = 14). Our platform displayed an over 99.1% $NO_3^-$-to-$NH_3$ conversion efficiency from a typical industrial wastewater level to a drinkable water level. Further, it delivered a conversion of $NO_3^-$ into practical $NH_4Cl$ products with near-unity efficiency. Through a combination of theoretical simulations, in situ Raman, in situ ATR-IR, and KIE experiments, we have gained insights into the underlying mechanisms responsible for such highly selective and active $NH_3$ synthesis over Pd-Cl/$Cu_2O$. The presents of Cl ligand induced a prominent shift in the $d$-band center of Pd atoms, facilitating dangling O-H water dissociation and fast *H desorption. The constructed local $H^+$-abundant environments supported the free $H^+$ feeding to *NO intermediate hydrogenation, and thus realizing efficient $NO_3^-$-to-$NH_3$ conversion. The success of the halogen-mediated strategy presented in this study paves the way for the utilization of other halogen elements, such as F, Br, and I, in Pd-(F, Br, I)/$Cu_2O$ catalyst systems for alkaline $NO_3RR$ to $NH_3$. The broader applicability of this approach demonstrates its potential for achieving sustainable $NH_3$ synthesis in alkaline conditions and inspiring innovated design of environmentally friendly technologies in the field of water treatment and environmental remediation.

## Methods
### Chemicals
Salicylic acid ($C_7H_6O_3$), trisodium citrate dihydrate ($Na_3C_6H_5O_7 \cdot 2H_2O$), sodium hydroxide (NaOH), sodium hypochlorite (NaClO), hydrochloric acid (HCl, 38%), potassium nitrate ($^{14}KNO_3$), potassium nitrite ($KNO_2$), potassium iodide (KI), and ethanol ($C_2H_6O$) were purchased from Sinopharm Chemical Reagent Co., Ltd. Sulfanilamide ($C_6H_8N_2O_2S$), $p$-dimethylaminobenzaldehyde (PDAB), sodium nitroferricyanide (III) dihydrate ($Na_2Fe(CN)_5NO \cdot 2H_2O$), N-(1-Naphthyl)

ethylenediamine dihydrochloride ($C_{12}H_{14}N_2 \cdot 2HCl$), potassium hydroxide (KOH), potassium nitrate ($^{15}KNO_3$), dimethyl sulfoxide (DMSO-$d$6), maleic acid ($C_4H_4O_4$), ammonium chloride ($^{14}NH_4Cl$, $^{15}NH_4Cl$), palladium chloride ($PdCl_2$), palladium diacetylacetonate ($Pd(O_2CCH_3)_2$), palladium oxide (PdO), palladium bromide ($PdBr_2$), palladium nitrate $Pd(NO_3)_2$, and palladium(II) trifluoroacetate ($Pd(O_2CCF_3)_2$) were purchased from Aldrich Chemical Reagent Co., Ltd. All reagents were analytical reagent grades and used as received without further purification. The water used in this research was purified through a Millipore system.

### Preparation of $Cu_2O$
In a typical procedure, Cu foam (10 × 15 × 0.5 $mm^3$) was ultrasonically washed with acetone, 2 M HCl, ultrapure water and ethanol to clean the surface, respectively. The dried pre-treated Cu foam was then anodized in a 3 M KOH solution to form the blue $Cu(OH)_2$ nanowires by galvanostatic deposition at 20 mA $cm^{-2}$ for 900 s. The brick red lotus-like $Cu_2O$ was synthesized by annealing $Cu(OH)_2$ nanowires in a tube furnace at 550 °C for 2 h in $N_2$ atmosphere. The $Cu_2O$ substrate plays a dual role in the catalysis. On one hand, it effectively facilitates the adsorption, activation and conversion of nitrate. On the other hand, it serves as a platform for loading Cl-coordinated Pd single atoms and provides Cu active sites to boost the $NO_3RR$ process.

### Preparation of Pd-Cl/$Cu_2O$
A certain amount of 0.03 M $PdCl_2$ solution was immersed into the Cu foam-supported lotus-like $Cu_2O$ (10 × 10 $mm^2$), followed by reduction at 170 °C for 15 min in 10% $H_2$/Ar atmosphere to obtain the Pd-Cl/$Cu_2O$ catalyst. Pd-Cl/$Cu_2O$ catalysts with various Pd loadings were prepared according to the above procedure by changing the additional amount of $PdCl_2$ solution. The Cl acted as a synthetic directing agent to stabilize Pd single atoms.

### Preparation of Pd/$Cu_2O$
A certain volume of 0.03 M $Pd(O_2CCH_3)_2$ solution was soaked into the Cu foam-supported lotus-like $Cu_2O$ (10 × 10 $mm^2$), followed by reduction at 250 °C for 15 min in 10% $H_2$/Ar atmosphere to obtain the Pd/$Cu_2O$ catalyst.

### Preparation of Pd-F/$Cu_2O$, Pd-Br/$Cu_2O$ and Pd-I/$Cu_2O$
For the synthesis of Pd-F/$Cu_2O$, a certain amount of 0.03 M $Pd(O_2CCF_3)_2$ was dropped into the Cu foam-supported lotus-like $Cu_2O$ (10 × 10 $mm^2$), followed by heating at 250 °C for 15 min in 10% $H_2$/Ar atmosphere. The Pd-Br/$Cu_2O$ or Pd-I/$Cu_2O$ were synthesized under the same procedure except that $Pd(O_2CCF_3)_2$ was replaced by $PdBr_2$ or $Pd(NO_3)_2$ and KI.

### Electrochemical testing
Before the $NO_3RR$ tests, Nafion 117 membrane was pretreated as followed: first oxidizing in 5% $H_2O_2$ solution at 80 °C for 1 h to eliminate organic impurities, next boiling in deionized (DI) water for 1 h to clean the redundant $H_2O_2$ and reach the expansion, then using 0.5 M $H_2SO_4$ at 80 °C for 1 h to remove metallic impurities and residual ammonia contaminations and to protonate the membrane, finally operating DI water to rinse the excess acid and further expand the Nafion 117. The procedure should be repeated at least every 3 days to reuse the membrane.

The $NO_3RR$ was measured on an electrochemical workstation (PARSTAT 4000) with a three-electrode system in a typical H-type cell, including as-prepared catalyst electrodes (working electrode, WE), platinum electrode (counter electrode, CE), and a saturated calomel electrode (reference electrode, RE). Nafion 117 membrane was fixed between the anode and cathode cells, and each cell contained 32 mL of 1 M KOH with 56 mM $NO_3^-$ electrolyte). All potentials reported in this

work were referred to RHE scale via calibration by the following equation: $E$(vs. RHE) = $E$(vs. SCE) + 0.244 + 0.0591 × pHvalue. The theoretical potential for nitrate reduction to ammonia was 0.69 V vs. RHE at pH = 14[30,71]. The error bars were the relative standard deviations obtained by at least three repeated tests. The CV-activation before reaction was conducted to remove impurities on the electrode surface. For the chronoamperometry measurement, the potential was applied from −0.2 to −0.6 vs. RHE. LSV was carried out in a voltage window from 0.2 to −1.0 V vs. RHE at scan rates of 10 mV·s$^{-1}$.

### Detection and quantification of NH$_3$ using UV-vis

The concentration of NH$_3$ was spectrophotometrically detected by the salicylic acid method[72]. In detail, the electrolyte from the cathode cell was collected and diluted to the detection range. Then, 2 mL of diluted sample was mixed with 2 mL of 1 M NaOH solution containing 5 wt% salicylic acid and 5 wt% sodium citrate. Subsequently, 1 mL of 0.05 M NaClO and 0.2 mL of C$_5$FeN$_6$Na$_2$O solution (1 wt%) were added to the mixture component and shaken well. After stewing for 2 h, UV-vis spectrophotometer measurements were performed with the range from 500 to 800 nm and recorded the absorbance at the wavelength of 655 nm. The concentration-absorbance curve was calibrated using standard NH$_4$Cl solution with concentrations of 0.1, 0.5, 1.0, 2.0, 3.0, 5.0 and 10.0 μg mL$^{-1}$ in 1 M KOH with NO$_3^-$. And the fitting curve ($y = 0.104x - 0.014$, $R^2 = 0.999$) displayed a good linear relation of absorbance value with NH$_4^+$ concentration.

### Detection and quantification of NO$_3^-$ using UV-vis

The electrolyte from the cathode cell was collected and diluted to the detection range. 5 mL of diluted sample solution was mixed with 0.1 mL of 1 M HCl. After stewing for 20 min, the UV-vis absorbance at the wavelength ranging from 215 to 280 nm was detected[73,74]. The intensities at wavelengths of 220 and 275 nm were recorded, and the final absorbance difference was calculated using the equation: $A = A_{220 \, nm} - A_{275 \, nm}$. The concentration-absorbance difference curve was calibrated using standard KNO$_3$ solution with 5, 10, 15, 20, 25, 30 and 50 μg mL$^{-1}$ concentrations. And the fitting curve ($y = 0.051x + 0.013$, $R^2 = 0.999$) displayed a good linear relation of absorbance value with NO$_3^-$ concentration.

### Detection and quantification of NO$_2^-$ using UV-vis

The configuration of color reagent was as follows[73,74]: First, 0.5 g of sulfonamide was dissolved in 50 mL of 2.0 M HCl solution, which was marked as reagent A. Then, 20 mg of N-(1-naphthyl) ethylenediamine dihydrochloride was dispersed in 20 mL of DI water, which was denoted as reagent B. Subsequently, 0.1 mL of reagent A was dropped into 5 mL of standard or diluted sample solutions, mixing up and stewing for 10 min. Furthermore, 0.1 mL of reagent B was injected into the above solution, shaking up and resting for 30 min. The UV-vis absorbance at the wavelength ranging from 400 to 640 nm was recorded, in which the typical absorption peak of NO$_2^-$ was located at 540 nm. The concentration-absorbance difference curve was calibrated using standard KNO$_2$ solution with concentrations of 0.05, 0.1, 0.2, 0.5, 1.0, 2.0 and 3.0 μg mL$^{-1}$. And the fitting curve ($y = 0.768x - 0.012$, $R^2 = 0.999$) displayed a good linear relation of absorbance value with NO$_2^-$ concentration.

### Detection and quantification of NH$_3$ using $^1$H NMR

To support the UV-vis results, $^{14}$NO$_3^-$ and $^{15}$NO$_3^-$ isotope labeling experiments were conducted on Bruker AVANCE III HD NMR spectrometer (600 MHz). The pH value of the diluted electrolyte after NO$_3$RR was adjusted to 2 with 1 M HCl. Then 0.5 mL of the above solutions was mixed with 0.1 mL DMSO-$d6$ with 0.04% C$_4$H$_4$O$_4$, which served as a solvent and maleic acid C$_4$H$_4$O$_4$ as the internal standard. $^1$H NMR was recorded to quantitatively analyze of NH$_3$ product according to the corresponding standard curves.

### Electrochemical in situ Raman spectroscopy

In situ Raman was tested by inVia Reflex (Renishaw, UK) with a 633 nm laser as the excitation source. The NO$_3$RR was performed in the custom-made Teflon reactor with a quartz window, in which the Ag/AgCl (Supplementary Fig. 25), Pt wire, and catalysts coated on Au electrode were used for the reference, counter, and working electrode, respectively. In situ Raman spectra were recorded in electrolytes with NO$_3^-$ by the potential from open circuit potential (OCP) to −0.8 V vs. RHE.

### Electrochemical in situ ATR-IR spectroscopy

ATR-IR was measured on a Nicolet iS50 FT-IR spectrometer equipped with an MCT detector and cooled by liquid nitrogen during the electrochemical process (Supplementary Fig. 29). The NO$_3$RR was performed in the custom-made reactor with three-electrode, in which the Ag/AgCl and Pt wire were used for the reference and counter electrode, respectively. The working electrode was prepared as followed: First, the Si semi-cylindrical prism was polished with Al$_2$O$_3$ powder and sonicated in acetone and deionized water. The Si was pretreated in a piranha solution at 60 °C for 20 min to clean the organic contaminants. Then the reflecting surface of Si was plated in the Au precursor mixture at 60 °C for 10 min, obtaining the Au-coated Si (20 mm in diameter) conductive substrate. Finally, the catalyst ink was dropped on the substrate reflecting surface for employment in the reaction. In situ ATR-IR spectra were recorded in electrolyte with NO$_3^-$ by the potential from 0.3 V to −0.7 V vs. RHE. The spectrum collected at OCP was used for background subtraction.

### Electrochemical online DEMS tests

The online DEMS tests were performed in customized reactors containing 1 M KOH with 56 mM NO$_3^-$ electrolyte. Ar was continuously bubbled into the electrolyte. Catalysts coated on breathable film with gold plating layer, Pt wire, and saturated Ag/AgCl electrode were used as the working, the counter and the reference electrode, respectively. After the baseline of the mass spectrometry kept steady, the potential of OCP and −0.4 V vs. RHE were applied alternately with an interval of 3 min. Accordingly, the differential mass signals appeared when the gaseous products formed on the electrode surface. The mass signal returned to baseline after the electrochemical measurement was over. To avoid accidental errors, the next cycle started using the same conditions. After five cycles, the experiment was ended.

### Direct ammonia product synthesis

To evaluate the NH$_3$ removal efficiency via Ar stripping and the NH$_3$ collection efficiency by acid trap, 50 mL of cathodic electrolyte after NO$_3$RR test was sealed in a conical flask at 70 °C and flowed in 100 sccm Ar gas for 5 h to perform the Ar stripping to purge the NH$_3$ gas out. The outlet gas stream was meanwhile purged into 50 mL of 2 M HCl to collect the NH$_3$ product. The amount of NH$_3$ in all the solutions was measured by the salicylic acid method mentioned above, and the removal efficiency and collection efficiency were calculated as following equations, respectively:

$$\text{Removed NH}_3 \text{ via Ar stripping} = 1 - \frac{\text{NH}_3 \text{ left after Ar stripping (mol)}}{\text{initial NH}_3 \text{ (mol)}} \tag{1}$$

$$\text{Acid collected NH}_3 = \frac{\text{NH}_3 \text{ in acid trap (mol)}}{\text{removed NH}_3 \text{ via Ar } stripping \text{ (mol)}} \tag{2}$$

To produce the NH$_4$Cl product and estimate the production efficiency, the 50 mL of HCl with the trapped NH$_3$ was dried by rotary evaporator at 70 °C in an oven overnight. The final NH$_4$Cl was measured by a balance and analyzed by XRD. The collection efficiency of

$NH_4Cl$ from the acid trap was calculated by following equation:

$$\text{Collected NH}_4\text{ Cl from acid trap} = \frac{\text{collected dried out NH}_4\text{ Cl (mol)}}{\text{acid collected NH}_3\text{ (mol)}}$$

$$(3)$$

## DFT computational details

All calculations were carried out by spin-polarized DFT with the Vienna Ab initio Simulation Package (VASP)[75,76]. Electron exchange-correlation was expressed by the Perdew–Burke–Ernzerhof (PBE) functional within the generalized gradient approximation (GGA)[77]. To describe the ionic cores, the projector augmented wave (PAW) pseudopotential was applied[78,79]. The Monkhorst–Pack K-points were set to be $2 \times 2 \times 1$ for geometry optimization and density of states (DOS) calculations. The plane wave energy cutoff, and convergence criterion for electronic energy and forces were set as 450 eV, $10^{-5}$ eV, and 0.02 eV/Å, respectively (Supplementary Fig. 64). A vacuum layer of 15 Å was adopted in the models[80]. Aqueous phase $H_2O$ and $NO_3^-$ were as the energetics references.

$Cu_2O$ with crystal planes (111) was modeled with a periodic 4-layer, where the lower two layers were fixed and the upper two layers were relaxed. The model included 64 Cu atoms and 32 O atoms. The optimized lattice constants were a = 12.09 Å and b = 10.47 Å, the thickness of this model was 8.79 Å. The computational hydrogen electrode (CHE) model was used to calculate the change in Gibbs free energy ($\Delta G$)[81]. In CHE model, $H^+ + e^- \rightleftharpoons 1/2$ $H_2(g)$ was equilibrated at 0 V vs. the reversible hydrogen electrode (RHE) at all pH values.

Constant-potential calculations were applied using the code freely available from Duan and Xiao[82]. At the applied potential ($U$) on the standard hydrogen electrode (SHE) scale, the number of electrons and the atomic coordinates of the system are optimized simultaneously. The chemical potential of the electron ($\bar{\mu}_e$) is calculated as

$$\mu e = \mu e,\text{SHE} + U \qquad (4)$$

where $\mu_{e,\text{SHE}}$ is the electronic chemical potential of the system relative to the SHE.

$$\mu e,\text{SHE} = E_f/e - V sol + \varphi 0/e \qquad (5)$$

where $E_f$ is the Fermi level, $V_{sol}$ is the potential deep in the solution, and $\varphi_O = -4.6$ eV for the SHE. The grand canonical energy of the system is defined as

$$\Omega = E_{\text{DFT}} + \Delta n \cdot (U - V_{sol} + \varphi_0/\text{e}) \qquad (6)$$

where $E_{\text{DFT}}$ is the energy calculated from the DFT and $\Delta n$ is the number of electrons added or removed from the system.

The chemical potential of the electron ($\bar{\mu}_e$) is derived as

$$\overline{\mu}_e = \partial \Omega / \partial n = E_f/\text{e} - V_{sol} + \varphi_0/\text{e} + U \qquad (7)$$

where $\Omega$ is the grand canonical energy.

Calculation of thermodynamic corrections

The zero-point energy for each species is calculate by

$$E_{ZPE} = \sum_i \frac{h\nu_i}{2} \qquad (8)$$

where $\nu_i$ is the vibration frequency. The entropy contributions of translational, rotational, vibrational, and electronic motion can be calculated by

$$S_t = R\left\{ \ln\left[ \left(\frac{2\pi m k_B T}{h^2}\right)^{3/2} \frac{k_B T}{P} \right] + \frac{5}{2} \right\} \qquad (9)$$

$$S_r = R\left[ \ln\left(\frac{T}{\sigma_r} * \frac{8\pi^2 I k_B}{h^2}\right) + 1 \right] \qquad (10)$$

$$S_v = R\sum_i \left\{ \frac{h\nu_i}{k_B T} \frac{e^{-\frac{h\nu_i}{k_B T}}}{1 - e^{-\frac{h\nu_i}{k_B T}}} - \ln\left[1 - e^{-\frac{h\nu_i}{k_B T}}\right] \right\} \qquad (11)$$

$$S_e = R * \ln(N + 1) \qquad (12)$$

where $N$ is the number of unpaired electrons, $R$ is the gas constant, $P$ is the pressure, $k_B$ is the Boltzmann constant[83].

From the above formula, considering that the vibration frequency of the catalyst substrate is small, its corresponding correction is very small and does not affect the calculation results. Thus, we mainly made corrections to gas molecules and adsorbents on the catalyst, and entropic effects of the catalyst substrate would not be considered further.

$H_2O$ as the proton source in alkaline conditions. The pH has an influence on the major proton donor in the research system. For alkaline nitrate reaction pathways (pH = 14), we have considered $H_2O$ as the proton source. Under this condition, the H* path will be through the alkaline pathway.

$$H_2O + e^- + * \rightarrow H_2O^* + e^- \rightarrow H^* + OH^- \rightarrow 1/2H_2 + OH^- + * \qquad (13)$$

$$G_{OH^-} - G_{H_2O} - G_{e^-} = G_{1/2H_2} - eU_{RHE} \qquad (14)$$

For electrochemical steps in nitrate reduction, the free energy changes are calculated using the products and reactants of the following reaction equation:

$$H_2O + X^* + e^- \rightarrow XH^* + OH^- \qquad (15)$$

$$\Delta G(U) = G_{XH^*} - G_{X^*} + G_{OH^-} - G_{H_2O} - G_{e^-} = G_{XH^*} - G_{X^*} + G_{1/2H_2} - eU_{RHE} \qquad (16)$$

The calculation method for pH and potential effects. The free energies of adsorption of ionic species are calculated using thermodynamic Hess cycles, which cycles include the effects of entropy, solvation energy, protonation energy (including pH effects) and potential effects. The Gibbs energy formulas for nitrate reduction reaction steps are reported by Muhich et al.[16,18,84]. The method is based on that of Calle-Vallejo et al.[85] and Liu et al.[86]. The free energy of anion A$^-$ is calculated according to:

$$A^- + * \leftrightarrow A^* + e^- \qquad (17)$$

$$\Delta G_{\text{ads}}(A^-) = E_{A^*} + [G_{H^+} + G_{e^-}] - [G_{HA} - \Delta G_{\text{sol}} - \Delta G_{\text{protonation}}] - E^* \qquad (18)$$

Where $E_*$ and $E_{A^*}$ are the DFT computed enthalpies of bare surface and A* adsorbed to the surface, respectively. $G_{H_2}$ and $G_{HA}$ are the Gibbs free energies of desorbed species $H_2$ and HA, respectively, in the gas

phase at 300 K, as calculated from the following:

$$G_{HA} = E_{HA} + E_{ZPE} - T^*S \qquad (19)$$

where $E_{HA}$ is the DFT computed energy of HA in the gas phase, $T$ is the temperature (300 K), $E_{ZPE}$ is the contribution of the zero-point energy, $S$ is the entropic contributions to the free energy obtained using the JANAF database. The solvation energy is described:

$$\Delta G_{sol} = G_{HA(g)} - G_{HA(l)} \qquad (20)$$

The pH accounts for the effects on the free energies of the species. Free energy modifications due to pH were calculated according to:

$$\Delta G_{protonation} = G^O - 2.303kT(pK_a - pH) = G_{A^-} + G_{H^+} - G_{HA(l)}$$
$$- 2.303kT(pK_a - pH) \qquad (21)$$

$G_{HA(g)}$ and $G_{HA(l)}$ are the free energies of HA molecule in the gas and liquid phases respectively. $k$ is the Boltzmann constant. $K_a$ is the acid dissociation constant for the A$^-$ anion. The standard state (25 °C, 100 k$Pa$, 1 mol/kg) energies of ion and neutral species in aqueous solution ($G_{HA(g)}$, $G_{HA(l)}$, $G_{A^-}$, $G_{H^+}$, $K_a$) are taken from the CRC handbook.

The computational hydrogen electrode (CHE) is used to account for potential effects on reaction energies[87]:

$$\Delta G = \Delta E + \Delta E_{ZPE} - T^*\Delta S + 0.0591^*pH - eU_{RHE} \qquad (22)$$

where $\Delta E$ is the DFT computed reaction (electronic) energy, $\Delta E_{ZPE}$ and $\Delta S$ are the zero-point energy difference and the entropy difference between the adsorbed state and the gas phase, respectively. 0.0591*pH represents the free-energy contribution due to the variations in H concentration. We considered the effect of a potential bias on all states involving one electron or hole in the electrode by shifting the energy of this energy by $eU_{RHE}$, where $U_{RHE}$ is the electrode potential relative to the reversible hydrogen electrode (RHE).

## Data availability
All experimental data reported in this study and Supplementary Information are available from the corresponding author upon reasonable request.

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

## Acknowledgements

We gratefully thank the National Natural Science Foundation of China (Grant No. 22376222 for M.L., 52372253 for J.F., 22022602 for W.Liu), the Science and Technology Innovation Program of Hunan Province (2023RC1012 for M.L.), Central South University Research Programme of Advanced Interdisciplinary Studies (Grant No. 2023QYJC012 for M.L.), and Central South University Innovation-Driven Research Programme (Grant No. 2023CXQD042 for J.F.), and China Postdoctoral Science Foundation (2023T160735 and 2022M723547 for W.Liao). We are grateful for resources from the High Performance Computing Center of Central South University. We would like to acknowledge the help from Beam Lines BL01C1 in the National Synchrotron Radiation Research Center (NSRRC, Hsinchu, Taiwan) for various synchrotron-based measurements.

## Author contributions

M.L., W.Liao, J.W., C.L. and J.F. conceived the research and supervised the project. W.Liao and J.W. performed the experiments. M.L., W.Liao, J.W., S.C., Q.W., Y.C., T.L., X.W., Y.W., W.Li, T.C., C.M., H.L., Y.L., W.Liu, J.F. and B.X. analyzed the data. K.L. and G.N. carried out the DFT simulations. W.Liao and C.L. wrote the draft. All authors discussed the results and contributed to the writing of the manuscript.

## Competing interests

The authors declare no competing interests.
