## [Peer Review File · Nature Communications]

REVIEWER COMMENTS

Reviewer #1 (Remarks to the Author):

The manuscript by Liao et al. describes the use of H_3O^+ intermediates on Pd-Cl/Cu₂O under alkaline conditions to improve its nitrate reduction to ammonia performance. Nitrate electrochemical reduction is receiving much attention from the research community as a green alternative for ammonia production. Despite the interest in the reaction reported, fundamental aspects are not properly addressed, making the quality of the manuscript below what is expected for Nature Communications.

First, the major drawbacks to the synthesis of ammonia from nitrates in polluted waters are its low concentration and the various interference, even toxic ions, like heavy metals and CODs (Joule 2021, 5, 290-294). It is unreasonable to state the fertilizer production prospect without considering the separation costs for raw materials and the products.

Another critical aspect is novelty. Amper level current density even (2 A cm⁻²) has been obtained at lower overpotential by several studies (Adv Sci (Weinh) 2021, 8, 2004523; Nat. Catal. 2023, 6, 402-414; Nat. Commun. 2022, 13, 7899.). Although the catalysts are different, the approach is similar, significantly reducing the novelty of the described work.

A collection of (some) other issues to address is given below:

-How does the proposed Pd-Cl/Cu₂O catalyst compare to the CuCo or Co(OH)₂ catalysts for nitrate reduction? Perhaps such details can be added to your supplemental table 5 and Fig. 5f to help complete the comparison against recent literature (Adv Sci (Weinh) 2021, 8, 2004523; Nat. Commun. 2022, 13, 7899.).

-In the abstract, the statement that “nitrate reduction to ammonia suffers from low catalytic activities, especially in high pH conditions” is not rigorous. As listed in supplementary table 5, most catalytic systems with remarkable performance are under alkaline conditions.

-In the DFT calculation part, more evidence should support the view that nitrate reduction over the Pd-Cl/Cu₂O catalysts goes through multiple proton-coupled electron transfer processes instead of sequential proton–electron transfer processes (Nat. Catal. 2022, 5, 798-806; Chem. Sci. 4, 2710–2723 (2013)). The specific results will significantly affect the accuracy of DFT calculation.

-As the author stated, the DFT calculation results inspired the experimental parts. Why did the author choose Cu₂O and Pd-modified Cu₂O as catalysts? Why did the author select Cl as the coordination element instead of F, Br, I, O, or S? The logic is confusing.

-In the in situ ATR-IR part (Figs. 4f, g, and S22), the assignment of N-related intermediates should be reconsidered. If the IR spectra of the standard sample (for example, nitrate, nitrite, NO, and ammonia) can be supplemented, the assignment would be more persuasive.

-To state the conclusion that “Cl-mediated H⁺ feeding could boost *NO intermediates hydrogenation in alkaline NO₃RR for promising NH₃ synthesis over Pd-Cl/Cu₂O.”, the in situ ATR IR spectra of Pd-Cl/Cu₂O (Fig.4f) should compare with Pd/Cu₂O (Fig. S22), instead of Cu₂O (Fig.4g).

-In Fig 5a and other performance-related figures, the total Faradaic efficiency is far from ~100%. How about the other products, such as N₂, N₂O, NH₂OH, H₂, etc? It's recommended to provide a more complete product distribution map.

Reviewer #2 (Remarks to the Author):

In this work, Liu et al. proposed a universal halogen-mediated H⁺ feeding strategy to enhance alkaline NO₃RR to ammonia, achieving near-100% NH₃ Faradaic efficiency (at pH = 14) with a record current density of 2 A cm⁻² and enabling an over 99% NO₃⁻ to NH₃ conversion efficiency. The authors performed in situ spectroscopy experiments and DFT calculations to reveal the role of halogen ligands, which constructed local H⁺-abundant environments for *NO intermediate hydrogenation and finally effective NO₃⁻ to NH₃ conversion. This is an interesting study that will appeal to a broad audience, as it incorporates elements of data science, computational chemistry, and experiments all to study an important application. Therefore, I recommend this work for publication after pending minor revisions. Some specified comments are listed as follows:

1. Electrochemical nitrate reduction has potential application prospects for wastewater treatment, and both N₂ and NH₃ as the targeted products have been reported in many published literatures (Environ. Sci. Technol. 56, 614-623 (2022); Nat. Nanotechnol. 17, 759-767 (2022)). Then what are the significance and advantages of NH₃ as the main product compared to N₂?
2. In Fig. 2d, the free energy change for the desorption of *NH₃ (*NH₃ → NH₃) is larger than that of the hydrogenation of *NO into *NOH (*NO + H⁺ → *NOH) in pure Cu₂O and Pd/Cu₂O. The authors should check the expression of the potential-determining step (PDS) whether it is suitable for the Cu₂O and Pd/Cu₂O catalysts.

3. In the manuscript, the authors select the Pd-Cl/Cu₂O as the catalyst for highly efficient NO₃RR to ammonia. The effect of Pd and Cl components has been well-elaborated in which Pd accelerates the H₂O dissociation, and the Cl ligand facilitates the desorption of *H for H⁺ feeding. Besides, what is the role of Cu₂O substrate?

4. In the discussion of Fig. 4d-e, the author claimed “At more negative potentials, dangling O-H water exhibited a smoother area change and a steeper shift slope (-23.0 cm⁻¹ V⁻¹) compared to the other two interfacial H₂O structures, certifying the preferential dissociation of dangling O-H water on Pd-Cl/Cu₂O”. The authors should provide a more detailed discussion to help the readers to understand this conclusion.

5. The authors have provided abundant characterizations of the catalysts before and after NO₃RR to ammonia, to verify the structural stability. Although the NO₃RR performance stability over Pd-Cl/Cu₂O has been demonstrated for successive 10 cycles, more experimental data are expected to support the durability of the catalysts, especially the i-t curves for each cycle.

6. There are some minor comments and suggestions. 1) The authors should use the formula editor to unify the format of the formulas in the Methods part. 2) The ordinate of Fig. 5d should be modified to “NH₃ partial current density”. 3) The color identification of N element needs to be added in Fig. 2d.

7. For the integrity of the research background, more recent works about NH₃ electrosynthesis are suggested to be referred: J. Colloid Interface Sci. 2023, 638, 650; Front. Chem. Sci. Eng. 2023, 17, 726; Nano Res. Energy 2022, 1, e9120022; Chem. Commun. 2022, 58, 8097; Inorg. Chem. Commun. 2023, 151, 110621.

8. To further confirm the electrochemical stability of the catalyst, the EIS spectra before and after the long-term electrolysis are suggested to be added.

Reviewer #3 (Remarks to the Author):

The authors report on the high performance characteristics of Pd on CuO₂ with bound Cl ions to catalyze the nitrate reduction reaction to NH₃. This reaction is very important in terms of both environmental remediation and sustainable synthesis of ammonia, a crucial commodity chemical. While the experimental results and efficiencies are impressive, there are several major concerns that are present, especially among the computational work which must be addressed. Without these additions, the work is not fully substantiated nor is the methodology fully sound.

Below I first outline key computational chemistry concerns, then add some additional questions and comments.

The computational work has some key holes which need to be addressed before the findings are publishable.

1. The choice of k-point sampling and cut-off energy are not justified. There is no indication how the results compare to more rigorous computations (i.e. a 4 X 4X 1 finer k-point mesh or say 500 eV cut-off energy). Without these numbers it is impossible to discern the quality of the results
2. The authors state that that the computational hydrogen electrode was used. However, they never state what potential was used in the CHE model. This is crucial as the behavior of electrochemical systems are highly dependent on potential.
3. There is no mention about the pH used in the calculations. The effects of pH have a drastic effect on adsorption energies of different species, particularly nitrate.
4. The authors report that they have calculated reaction pathways based on free energies. However, based on the results presented, this seems highly dubious. Nowhere is it mentioned how they calculate the zero point energy or the entropy of the reactions. The required phonon calculations are very expensive and are easy to get wrong in periodic systems. Therefore, details must be provided as to the conditions/methods used.
5. It is unclear in the reaction pathway if the authors used H^* as the reactant for protonation steps or solution phase H^+ . In their case, the pH (which again is not explicitly described or seemingly included in the energetic models) will have a large effect on the energetics of proton transfer.
6. The model for the Pd-Cl is under described. For example, how many Cl ions were included, why was this chosen, what is the effect of more or fewer Cl ions, how was the negative charge on the Cl accounted for? None of these are described let alone justified. Without these answers, the validity of the results cannot be assessed.
7. It is unclear what the reference states of the molecules are, for example, are the energetics references to H_2O and NO_3^- in the gas phase or some sort of aqueous reference.
8. In terms of the results themselves, there are some questions. The authors state that the pure Pd is limited by the large H^* dissociation free energy (0.69 eV). This is confusing as once formed and on the surface the H^* should be able to react (at least thermodynamically) with the nitrate intermediate. Therefore the H desorption barrier seems irrelevant.
9. Similarly, examining the dissociation of H-OH on the Pd-Cl, the entire processes is only downhill from solution phase H_2O by 0.1 eV. This is very small, especially if the H_2O is referenced to the gas phases, which would significantly over estimate the binding compared to a proper solution phase reference point.
10. The NO_3 reduction pathway is highly complex with multiple possible branches. The authors need to describe how they choose this particular pathway and if they considered alternative pathways. If they did not, then it is very possible that these results are uninformative as to the chemistry at play.
11. In Figure 2b, the authors need to elaborate what the 0.03 e- is, it is unclear.

In general, it would be helpful early on to establish that PdCl samples are generated by having Cl in the solution when synthesizing the material.

It is unclear if the Cl effect arise from the mere presence of Cl or from the synthesis. To confirm one way or the other, additional experiments where Cl ions are added post synthesis to the Pd/Cu₂O samples

The discussion of the effect of other halogen mediation is weak.

Given the use of single atom catalysis for the work, it is highly surprising that the authors did not reference key works on single atom catalysis for nitrate reduction, specifically by Wu et al¹ and Chen et al.² Here the form examined Pd single atoms in Cu, reflecting the current work, and the latter used Ru in Cu while coupling electrochemical conversion to NH₃ recovery in a production train highly parallel to that in this work. At a minimum, these contributions should be noted.

1. Wu, X.; Nazemi, M.; Gupta, S.; Chismar, A.; Hong, K.; Jacobs, H.; Zhang, W.; Rigby, K.; Hedtke, T.; Wang, Q.; Stavitski, E.; Wong, M. S.; Muhich, C.; Kim, J.-H., Contrasting Capability of Single Atom Palladium for Thermocatalytic versus Electrocatalytic Nitrate Reduction Reaction. *ACS Catalysis* 2023, 13, (10), 6804-6812.

2. Chen, F.-Y.; Wu, Z.-Y.; Gupta, S.; Rivera, D. J.; Lambeets, S. V.; Pecaut, S.; Kim, J. Y. T.; Zhu, P.; Finprock, Y. Z.; Meira, D. M., Efficient conversion of low-concentration nitrate sources into ammonia on a Ru-dispersed Cu nanowire electrocatalyst. *Nature nanotechnology* 2022, 1-9.

Manuscript number: NCOMMS-23-34364-T

Title: Sustainable conversion of alkaline nitrate to ammonia at activities greater than 2 A cm⁻²

A Point-to-Point Response to Reviewer's Comments

Dear Editor and Reviewers,

Thank you for taking time and effort to carefully examine our manuscript. The comments are highly appreciated and helpful to improve this work. We have incorporated the corresponding changes into the manuscript (highlighted in the revised manuscript) and supporting information in order to address the editor's concerns and the requests of the reviewers. These changes are specified and discussed in a point-to-point response to the reviewers' comments, as shown below:

***Reviewer #1:** The manuscript by Liao et al. describes the use of H₃O⁺ intermediates on Pd-Cl/Cu₂O under alkaline conditions to improve its nitrate reduction to ammonia performance. Nitrate electrochemical reduction is receiving much attention from the research community as a green alternative for ammonia production. Despite the interest in the reaction reported, fundamental aspects are not properly addressed, making the quality of the manuscript below what is expected for Nature Communications.*

Response: We sincerely appreciate that the reviewer finding our work interesting and providing several insightful comments that have helped to improve the quality of the manuscript. We have diligently considered each of the reviewer's suggestions and have made corresponding revisions as shown below.

***Comment 1:** First, the major drawbacks to the synthesis of ammonia from nitrates in polluted waters are its low concentration and the various interference, even toxic ions, like heavy metals and CODs (Joule 2021, 5, 290-294). It is unreasonable to state the*

fertilizer production prospect without considering the separation costs for raw materials and the products.

Response: Thanks for raising the question about the statement related to fertilizer production. We modified the statement according with additional information for the clarification of the limitations suggested.

As the reviewer mentioned, the conversion of nitrate from actual industrial and agricultural wastewater into ammonia through electrochemical means has problems such as low nitrate concentration and various interferences. To achieve the application of the nitrate conversion in wastewater to ammonia, we mainly consider the cost from three aspects: 1) Concentrate nitrate. 2) Remove interference. 3) Recover product. Hence, we agree with the opinion that the separation costs for raw materials and products are very important indicators before stating the fertilizer production prospect. We have added more related information and cited the reference (Joule 5, 290-294 (2021), #17) both in the discussion and introduction part.

On the other hand, our work is still in the basic research stage mainly committed to addressing the issues of *electrochemical* nitrate reduction in alkaline systems from a scientific perspective. Through the halogen-mediated proton feeding strategy, the problem is effectively solved and the catalysts achieve a high synthetic ammonia performance in the laboratory. Overcoming the challenge of these costs in industrial applications, and attempting to apply the designed catalyst in nuclear wastewater and other application scenarios will be quite interesting for a following project and may be out of the scope of the current paper.

Corresponding revision:

➤ The corresponding content has been revised to “The outflowing NH_3 gas was trapped in a HCl solution (~93.1%), subsequently performed rotary evaporation, and finally collected ~80.5% of NH_4Cl powder. While the high-purity NH_4Cl powder (confirmed by XRD measurement in Supplementary Fig. 57) sheds light on the potential as a fertilizer for agricultural production, the limitations induced by additional cost of nitride concentration, interference (such as heavy metals and CODs) removal and product recovery¹⁷ needs to be considered and overcome for practical ammonia products.

Overall, Pd-Cl/Cu₂O catalysts delivered a complete process that directly converted NO₃⁻-containing influent into practical ammonia products in alkaline conditions.”

(Page 17, Lines 312-321, Revised manuscript).

Comment 2: *Another critical aspect is novelty. Amper level current density even (2 A cm⁻²) has been obtained at lower overpotential by several studies (Adv. Sci. 2021, 8, 2004523; Nat. Catal. 2023, 6, 402-414; Nat. Commun. 2022, 13, 7899.). Although the catalysts are different, the approach is similar, significantly reducing the novelty of the described work.*

Response: We thank the reviewer for raising the concern about the novelty (ample level current density) and clarified it as the following with a direct quantitative comparison (Figure R1).

For the performance, considering the novelty resulted from the large current density, we implemented a quantitative comparison among our work and references suggested (Figure R1). We found that the maximum current densities of Cu₅₀Co₅₀ nanosheets and Ru₁₅Co₈₅ hollow nanododecahedrons (Nat. Commun. 13, 7899 (2022); Nat. Catal. 6, 402-414 (2023).) are approximately 1000 mA cm⁻² at -0.4 V vs. RHE, which do not exceed the value in our work of ~2200 mA cm⁻² at -0.6 V vs. RHE. Although the reported electroreduction Co(OH)₂ nanoarrays (Adv. Sci. 8, 2004523 (2021)) exhibit a current density of 2200 mA cm⁻² at -0.24 V vs. RHE with NH₃ Faradaic efficiency (FE) of 98% and NH₃ yield rate of 10.4 mmol h⁻¹ cm⁻², its reaction time only runs 200 s. Comparatively, our designed Pd-Cl/Cu₂O nanocrystals maintain a current density of ~2200 mA cm⁻² for 3600 s, and deliver a higher NH₃ FE of 99.1% and more excellent NH₃ yield rate of ~19.4 mmol h⁻¹ cm⁻². Accordingly, our platform displays the performance advantages. Also, we have cited these references (#23, #39, #40) for comparison in the revised manuscript.

Meanwhile, the strategy proposed in this work (a halogen-mediated proton feeding) can feasibly be extended to other halogen element Pd-(F, Br, I)/Cu₂O systems. The

broad applicability of this approach demonstrates its potential for achieving sustainable NH_3 synthesis in alkaline conditions and inspiring innovated design of environmentally friendly technologies in the field of water treatment and environmental remediation.

Figure R1. Time-dependent current density curves of different catalysts at various potentials for electrocatalysis. **a**, Pd-Cl/ Cu_2O (Our work). **b**, Electroreduction $\text{Co}(\text{OH})_2$ nanoarrays (*Adv. Sci.* **8**, 2004523 (2021)). **c**, $\text{Cu}_{50}\text{Co}_{50}$ (*Nat. Commun.* **13**, 7899 (2022)). **d**, $\text{Ru}_{15}\text{Co}_{85}$ (*Nat. Catal.* **6**, 402-414 (2023)).

Corresponding revision:

- The corresponding content has been revised to “we simultaneously achieve a NH_3 FE of $\sim 100\%$ with a current density of $\sim 2 \text{ A cm}^{-2}$ for 1 h and NH_3 yield rate of $\sim 330 \text{ mg h}^{-1} \text{ cm}^{-2}$ at 1 M NO_3^- concentration (pH 14), outperforming previous results with large current densities^{23,39,40}.” (Page 5, Lines 90-93, Revised manuscript).
- Figure R1a has been added to **Supplementary Figure 55** (Page 60, Revised supplementary information).

A collection of (some) other issues to address is given below:

Comment 3: *How does the proposed Pd-Cl/ Cu_2O catalyst compare to the CuCo or $\text{Co}(\text{OH})_2$ catalysts for nitrate reduction? Perhaps such details can be added to your*

supplemental table 5 and Fig. 5f to help complete the comparison against recent literature (Adv Sci (Weinh) 2021, 8, 2004523; Nat. Commun. 2022, 13, 7899.).

Response: We introduce more information as suggested. Our designed Pd-Cl/Cu₂O catalyst maintains a current density of ~2200 mA cm⁻² for 3600 s, and achieves a NH₃ FE of 99.1% and NH₃ yield rate of ~19.4 mmol h⁻¹ cm⁻². Compared to Cu₅₀Co₅₀ nanosheets or electroreduction Co(OH)₂ nanoarrays (Table R1), our platform exhibits superiority in current density, reaction time, and NH₃ yield rate. We have added the details of these two references (Adv. Sci. 8, 2004523 (2021), #39; Nat. Commun. 13, 7899 (2022), #40.) to Figure 5f and Supplementary Table 6.

Table R1. Comparison of NO₃RR performance of Pd-Cl/Cu₂O with reported works.

Catalyst	Reaction time (s)	Reaction current (mA cm ⁻²)	NH ₃ FE (%)	NH ₃ yield rate (mmol h ⁻¹ cm ⁻²)	Ref.
Pd-Cl/Cu ₂ O	3600	~2200	99.1	~19.4	This work
Electroreduction Co(OH) ₂ nanoarrays	200	2200	98.0	10.4	Adv. Sci. 2021, 8, 2004523
Cu ₅₀ Co ₅₀ nanosheet	3600	1035	100 ± 1	4.8	Nat. Commun. 2022, 13, 7899

Corresponding revision:

➤ Figure 5f has been updated (Page 18, Line 322, Revised manuscript).

Figure 5. f. NO₃RR performance comparison of reported electrocatalysts.

➤ The details of these two references have been added to Supplementary Table 6 (Pages 74-75, Revised supplementary information).

Supplementary Table 6 Comparison of NO₃RR performance of Pd-Cl/Cu₂O with reported works.

Catalyst	Electrolyte	NH ₃ FE (%)	NH ₃ production rate (mg h ⁻¹ cm ⁻²)	NH ₃ partial current (mA cm ⁻²)	Ref.
Pd-CI/Cu ₂ O	56 mM KNO ₃ + 1 M KOH	99.2	30.1	~350	This work
Pd-CI/Cu ₂ O	1000 mM KNO ₃ + 1 M KOH	99.1	~330	~2180	This work
Ru ₁₅ Co ₈₅ HNDs	1000 mM KNO ₃ + 1 M KOH	97	~119	~1000	Nat. Cat. 2023, 6, 404-412
Ru-CuNW	32 mM KNO ₃ + 1 M KOH	96	76.6	965	Nat. Nanotechnol. 2022, 17, 759-767
Cu-PTCDA	8.1 mM KNO ₃ + 1 M PBS	85.9	0.44	-	Nat. Energy 2020, 5, 605-613
Electroreduction Co(OH) ₂ nanoarrays	100 mM KNO ₃ + 1 M KOH	98	176.8	~2156	Adv. Sci. 2021, 8, 2004523
Cu ₅₀ Co ₅₀ nanosheet	100 mM KNO ₃ + 1 M KOH	100	81.6	1035	Nat. Commun. 2022, 13, 7899
Fe single-atom catalyst (SAC)	500 mM KNO ₃ + 0.1 M K ₂ SO ₄	75	7.8	30	Nat. Commun. 2021, 12, 2870
Fe-PPy SAC	100 mM KNO ₃ + 0.1 M KOH	~100	2.8	34.6	Energy Environ. Sci. 2021, 14, 3522-3531
Pd	20 mM NaNO ₃ + 0.1 M NaOH	35	0.34	4.25	ACS Catal. 2021, 11, 12, 7568-7577
Ru nanoclusters	1000 mM KNO ₃ + 0.1 M KOH	~100	17.5	251	J. Am. Chem. Soc. 2020, 142, 7036-7046
Cu/Cu ₂ O NWAs	14.3 mM KNO ₃ + 0.5 M Na ₂ SO ₄	95.8	4.2	52.5	Angew. Chem. Int. Ed. 2020, 59, 5350-5354
Cu ₅₀ Ni ₅₀ alloy	100 mM KNO ₃ + 1 M KOH	99	7.1	90	J. Am. Chem. Soc. 2020, 142, 5702-5708
CuPd	1000 mM KNO ₃ + 1 M KOH	92.5	18.1	240	Nat. Commun. 2022, 13, 2338
Rh@Cu	100 mM KNO ₃ + 0.1 M Na ₂ SO ₄	93	13.6	162	Angew. Chem. Int. Ed. 2022, 61, e202202556
FTO-E	100 mM NaNO ₃ + 0.1 M PBS	87.6	1.2	-	Angew. Chem. Int. Ed. 2023, 62, e202215782
Cu@C	1 mM KNO ₃ + 1 M KOH	72	0.47	-	Adv. Mater. 2022, 34, 2204306
Cu/Cu ₂ O nanowires	3.2 mM NaNO ₃ + 0.5	81.2	4.1	-	Angew. Chem. Int.

	M Na ₂ SO ₄				Ed. 2020, 59, 5350-5354
a-RuO ₂	3.2 mM NaNO ₃ + 0.5 M Na ₂ SO ₄	97.46	2.0		Angew. Chem. Int. Ed. 2022, 134, e202202604
Poly-Cu ₁₄ cba	4 mM KNO ₃ + 0.5 M K ₂ SO ₄	90	2.8		Angew. Chem., Int. Ed. 2022, 134, e202114538
Ru ₁ Cu ₁₀ /rGO 1 wt.% loading	100 mM KNO ₃ + 1 M KOH	63	1.3	37	Adv. Mater. 2023, 35, 2202952
O-SiNW/Au	10 mM HNO ₃ + 0.5 M K ₂ SO ₄	95.6	4.4		Angew. Chem., Int. Ed. 2022, 61, e202204117
Fe/Ni ₂ P	50 mM KNO ₃ + 0.2 M K ₂ SO ₄	94.3	4.2		Adv. Energy Mater. 2022, 12, 2103872

Comment 4: *In the abstract, the statement that “nitrate reduction to ammonia suffers from low catalytic activities, especially in high pH conditions” is not rigorous. As listed in supplementary table 5, most catalytic systems with remarkable performance are under alkaline conditions.*

Response: We modified the statement to make it more accurate based on the suggestion.

Corresponding revision:

➤ The corresponding content has been revised to “Alkaline electrocatalytic NO₃⁻ reduction reaction (NO₃RR) emerges as an attractive route for enabling NO₃⁻ removal and sustainable ammonia (NH₃) synthesis. However, it suffers from insufficient proton (H⁺) supply in high pH conditions, restricting NO₃⁻-to-NH₃ activity. Herein, we propose a halogen-mediated H⁺ feeding strategy to enhance the alkaline NO₃RR performance.”

(Page 2, Lines 26-30, Revised manuscript).

Comment 5: *In the DFT calculation part, more evidence should support the view that nitrate reduction over the Pd-Cl/Cu₂O catalysts goes through multiple proton-coupled electron transfer processes instead of sequential proton–electron transfer processes (Nat. Catal. 2022, 5, 798-806; Chem. Sci. 4, 2710–2723 (2013). The specific results will significantly affect the accuracy of DFT calculation.*

Response: Based on this insightful comment, we introduced additional information of nitrate reduction pathways (Figure R2) for the clarification.

To determine the type of proton-electron transfer in the nitrate reduction process, we have supplemented the nitrate reduction pathways on Cu₂O, Pd/Cu₂O and Pd-Cl/Cu₂O models in our research system (pH 14). According to Nernst's equation ($E_{\text{RHE}} = E_{\text{SHE}} + 0.0591 \times \text{pH}$) and theoretical nitrate reduction potential ($E^0 = -0.12 \text{ V vs. RHE}$), the corresponding reaction potential at pH 14 is -0.95 V vs. SHE . Thus, we calculate the reaction pathway on the three catalysts at the potential of -0.95 V vs. SHE and compare the charge vibration of each intermediate on the three catalysts between the potentials of zero charge and -0.95 V vs. SHE .

Take Pd-Cl/Cu₂O as an example. Before adsorbing nitrate (Figure R2a), the charge number of Pd-Cl/Cu₂O at -0.95 V vs. SHE is more than that of the zero charge potential ($1.78 e$), indicating that Pd-Cl/Cu₂O has obtained electrons at the potential of -0.95 V vs. SHE . That is, the electrons are first transferred to Pd-Cl/Cu₂O. After adsorbing nitrate ($^*\text{NO}_3$), the charge number on Pd-Cl/Cu₂O at -0.95 V vs. SHE is still more than that of the zero charge potential ($1.76 e$), which demonstrates many electrons transfer to the reaction system under potential driving. As shown in Figure R2b, in the hydrogenation process of $^*\text{NO}_2$ to $^*\text{NO}_2\text{H}$ ($^*\text{NO}_2 + e^- + \text{H}^+ \rightarrow ^*\text{NO}_2\text{H}$), the charge number on Pd-Cl/Cu₂O with absorbed NO_2H at the potential of zero charge ($932 e$) is smaller than that of the system where the catalyst adsorbs NO_2 ($^*\text{NO}_2$, $932.8 e$) at the potential of -0.95 V vs. SHE (H^+ has not yet participated in the reaction). This result proves that electrons are transferred to Pd-Cl/Cu₂O under the action of potential before H^+ coupling. There are similar transfer phenomena in the following nitrate intermediate hydrogenation process on Pd-Cl/Cu₂O ($^*\text{NO} \rightarrow ^*\text{NOH}$, $^*\text{N} \rightarrow ^*\text{NH}$, $^*\text{NH} \rightarrow ^*\text{NH}_2$, $^*\text{NH}_2 \rightarrow ^*\text{NH}_3$). The phenomena in which electrons are transferred to the catalyst before H^+ coupling also occur on Cu₂O and Pd/Cu₂O. Thus, the three catalysts undergo a sequential electron-proton transfer process instead of a multiple proton-coupled electron transfer process in the nitrate reduction pathway (Figure R2).

We have revised the corresponding description and cited these references (Nat. Catal. 5, 798-806 (2022), #41; Chem. Sci. 4, 2710-2723 (2013), #42.) in the revised

manuscript.

Figure R2. **a**, The charge vibration of each intermediate on the catalysts between the potentials of zero charge and -0.95 V vs. SHE. **b**, The charge of each intermediate on Pd-Cl/Cu₂O at the potentials of zero charge and -0.95 V vs. SHE. **c**, The charge of each intermediate on Pd/Cu₂O at the potentials of zero charge and -0.95 V vs. SHE. **d**, The charge of each intermediate on Cu₂O at the potentials of zero charge and -0.95 V vs. SHE.

Corresponding revision:

- Figure R2 has been added to **Supplementary Figure 9** (Page 13, Revised supplementary information).
- The corresponding notes “To determine the type of proton-electron transfer in the nitrate reduction process, the nitrate reduction pathways on Cu₂O, Pd/Cu₂O and Pd-Cl/Cu₂O models in our research system (pH 14) have been calculated. According to Nernst's equation ($E_{RHE} = E_{SHE} + 0.0591 \times \text{pH}$) and theoretical nitrate reduction potential ($E^0 = -0.12 \text{ V vs. RHE}$), the corresponding reaction potential at pH 14 is -0.95 V vs. SHE. Thus, we calculate the reaction pathway on the three catalysts at the potential of -0.95 V vs. SHE and compare the charge vibration of each intermediate on

the three catalysts between the potentials of zero charge and -0.95 V vs. SHE.

Take Pd-Cl/Cu₂O as an example. Before adsorbing nitrate (Supplementary Fig. 9a), the charge number of Pd-Cl/Cu₂O at -0.95 V vs. SHE is more than that of the zero charge potential (1.78 e), indicating that Pd-Cl/Cu₂O has obtained electrons at the potential of -0.95 V vs. SHE. That is, the electrons are first transferred to Pd-Cl/Cu₂O. After adsorbing nitrate (*NO₃), the charge number on Pd-Cl/Cu₂O at -0.95 V vs. SHE is still more than that of the zero charge potential (1.76 e), which demonstrates many electrons transfer to the reaction system under potential driving. As shown in Supplementary Fig. 9b, in the hydrogenation process of *NO₂ to *NO₂H (*NO₂ + e⁻ + H⁺ → *NO₂H), the charge number on Pd-Cl/Cu₂O with absorbed NO₂H at the potential of zero charge (932 e) is smaller than that of the system where the catalyst adsorbs NO₂ (*NO₂, 932.8 e) at the potential of -0.95 V vs. SHE (H⁺ has not yet participated in the reaction). This result proves that electrons are transferred to Pd-Cl/Cu₂O under the action of potential before H⁺ coupling. There are similar transfer phenomena in the following nitrate intermediate hydrogenation process on Pd-Cl/Cu₂O (*NO → *NOH, *N → *NH, *NH → *NH₂, *NH₂ → *NH₃). The phenomena in which electrons are transferred to the catalyst before H⁺ coupling also occur on Cu₂O and Pd/Cu₂O. Thus, the three catalysts undergo a sequential electron-proton transfer process in the nitrate reduction pathway (Supplementary Fig. 9).” has been added to Supplementary Figure 9 (Pages 13-14, Revised supplementary information).

➤ The corresponding content has been revised to “In such a sequential electron-proton transfer process (Supplementary Fig. 9)^{41,42}, the hydrogenation of *NO₂ into *NO₂H (*NO₂ + e⁻ + H⁺ → *NO₂H) was the potential-determining step (PDS), which involved a high ΔG of 0.68 eV over pure Cu₂O (Supplementary Fig. 4). Pd/Cu₂O also presented a relatively high ΔG of PDS (0.53 eV). Pd-Cl/Cu₂O showed the lowest ΔG of PDS (0.40 eV) and correspondingly advanced the progress of NO₃RR.” (Page 8, Lines 133-139, Revised manuscript).

Comment 6: *As the author stated, the DFT calculation results inspired the experimental parts. Why did the author choose Cu₂O and Pd-modified Cu₂O as catalysts? Why did*

the author select Cl as the coordination element instead of F, Br, I, O, or S? The logic is confusing.

Response: We introduce more explanations about catalyst selection to clarify the potential confusion.

The reasons for selecting Cu₂O and Pd-modified Cu₂O as catalysts and regulating Pd/Cu₂O with Cl coordination element are as follows:

1) Choosing Cu₂O as catalysts. The electrochemical nitrate (NO₃⁻) reduction reaction occurs at the cathode, however, NO₃⁻ as anions are difficult to adsorb on the cathode surface with abundant electrons due to the repulsion of isotropic charges. DFT calculations and experiments in reported literatures indicate that Cu₂O can effectively facilitate the adsorption and activation of NO₃⁻ (Environ. Sci. Technol. 56, 10299-10307 (2022); Adv. Funct. Mater. 2303803 (2023)), which could be attributed to the Cu atoms with unpaired *d*-electrons and similar *d*-band energy to the nitrate's LUMO π^* (Sov. Electrochem. 7, 312 (1971); Small Methods 4, 2000672 (2020)). Thus, we choose Cu₂O as the research platform to promote the adsorption and conversion of NO₃⁻, which was a prerequisite for achieving high-performance NO₃RR.

2) Choosing Pd-modified Cu₂O as catalysts. Although Cu₂O has the excellent ability to adsorb and activate NO₃⁻, it is subjected to sluggish hydrogenation of NO_x intermediates owing to the insufficient proton (H⁺) supply in alkaline NO₃RR, retarding the effective NO₃⁻-to-NH₃ conversion. Introducing active components that accelerate H₂O dissociation to provide H⁺ could be a valid strategy. As we know, Pd has displayed a favorable behavior to activate H₂O under high pH conditions (Energy Environ. Sci. 12, 2620-2645 (2019); Nat. Commun. 7, 13549 (2016); ACS Catal. 12, 4840-4847 (2022)). Hence, we modify Pd species into Cu₂O to promote H₂O dissociation in the expectation of supplying H⁺ in alkaline systems.

3) Regulating Pd/Cu₂O with Cl coordination. Considering that the intense interaction between the *d* orbitals of Pd metals and *s* orbitals of *H (* denotes the adsorbed state) brings about the strong Pd-H binding (Nano Energy 34, 306-312 (2017)), which would affect the desorption of *H and thus hinder the nitrate intermediates hydrogenation. Regulating the Pd 3*d* electronic orbital structure is a direct way to

weaken *H binding on Pd sites. Compared with bulk materials, single-atom catalysts (SACs) are electronic structures adjustable through the appropriate ligands modification (*Adv. Mater.* 31, e1900509 (2019); *J. Am. Chem. Soc.* 144, 18155-18174 (2022)). Among the nonmetal element ligands (F, Cl, Br, I, O, or S), Cl with the strongest first electron affinity, possesses a unique electron-withdrawing effect (*Nat. Commun.* 13, 6875 (2022)). The modification of Cl would seize the electrons of Pd SACs to adjust its *d*-band center, which is beneficial for the *H desorption (*Angew. Chem. Int. Ed.* 57, 5076-5080 (2018)). The release of H⁺ could participate in the nitrate intermediate hydrogenation for targeted NH₃ synthesis. Therefore, we select Cl as the coordination element to regulate the electronic structure of Pd atoms on Cu₂O.

Corresponding revision:

➤ The content “In this study, we design and realize a halogen-mediated alkaline electrocatalytic platform to overcome the limitation and achieve high-speed conversion of NO₃⁻ to NH₃ while maintaining an ideal FE. Modifying Pd species on Cu₂O platform with excellent NO₃⁻ adsorption and conversion ability^{32,33} could favor H₂O dissociation under high pH conditions³⁴⁻³⁶. But the intense interaction between the *d* orbitals of Pd and *s* orbitals of *H (* denotes the adsorbed state) brings strong Pd-H binding³⁷, which affects the desorption of *H. Halogen elements with high first-electron-affinity³⁸ can tailor the 3*d* orbital electron structure of Pd atom to regulate *H release, thereby breaking the bottleneck of FE due to the scarce proton (H⁺) feeding in high pH conditions. Here we develop Cl-coordinated Pd SACs-dispersed Cu₂O matrix (Pd-Cl/Cu₂O) nanocrystal to carry out alkaline NO₃RR.” has been added to the introduction part of the revised manuscript (Page 4, Lines 80-86, Page 5, Lines 87-90, Revised manuscript).

Comment 7: In the in situ ATR-IR part (Figs. 4f, g, and S22), the assignment of N-related intermediates should be reconsidered. If the IR spectra of the standard sample (for example, nitrate, nitrite, NO, and ammonia) can be supplemented, the assignment would be more persuasive.

Response: Based on this inspiring suggestion, we have supplemented the IR spectra of

the KNO_3 , KNO_2 , NH_4Cl , CH_3CONH_2 and NO standard samples. According to the results in Figure R3, the characteristic peaks in the *in situ* ATR-IR spectra (Figures 4f-g and Supplementary Figure 31) have been reassigned, and we have made corresponding modifications to the descriptions in the manuscript.

Specifically, the absorption bands at 1362 and 1216 cm^{-1} are assigned to $\nu(\text{NO}_3^-)$ and $\nu(\text{NO}_2^-)$, respectively, according to the results of standard KNO_3 and KNO_2 powders. Based on the spectra of NH_4Cl and $\text{C}_2\text{H}_5\text{NO}$, we have identified the characteristic vibration of N-H bond, ranging from 2900 - 3500 cm^{-1} (red dashed region) and 1380 - 1760 cm^{-1} (red dashed lines). The peaks at 1674 cm^{-1} in the spectra of $\text{C}_2\text{H}_5\text{NO}$ can be indexed to $\nu(\text{C}=\text{O})$. In addition, standard NO gas exhibits the characteristic bands at 1260 , 1400 , 1540 , and 1640 cm^{-1} , in accordance with the previously reported works (J. Catal. 237, 393–404, (2006); Langes Handbook of Chemistry.). Besides, the obvious negative peaks at 1114 cm^{-1} in the *in situ* ATR-IR spectra (Figures 4f-g and Supplementary Figure 31) can be ascribed to the Si-O signal (Adv. Mater. Lett. 7, 480-484 (2016); Mater. Sci. Eng. B 105, 209-213 (2003).), which is derived from the reduction of surface SiO_2 on the Si semi-cylindrical prism substrate under the applied potentials.

Figure R3. The IR spectra of the KNO_3 , KNO_2 , NH_4Cl , $\text{C}_2\text{H}_5\text{NO}$ and NO standard samples.

Figure 4. f,g, *In situ* ATR-IR spectra of Pd-Cl/Cu₂O (f) and Pd/Cu₂O catalysts (g).

Supplementary Figure 31. *In situ* ATR-IR spectra of Cu₂O catalysts.

Corresponding revision:

- The corresponding content has been revised to “Under the driven of applied potential from 0.2 to -0.7 V *versus* RHE, the detected N-O peaks (at 1540 cm⁻¹)⁵³ in the spectra of Pd-Cl/Cu₂O demonstrated the deoxygenation of NO₃⁻ to the intermediate *NO. The conspicuous peaks of hydro-nitrogen intermediates (-NH) at 3200-3380 cm⁻¹ indicated the effective hydrogenation of *NO intermediates on Pd-Cl/Cu₂O⁵⁴⁻⁵⁷. In comparison, Pd/Cu₂O displayed the stronger *NO intermediate peaks and the weaker -NH signals (Figs. 4f-g). These results evidenced that Cl-mediated H⁺ feeding could boost *NO intermediates hydrogenation in alkaline NO₃RR for promising NH₃ synthesis over Pd-Cl/Cu₂O.” (Page 12, Lines 225-228, Page 13, Lines 229-232, Revised manuscript).
- The content “Si-O signal was derived from the reduction of surface SiO₂ on the Si semi-cylindrical prism substrate under the applied potentials.” has been added (Page 13,

Lines 237-239, Revised manuscript).

➤ Figures 4f-g and **Supplementary Figure 31** have been revised (Page 13, Line 233, Revised manuscript; Page 36, Revised supplementary information).

➤ Figure R3 has been added to **Supplementary Figure 30** (Page 35, Revised supplementary information).

➤ The corresponding notes “We tested the IR spectra of the KNO_3 , KNO_2 , NH_4Cl , CH_3CONH_2 and NO standard samples. According to the results in Supplementary Fig. 30, the characteristic peaks in the *in situ* ATR-IR spectra (Figs. 4f-g and Supplementary Fig. 31) have been assigned.

Specifically, the absorption bands at 1362 and 1216 cm^{-1} are assigned to $\nu(\text{NO}_3^-)$ and $\nu(\text{NO}_2^-)$, respectively, according to the results of standard KNO_3 and KNO_2 powders. Based on the spectra of NH_4Cl and $\text{C}_2\text{H}_5\text{NO}$, we have identified the characteristic vibration of N-H bond, ranging from 2900 - 3500 cm^{-1} (red dashed region) and 1380 - 1760 cm^{-1} (red dotted lines). The peaks at 1674 cm^{-1} in the spectra of $\text{C}_2\text{H}_5\text{NO}$ can be indexed to $\nu(\text{C}=\text{O})$. In addition, standard NO gas exhibits the characteristic bands at 1260 , 1400 , 1540 , and 1640 cm^{-1} , in accordance with the previously reported works (J. Catal. 237, 393–404, (2006); Langes Handbook of Chemistry.). Besides, the obvious negative peaks at 1114 cm^{-1} in the *in situ* ATR-IR spectra (Figs. 4f-g and Supplementary Fig. 31) can be ascribed to the Si-O signal (Adv. Mater. Lett. 7, 480-484 (2016); Mater. Sci. Eng. B 105, 209-213 (2003).), which is derived from the reduction of surface SiO_2 on the Si semi-cylindrical prism substrate under the applied potentials.” has been added to **Supplementary Figure 30** (Page 35, Revised supplementary information).

Comment 8: *To state the conclusion that “Cl-mediated H^+ feeding could boost *NO intermediates hydrogenation in alkaline NO_3RR for promising NH_3 synthesis over Pd-Cl/ Cu_2O .”, the *in situ* ATR-IR spectra of Pd-Cl/ Cu_2O (Fig. 4f) should compare with Pd/ Cu_2O (Fig. S22), instead of Cu_2O (Fig. 4g).*

Response: We have implemented a new set of experiments to compare the *in situ* ATR-IR spectra of Pd-Cl/ Cu_2O and Pd/ Cu_2O . Correspondingly, we have made modifications to the descriptions and figures in the manuscript.

Corresponding revision:

➤ The corresponding content has been revised to “Under the driven of applied potential from 0.2 to -0.7 V *versus* RHE, the detected N-O peaks (at 1540 cm⁻¹)⁵³ in the spectra of Pd-Cl/Cu₂O demonstrated the deoxygenation of NO₃⁻ to the intermediate *NO. The conspicuous peaks of hydro-nitrogen intermediates (-NH) at 3200-3380 cm⁻¹ indicated the effective hydrogenation of *NO intermediates on Pd-Cl/Cu₂O⁵⁴⁻⁵⁷. In comparison, Pd/Cu₂O displayed the stronger *NO intermediate peaks and the weaker -NH signals (Figs. 4f-g). These results evidenced that Cl-mediated H⁺ feeding could boost *NO intermediates hydrogenation in alkaline NO₃RR for promising NH₃ synthesis over Pd-Cl/Cu₂O.” (Page 12, Lines 225-228, Page 13, Lines 229-232, Revised manuscript).

Figure 4. f,g. *In situ* ATR-IR spectra of Pd-Cl/Cu₂O (f) and Pd/Cu₂O catalysts (g).

Comment 9: *In Fig 5a and other performance-related figures, the total Faradaic efficiency is far from ~100%. How about the other products, such as N₂, N₂O, NH₂OH, H₂, etc? It's recommended to provide a more complete product distribution map.*

Response: Following this kind suggestion, we have supplemented online differential electrochemical mass spectrometry (DEMS) experiments of the catalysts to detect the intermediates and gas products generated during the NO₃RR (Figure R4), such as H₂, N₂, NH₃, NO, NH₂OH, N₂O.

Under the potential of -0.4 V *versus* RHE, Cu₂O and Pd/Cu₂O catalysts produce other gas products in addition to NH₃ (17), including *m/z* signals of H₂ (2), N₂ (28), NO (30), and N₂O (44). Among the many byproducts, a large amount of NO generated from Cu₂O reveals that *NO intermediates are difficult to hydrogenate and subsequently

desorb due to insufficient proton supply in alkaline NO₃RR. In Pd/Cu₂O, the production of NH₃ is improved, but the amount of H₂ is also significantly increased. The phenomenon indicates that although the introduction of Pd accelerates the water dissociation, the serious HER process limits the conversion of *NO intermediate to NH₃. Thanks to the halogen-mediated proton feeding strategy, the optimal catalyst Pd-Cl/Cu₂O mainly generates NH₃ with negligible other gas products during the potentiostatic process at -0.4 V *versus* RHE. We have added these data to the manuscript.

Figure R4. Online differential electrochemical mass spectrometry (DEMS) measurements of NO₃RR over **a**, Cu₂O, **b**, Pd/Cu₂O, and **c**, Pd-Cl/Cu₂O under the potential of -0.4 V *versus* RHE.

Corresponding revision:

➤ The corresponding content has been revised to “The three catalysts, Pd-Cl/Cu₂O, Pd/Cu₂O and Cu₂O, all showed a high NO₃⁻ conversion rate of ~99%, indicating that Cu₂O matrix has a strong NO₃⁻ removal ability. Yet, the NO₃⁻ to NH₃ conversion rates of Cu₂O and Pd/Cu₂O within 1 h electrolysis were as low as ~20% and ~45%, respectively, accompanied by producing 26% and 21% of NO₂⁻ (Supplementary Fig. 44). Electrochemical online differential electrochemical mass spectrometry (DEMS) showed that Cu₂O and Pd/Cu₂O also generated the gas products during the potentiostatic process at -0.4 V *versus* RHE (Supplementary Figs. 45a-b), including the *m/z* signals of NH₃ (17), H₂ (2), N₂ (28), NO (30), NH₂OH (33), and N₂O (44). Notably, nearly all the NO₃⁻ was converted into NH₃ on Pd-Cl/Cu₂O within 1 h electrolysis, and the corresponding selectivity reached ~99.1% with negligible NO₂⁻ and gas products (Fig. 5c and Supplementary Fig. 45c). The residual NO₃⁻ and NO₂⁻ concentrations were both significantly lower than the World Health Organization (WHO) regulations for drinking water (Supplementary Fig. 46)^{64,65}. These results corroborated the excellent

NO_3^- removal rate and high NH_3 selectivity of Pd-Cl/ Cu_2O .” (Page 15, Lines 272-282, Page 16, Lines 283-286, Revised manuscript).

➤ Figure R4 has been added to **Supplementary Figure 45** (Page 50, Revised supplementary information).

➤ The content “**Electrochemical online DEMS tests.** The online DEMS tests were performed in customized reactors containing 1 M KOH with 56 mM NO_3^- electrolyte. Ar was continuously bubbled into the electrolyte. Catalysts coated on breathable film with gold plating layer, Pt wire, and saturated Ag/AgCl electrode were used as the working, the counter and the reference electrode, respectively. After the baseline of the mass spectrometry kept steady, the potential of OCP and -0.4 V *versus* RHE were applied alternately with an interval of 3 min. Accordingly, the differential mass signals appeared when the gaseous products formed on the electrode surface. The mass signal returned to baseline after the electrochemical measurement was over. To avoid accidental errors, the next cycle started using the same conditions. After five cycles, the experiment was ended.” has been added to the Method part (Page 27, Lines 512-522, Revised manuscript).

Reviewer #2: In this work, Liu et al. proposed a universal halogen-mediated H^+ feeding strategy to enhance alkaline NO_3RR to ammonia, achieving near-100% NH_3 Faradaic efficiency (at $pH = 14$) with a record current density of $2 A cm^{-2}$ and enabling an over 99% NO_3^- -to NH_3 conversion efficiency. The authors performed in situ spectroscopy experiments and DFT calculations to reveal the role of halogen ligands, which constructed local H^+ -abundant environments for $*NO$ intermediate hydrogenation and finally effective NO_3^- -to NH_3 conversion. This is an interesting study that will appeal to a broad audience, as it incorporates elements of data science, computational chemistry, and experiments all to study an important application. Therefore, I recommend this work for publication after pending minor revisions. Some specified comments are listed as follows:

Response: We are grateful to the high evaluation of our work, in particular for board impact to chemistry and beyond. Also, we follow the insightful comments/suggestions from the reviewer, which further improve the quality of the manuscript.

Comment 1: Electrochemical nitrate reduction has potential application prospects for wastewater treatment, and both N_2 and NH_3 as the targeted products have been reported in many published literatures (*Environ. Sci. Technol.* 56, 614-623 (2022); *Nat. Nanotechnol.* 17, 759-767 (2022)). Then what are the significance and advantages of NH_3 as the main product compared to N_2 ?

Response: Based on this inspiring question, we offer more explanations as the following and add them into the manuscript.

Electrochemical nitrate reduction with N_2 as the targeted product is considered to be a promising method in the wastewater treatment field because of its low cost, environmental friendliness, and easy operation. The conversion of nitrate to N_2 is important and meaningful in a viewpoint for the removal of nitrate pollutants. However, N_2 is a low-value product compared with high value-added NH_3 . NH_3 can be used as a nitrogen source in agricultural fertilizers, a chemical feedstock for many industrial processes, and a high-energy-density carrier for renewable hydrogen. Thus, the

conversion of nitrate into NH₃ not only realizes the removal of pollutants, but also achieves the fabrication of high value-added products, which may create certain economic benefits.

Corresponding revision:

➤ The corresponding content “Nitrate reduction to NH₃, instead of N₂, significantly enhances the product value of the electrochemical process. NH₃ can serve as a valuable nitrogen source in agricultural fertilizers, a chemical feedstock for various industrial processes, and a high-energy-density carrier for renewable hydrogen.” has been added (Page 19, Lines 354-358, Revised manuscript).

*Comment 2: In Fig. 2d, the free energy change for the desorption of *NH₃ (*NH₃→NH₃) is larger than that of the hydrogenation of *NO into *NOH (*NO + H⁺ → *NOH) in pure Cu₂O and Pd/Cu₂O. The authors should check the expression of the potential-determining step (PDS) whether it is suitable for the Cu₂O and Pd/Cu₂O catalysts.*

Response: We thank the reviewer for the meticulous examination and implemented the corresponding correction.

1) The desorption barrier of ammonia is relatively large. We think that the theoretical simulation involves the calculation about the desorption of ammonia gas, while the ammonia produced by experiment exists in the form of ammonium ions in the electrolyte. There is a difference. To better reflect the aqueous NO₃RR process, we did not consider the energy barrier of ammonia gas desorption.

2) We also agree with the reviewer’s opinion that the expression of the PDS is inappropriate. To avoid the ambiguity, we have recalculated the nitrate reduction pathways on Cu₂O, Pd/Cu₂O and Pd-Cl/Cu₂O models at the potential of -0.95 V vs. SHE without considering the desorption barrier of ammonia gas. As shown in Figure R5, the potential-determining step (PDS) on the three models is the hydrogenation of *NO₂ into *NO₂H (*NO₂ + H⁺ + e⁻ → *NO₂H) and the lower energy is required on Pd-Cl/Cu₂O (0.40 eV) compared to Cu₂O (0.68 eV) and Pd/Cu₂O (0.53 eV). This result proves an easier process of nitrate reduction to ammonia occurred on Pd-Cl/Cu₂O.

Figure R5. Gibbs free energy diagram of various intermediates generated during NO₃RR over catalysts at the potential of -0.95 V vs. SHE.

Corresponding revision:

➤ Figure R5 has been added to **Figure 2d** and **Supplementary Figure 4** (Page 8, Line 141, Revised manuscript; Page 8, Revised supplementary information).

Figure 2. d, Gibbs free energy diagram of various intermediates generated during NO₃RR over Pd/Cu₂O and Pd-Cl/Cu₂O at the potential of -0.95 V versus SHE. Inset: NO₃RR pathway of Pd-Cl/Cu₂O.

Supplementary Figure 4. Gibbs free energy diagram of various intermediates generated during NO₃RR over Cu₂O at the potential of -0.95 V vs. SHE.

➤ The corresponding content has been revised to “Next, after considering the effects of potential, pH on NO₃RR pathway with multiple possible branches (Supplementary Figs. 4-8), the optimal pathway on Cu₂O, Pd/Cu₂O and Pd-Cl/Cu₂O models at the potential of -0.95 V *versus* SHE was proposed and the corresponding ΔG of each intermediate was calculated (Fig. 2d). In such a sequential electron–proton transfer process (Supplementary Fig. 9)^{41,42}, the hydrogenation of *NO₂ into *NO₂H (*NO₂ + e⁻ + H⁺ → *NO₂H) was the potential-determining step (PDS), which involved a high ΔG of 0.68 eV over pure Cu₂O (Supplementary Fig. 4). Pd/Cu₂O also presented a relatively high ΔG of PDS (0.53 eV). Pd-Cl/Cu₂O showed the lowest ΔG of PDS (0.40 eV) and correspondingly advanced the progress of NO₃RR.” (Page 8, Lines 130-139, Revised manuscript).

➤ The corresponding content “According to Nernst's equation ($E_{\text{RHE}} = E_{\text{SHE}} + 0.0591 \times \text{pH}$) and theoretical nitrate reduction potential ($E^0 = -0.12 \text{ V vs. RHE}$), the corresponding reaction potentials at pH 0 and 14 are -0.12 and -0.95 V vs. SHE, respectively.” has been added to the DFT computational details (Page 29, Lines 556-559, Revised manuscript).

Comment 3: *In the manuscript, the authors select the Pd-Cl/Cu₂O as the catalyst for highly efficient NO₃RR to ammonia. The effect of Pd and Cl components has been well-elaborated in which Pd accelerates the H₂O dissociation, and the Cl ligand facilitates the desorption of *H for H⁺ feeding. Besides, what is the role of Cu₂O substrate?*

Response: Here, we clarify the function of Cu₂O and emphasize it in the manuscript. Cu₂O in Pd-Cl/Cu₂O has two roles in alkaline NO₃RR system, as follows:

1) Cu₂O can effectively facilitate the adsorption, activation and conversion of nitrate as proved by DFT calculations (Figure 2d) and electrochemical nitrate removal experiments (Figure 5c and Supplementary Figure 44), which is a prerequisite for achieving high-performance NO₃RR.

Figure 2. d, Gibbs free energy diagram of various intermediates generated during NO_3RR over $\text{Pd}/\text{Cu}_2\text{O}$ and $\text{Pd-Cl}/\text{Cu}_2\text{O}$ at the potential of -0.95 V versus SHE. Inset: NO_3RR pathway of $\text{Pd-Cl}/\text{Cu}_2\text{O}$.

Figure 5. c, NO_3^- removal of catalysts measured in a 1 M KOH with 56 mM NO_3^- electrolyte (equals $790.3\text{ }\mu\text{g mL}^{-1}\text{ NO}_3^-\text{-N}$) at -0.4 V versus RHE. After 1 h electrolysis, only $7.1\text{ }\mu\text{g mL}^{-1}$ of $\text{NO}_3^-\text{-N}$ and $0.85\text{ }\mu\text{g mL}^{-1}$ of $\text{NO}_2^-\text{-N}$ remained, both below the WHO regulations for drinking water ($\text{NO}_3^-\text{-N} < 11.3\text{ }\mu\text{g mL}^{-1}$ and $\text{NO}_2^-\text{-N} < 0.91\text{ }\mu\text{g mL}^{-1}$).

Supplementary Figure 44. NO_3^- removal in a 1 M KOH with 56 mM NO_3^- electrolyte (equals $790.3\text{ }\mu\text{g mL}^{-1}\text{ NO}_3^-\text{-N}$) at -0.4 V vs. RHE.

2) Cu₂O could serve as a platform for loading Cl-coordinated Pd single atoms and provide Cu active sites to boost the NO₃RR process. Under the mediation of Cl, Pd single atoms on Cu₂O promote dangling O-H water dissociation and fast *H desorption (Figure 2a). The constructed local H⁺-abundant environments favor the free H⁺ feeding to nitrate intermediate hydrogenation, and thus realizing efficient NO₃⁻-to-NH₃ conversion on Cu sites (Figure 1 and Figure 4).

Figure 2. a, Gibbs free energy change of H₂O dissociation and H⁺ supply over catalysts.

Figure 1. Schematic diagram of Cl mediated H⁺ feeding to boost *NO intermediate hydrogenation and finally achieve efficient NO₃⁻-to-NH₃ conversion in alkaline NO₃RR over Pd-Cl/Cu₂O.

Figure 4. *NO intermediates hydrogenation. **a**, *In situ* Raman spectra of Pd-Cl/Cu₂O. **b**, Schematic diagram of the local H⁺-abundant environment construction over Pd-Cl/Cu₂O. **c-e**, *In situ* Raman spectra of Pd-Cl/Cu₂O (**c**), corresponding peak area (**d**) and Raman shift (**e**) of various interfacial H₂O structures. **f,g**, *In situ* ATR-IR spectra of Pd-Cl/Cu₂O (**f**) and Pd/Cu₂O (**g**) catalysts. Si-O signal was derived from the reduction of surface SiO₂ on the Si semi-cylindrical prism substrate under the applied potentials.

Corresponding revision:

➤ The corresponding content “The Cu₂O substrate plays a dual role in the catalysis. On one hand, it effectively facilitates the adsorption, activation and conversion of nitrate. On the other hand, it serves as a platform for loading Cl-coordinated Pd single atoms and provide Cu active sites to boost the NO₃RR process.” has been added (Page 21, Lines 399-403, Revised manuscript).

Comment 4: In the discussion of Fig. 4d-e, the author claimed “At more negative potentials, dangling O-H water exhibited a smoother area change and a steeper shift slope (-23.0 cm⁻¹ V⁻¹) compared to the other two interfacial H₂O structures, certifying the preferential dissociation of dangling O-H water on Pd-Cl/Cu₂O”. The authors should provide a more detailed discussion to help the readers to understand this conclusion.

Response: We have provided a detailed description with the statement as suggested. We have provided a detailed description with the statement. The three types of interfacial water can vary as a function of electrode potential due to the Stark effect. At more negative potentials, the changes in peak intensity and shift of these interfacial water would accordingly become obvious. Compared to the tetra-H₂O and tri-H₂O, dangling O-H water exhibited a smoother area change and a steeper shift slope (-23.0 cm⁻¹ V⁻¹) as the potential decreases, certifying the preferential consumption of dangling O-H water on Pd-Cl/Cu₂O.

Corresponding revision:

➤ The corresponding content has been revised to “The three types of interfacial water can vary as a function of electrode potential due to the Stark effect. At more negative potentials, the changes in peak intensity and shift of these interfacial water would accordingly become obvious. Compared to the tetra-H₂O and tri-H₂O, dangling O-H water exhibited a smoother area change and a steeper shift slope (-23.0 cm⁻¹ V⁻¹) as the potential decreases (Figs. 4c-e), certifying the preferential dissociation of dangling O-H water on Pd-Cl/Cu₂O.” (Page 12, Lines 209-215, Revised manuscript).

Comment 5: The authors have provided abundant characterizations of the catalysts before and after NO₃RR to ammonia, to verify the structural stability. Although the NO₃RR performance stability over Pd-Cl/Cu₂O has been demonstrated for successive 10 cycles, more experimental data are expected to support the durability of the catalysts, especially the i-t curves for each cycle.

Response: We have supplemented the i-t curves of NO₃RR over Pd-Cl/Cu₂O for successive 10 cycles (Figure R6). During the successive 10 cycles of NO₃RR tests, Pd-

Cl/Cu₂O not only behaved the favorable stability of NH₃ yield rate and FE, but also exhibited excellent current steadiness.

Figure R6. Time-dependent current density curves of NO₃RR over Pd-Cl/Cu₂O for successive 10 cycles.

Corresponding revision:

➤ Figure R6 has been added to **Supplementary Figure 47b** (Page 52, Revised supplementary information).

Supplementary Figure 47 | (a) NO₃RR performance stability over Pd-Cl/Cu₂O measured in a 1 M KOH with 56 mM NO₃⁻ electrolyte at -0.4 V vs. RHE. (b) Time-dependent current density curves of NO₃RR over Pd-Cl/Cu₂O for successive 10 cycles.

➤ The corresponding content “**In addition, Pd-Cl/Cu₂O behaved with favorable NO₃RR stability during ten-cycling tests (Supplementary Fig. 47).**” has been added (Page 16, Lines 286-288, Revised manuscript).

Comment 6: *There are some minor comments and suggestions. 1) The authors should*

use the formula editor to unify the format of the formulas in the Methods part. 2) The ordinate of Fig. 5d should be modified to “NH₃ partial current density”. 3) The color identification of N element needs to be added in Fig. 2d.

Response: We thank the reviewer for the meticulous reading and have revised the corresponding part of the manuscript.

Corresponding revision:

- The formulas in the Methods part have been used by the formula editor to unify the format (Page 23, Line 436, Page 24, Lines 454, 462 and 464-465, Page 25, Line 479, Page 28, Lines 532-533 and 538, Page 29, Line 555, Revised manuscript).
- The ordinate of Figure 5d has been revised to “NH₃ partial current density” (Page 18, Line 322, Revised manuscript).

Figure 5. d, NH₃ partial current densities of Pd-Cl/Cu₂O in a 1 M KOH electrolyte with 1000 mM NO₃⁻ under the potential range from -0.2 to -0.6 V *versus* RHE.

- The color identification of N element has been added to **Figure 2**. (Page 8, Line 141, Revised manuscript).

Comment 7: For the integrity of the research background, more recent works about NH₃ electrosynthesis are suggested to be referred: J. Colloid Interface Sci. 2023, 638, 650; Front. Chem. Sci. Eng. 2023, 17, 726; Nano Res. Energy 2022, 1, e9120022; Chem. Commun. 2022, 58, 8097; Inorg. Chem. Commun. 2023, 151, 110621.

Response: We have added these references to our manuscript for the integrity of the research background.

Corresponding revision:

➤ These related references (#10-12, #15, #21) have been cited. (Page 3, Lines 58-60 and 63-65, Revised manuscript).

Comment 8: *To further confirm the electrochemical stability of the catalyst, the EIS spectra before and after the long-term electrolysis are suggested to be added.*

Response: We have supplemented the EIS curves as suggested (Figure R7). The Nyquist plots and corresponding equivalent circuit diagram fitting results of Pd-Cl/Cu₂O (Table R2) show no significant changes before and after the NO₃RR tests, demonstrating the good electrochemical stability of the catalyst.

Figure R7. Nyquist plots of Pd-Cl/Cu₂O before and after NO₃RR tests.

Table R2. Fitting results for resistances of Pd-Cl/Cu₂O before and after NO₃RR tests according to EIS equivalent circuit diagram.

Resistance Sample tests	R _s (Ω cm ⁻²)	R _{bulk} (Ω cm ⁻²)	R _{ct} (Ω cm ⁻²)
Before tests	1.45	0.156	44.3
After tests	1.46	0.175	48.7

Note: R_s is the external circuit resistance. R_{bulk} is the bulk trapping resistance. R_{ct} is the interfacial charge transfer resistance.

Corresponding revision:

➤ The corresponding content has been revised to “The morphology, phase structure,

chemical valence state, atomic coordination environment and electrochemical properties remained steady after the electrolysis, further indicating the robust structure of catalysts (Supplementary Figs. 48-51 and Tables 3-4).” (Page 16, Lines 288-291, Revised manuscript).

➤ Figure R7 has been added to Supplementary Figure 49b (Page 54, Revised supplementary information).

➤ Table R2 has been added to Supplementary Table 4 (Page 72, Revised supplementary information).

Reviewer #3: The authors report on the high performance characteristics of Pd on Cu₂O with bound Cl ions to catalyze the nitrate reduction reaction to NH₃. This reaction is very important in terms of both environmental remediation and sustainable synthesis of ammonia, a crucial commodity chemical. While the experimental results and efficiencies are impressive, there are several major concerns that are present, especially among the computational work which must be addressed. Without these additions, the work is not fully substantiated nor is the methodology fully sound.

Below I first outline key computational chemistry concerns, then add some additional questions and comments.

The computational work has some key holes which need to be addressed before the findings are publishable.

Response: We thank the reviewer for finding our work impressive and providing several constructive suggestions, especially in the computational aspects. Following these insightful comments, we have implemented a series of calculations, clarifications, and modifications to enhance the quality of the paper.

Comment 1: The choice of K-point sampling and cut-off energy are not justified. There is no indication how the results compare to more rigorous computations (i.e. a 4 X 4X 1 finer k-point mesh or say 500 eV cut-off energy). Without these numbers it is impossible to discern the quality of the results.

Response: We have followed this constructive suggestion to provide more related information through additional calculations.

We have supplemented the calculations of various cut-off energies (from 350 to 500 eV) and different K points (from $1 \times 1 \times 1$ to $5 \times 5 \times 1$) on Cu₂O model before and after H₂O adsorption to investigate their impact on the energy of the entire reaction system. As shown in Figures R8a-b, under a series of cut-off energies, the energy difference between the Cu₂O model before and after H₂O adsorption is relatively small. Similarly, the energy change on the models under different K-points is also not significant (Figures R8c-d). Therefore, we chose the $2 \times 2 \times 1$ K-point mesh and 450

eV cut-off energy for calculation.

Figure R8. **a,b** The energy (**a**) and energy difference (**b**) on Cu₂O model before and after H₂O adsorption under various cut-off energies. **c,d** The energy (**c**) and energy difference (**d**) on Cu₂O model before and after H₂O adsorption under different K-points.

Corresponding revision:

➤ Figure R8 has been added to **Supplementary Figure 63** (Page 68, Revised supplementary information).

➤ The corresponding notes “**We have calculated the various cut-off energies (from 350 to 500 eV) and different K points (from $1 \times 1 \times 1$ to $5 \times 5 \times 1$) on Cu₂O model before and after H₂O adsorption to investigate their impact on the energy of the entire reaction system. As shown in Supplementary Figs. 63a-b, under a series of cut-off energies, the energy difference between the Cu₂O model before and after H₂O adsorption is relatively small. Similarly, the energy change on the models under different K-points is also not significant (Supplementary Figs. 63c-d). Therefore, we chose the $2 \times 2 \times 1$ K-point mesh and 450 eV cut-off energy for calculation.**” has been added to **Supplementary Figure 63** (Page 68, Revised supplementary information).

***Comment 2:** The authors state that the computational hydrogen electrode was used. However, they never state what potential was used in the CHE model. This is crucial as the behavior of electrochemical systems are highly dependent on potential.*

Response: We agree that the potential used in the model is crucial. We have introduced the missing information and added more analysis.

1) In Computational Hydrogen Electrode (CHE) model, $\text{H}^+ + \text{e}^- \rightleftharpoons 1/2 \text{H}_2(\text{g})$ was equilibrated at 0 V versus the reversible hydrogen electrode (RHE) at all pH values. Hence, we previously employed the CHE model to calculate the free energies of electrochemical reaction steps, with a reference potential of 0 V vs. RHE.

2) We have supplemented the nitrate reduction pathways on Cu_2O , Pd/ Cu_2O and Pd-Cl/ Cu_2O models at the potential of -0.12 V (theoretical nitrate reduction potential at pH 0) and -0.95 V vs. SHE (our research system at pH 14), which the potential is converted according to Nernst's equation ($E_{\text{RHE}} = E_{\text{SHE}} + 0.0591 \times \text{pH}$). At the potential of -0.12 V vs. SHE (Figure R9), the potential-determining step (PDS) in the reaction pathway of pure Cu_2O is the hydrogenation of *NO into *NOH (*NO + $\text{H}^+ + \text{e}^- \rightarrow$ *NOH) with a high Gibbs free energy change (ΔG) of 0.71 eV. Pd-Cl/ Cu_2O shows a significantly decreased PDS energy barrier (0.45 eV). In comparison, the PDS on Pd/ Cu_2O is the hydrogenation of *NO₂ into *NO₂H (*NO₂ + $\text{H}^+ + \text{e}^- \rightarrow$ *NO₂H), which presents the lowest ΔG of 0.28 eV. At the potential of -0.95 V vs. SHE (Figure R10), the PDS on the three models is the hydrogenation of *NO₂ into *NO₂H and the lower energy is required on Pd-Cl/ Cu_2O (0.40 eV) compared to Cu_2O (0.68 eV) and Pd/ Cu_2O (0.53 eV). The above results prove an easier process of nitrate reduction to ammonia occurred on Pd-Cl/ Cu_2O at the potential of -0.95 V vs. SHE.

Figure R9. Gibbs free energy diagram of various intermediates generated during NO₃RR over catalysts at the potential of -0.12 V vs. SHE.

Figure R10. Gibbs free energy diagram of various intermediates generated during NO₃RR over catalysts at the potential of -0.95 V vs. SHE.

Corresponding revision:

- Figure R9 has been added to **Supplementary Figure 5** (Page 9, Revised supplementary information).
- Figure R10 has been added to **Figure 2d** and **Supplementary Figure 4** (Page 8, Line 141, Revised manuscript, Page 8, Revised supplementary information).

Figure 2. d, Gibbs free energy diagram of various intermediates generated during NO₃RR over Pd/Cu₂O and Pd-Cl/Cu₂O at the potential of -0.95 V *versus* SHE. Inset: NO₃RR pathway of Pd-Cl/Cu₂O.

Supplementary Figure 4. Gibbs free energy diagram of various intermediates generated during NO₃RR over Cu₂O at the potential of -0.95 V *versus* SHE.

➤ The corresponding content “According to Nernst's equation ($E_{\text{RHE}} = E_{\text{SHE}} + 0.0591 \times \text{pH}$) and theoretical nitrate reduction potential ($E^0 = -0.12 \text{ V vs. RHE}$), the corresponding reaction potentials at pH 0 and 14 are -0.12 and -0.95 V *vs.* SHE, respectively.” has been added to the DFT computational details (Page 29, Lines 556-559, Revised manuscript).

➤ The corresponding content has been revised to “Next, after considering the effects of potential, pH on NO₃RR pathway with multiple possible branches (Supplementary Figs. 4-8), the optimal pathway on Cu₂O, Pd/Cu₂O and Pd-Cl/Cu₂O models at the potential of -0.95 V *versus* SHE was proposed and the corresponding ΔG of each intermediate was calculated (Fig. 2d). In such a sequential electron–proton transfer process (Supplementary Fig. 9)^{41,42}, the hydrogenation of *NO₂ into *NO₂H (*NO₂ + e⁻ + H⁺ → *NO₂H) was the potential-determining step (PDS), which involved a high ΔG of 0.68 eV over pure Cu₂O (Supplementary Fig. 4). Pd/Cu₂O also presented a relatively high ΔG of PDS (0.53 eV). Pd-Cl/Cu₂O showed the lowest ΔG of PDS (0.40 eV) and correspondingly advanced the progress of NO₃RR.” (Page 8, Lines 130-139, Revised manuscript).

➤ The content “To demonstrate the impact of potential and pH on the nitrate reduction process, we have calculated the pathways on Cu₂O, Pd/Cu₂O and Pd-Cl/Cu₂O models at the potential of -0.12 V (theoretical nitrate reduction potential at pH 0) and -0.95 V *vs.* SHE (our research system at pH 14), which the potential is converted according to Nernst's equation ($E_{\text{RHE}} = E_{\text{SHE}} + 0.0591 \times \text{pH}$). At the potential of -0.12 V *vs.* SHE

(Supplementary Fig. 5), the potential-determining step (PDS) in the reaction pathway of pure Cu₂O is the hydrogenation of *NO into *NOH (*NO + H⁺ + e⁻ → *NOH) with a high Gibbs free energy change (ΔG) of 0.71 eV. Pd-Cl/Cu₂O shows a significantly decreased PDS energy barrier (0.45 eV). In comparison, the PDS on Pd/Cu₂O is the hydrogenation of *NO₂ into *NO₂H (*NO₂ + H⁺ + e⁻ → *NO₂H), which presents the lowest ΔG of 0.28 eV. At the potential of -0.95 V vs. SHE (Fig. 2d and Supplementary Fig. 4), the PDS on the three models is the hydrogenation of *NO₂ into *NO₂H and the lower energy is required on Pd-Cl/Cu₂O (0.40 eV) compared to Cu₂O (0.68 eV) and Pd/Cu₂O (0.53 eV). The above results prove an easier process of nitrate reduction to ammonia occurred on Pd-Cl/Cu₂O at the potential of -0.95 V vs. SHE.” has been added to Supplementary Figure 5 (Page 9, Revised supplementary information).

***Comment 3:** There is no mention about the pH used in the calculations. The effects of pH have a drastic effect on adsorption energies of different species, particularly nitrate.*

Response: We have supplemented the nitrate reduction pathways on Cu₂O, Pd/Cu₂O and Pd-Cl/Cu₂O models at pH 0 and 14 as suggested by the reviewers. According to Nernst's equation ($E_{\text{RHE}} = E_{\text{SHE}} + 0.0591 \times \text{pH}$) and theoretical nitrate reduction potential ($E^0 = -0.12 \text{ V vs. RHE}$), the corresponding reaction potentials for these two pH values are -0.12 and -0.95 V vs. SHE. Thus, we calculate the reaction pathway on the three models at the potential of -0.12 V (pH 0) and -0.95 V vs. SHE (pH 14) to investigate the pH effect on the adsorption energies of different species, particularly nitrate. At pH 0 (Figure R11), Cu₂O (-1.39 eV), Pd/Cu₂O (-1.09 eV) and Pd-Cl/Cu₂O (-1.18 eV) all have strong nitrate adsorption ability. When the pH value increases to 14 (Figure R12), the corresponding nitrate adsorption energies of Cu₂O (-2.12 eV), Pd/Cu₂O (-1.64 eV) and Pd-Cl/Cu₂O (-1.21 eV) are larger than that of at pH 0. Therefore, Cu₂O-based catalysts are more favorable to adsorb nitrate under high pH conditions.

Figure R11. Gibbs free energy diagram of various intermediates generated during NO₃RR over catalysts at pH 0.

Figure R12. Gibbs free energy diagram of various intermediates generated during NO₃RR over catalysts at pH 14.

Corresponding revision:

- Figure R11 has been added to **Supplementary Figure 5** (Page 9, Revised supplementary information).
- Figure R12 has been added to **Figure 2d** and **Supplementary Figure 4** (Page 8, Line 141, Revised manuscript, Page 8, Revised supplementary information).

Figure 2. d, Gibbs free energy diagram of various intermediates generated during NO₃RR over Pd/Cu₂O and Pd-Cl/Cu₂O at the potential of -0.95 V *versus* SHE (pH 14). Inset: NO₃RR pathway of Pd-Cl/Cu₂O.

Supplementary Figure 4. Gibbs free energy diagram of various intermediates generated during NO₃RR over Cu₂O at the potential of -0.95 V *versus* SHE (pH 14).

➤ The corresponding content has been revised to “Next, after considering the effects of potential, pH on NO₃RR pathway with multiple possible branches (Supplementary Figs. 4-8), the optimal pathway on Cu₂O, Pd/Cu₂O and Pd-Cl/Cu₂O models at the potential of -0.95 V *versus* SHE was proposed and the corresponding ΔG of each intermediate was calculated (Fig. 2d). In such a sequential electron–proton transfer process (Supplementary Fig. 9)^{41,42}, the hydrogenation of *NO₂ into *NO₂H (*NO₂ + e⁻ + H⁺ → *NO₂H) was the potential-determining step (PDS), which involved a high ΔG of 0.68 eV over pure Cu₂O (Supplementary Fig. 4). Pd/Cu₂O also presented a relatively high ΔG of PDS (0.53 eV). Pd-Cl/Cu₂O showed the lowest ΔG of PDS (0.40 eV) and correspondingly advanced the progress of NO₃RR.” (Page 8, Lines 130-139, Revised manuscript).

➤ The corresponding content “According to Nernst's equation ($E_{RHE} = E_{SHE} +$

0.0591×pH) and theoretical nitrate reduction potential ($E^0 = -0.12$ V vs. RHE), the corresponding reaction potentials at pH 0 and 14 are -0.12 and -0.95 V vs. SHE, respectively.” has been added to the DFT computational details (Page 29, Lines 556-559, Revised manuscript).

Comment 4: *The authors report that they have calculated reaction pathways based on free energies. However, based on the results presented, this seems highly dubious. Nowhere is it mentioned how they calculate the zero point energy or the entropy of the reactions. The required phonon calculations are very expensive and are easy to get wrong in periodic systems. Therefore, details must be provided as to the conditions/methods used.*

Response: Based on the kind suggestion, we have provided more information about the method.

In the zero point energy or the entropy of the reactions calculation, we use finite difference to calculate the vibration frequency of Gama point (IBRION=5).

The formula is as follows:

Calculation of thermodynamic corrections

The zero-point energy for each species is calculate by

$$E_{ZPE} = \sum_i \frac{h\nu_i}{2}$$

where ν_i is the vibration frequency. The entropy contributions of translational, rotational, vibrational, and electronic motion can be calculated by

$$S_t = R \left\{ \ln \left[\left(\frac{2\pi m k_B T}{h^2} \right)^{\frac{3}{2}} \frac{k_B T}{P} \right] + \frac{5}{2} \right\}$$

$$S_r = R \left[\ln \left(\frac{T}{\sigma_r} * \frac{8\pi^2 I k_B}{h^2} \right) + 1 \right]$$

$$S_v = R \sum_i \left\{ \frac{h\nu_i}{k_B T} \frac{e^{-\frac{h\nu_i}{k_B T}}}{1 - e^{-\frac{h\nu_i}{k_B T}}} - \ln [1 - e^{-\frac{h\nu_i}{k_B T}}] \right\}$$

$$S_e = R * \ln(N+1)$$

where N is the number of unpaired electrons, R is the gas constant, P is the pressure,

k_B is the Boltzmann constant (Thermochemistry in gaussian. gaussian inc. 1-19 (2000)).

From the above formula, it can be seen that the vibration frequency of the catalyst substrate itself is small, its corresponding correction is very small and does not affect the calculation results. Thus, we mainly made corrections to gas molecules and adsorbents on the catalyst.

Corresponding revision:

➤ The corresponding formulas have been added to **DFT computational details**. (Page 30, Lines 580-590, Revised manuscript).

Comment 5: It is unclear in the reaction pathway if the authors used H^ as the reactant for protonation steps or solution phase H^+ . In their case, the pH (which again is not explicitly described or seemingly included in the energetic models) will have a large effect on the energetics of proton transfer.*

Response: More information has been introduced to clarify the potential ambiguity raised.

In the nitrate reduction reaction pathway, we use solution phase H^+ as the reactant for protonation steps. And the reaction process is following the formulas of $NO_3^- + 9H^+ + 8e^- \rightleftharpoons NH_3 + 3H_2O$.

We agree with the reviewer's opinion that pH will have a large effect on the energetics of H^+ transfer. To investigate the pH effect, we have supplemented the nitrate reduction pathways on Cu_2O , Pd/Cu_2O and $Pd-Cl/Cu_2O$ models at pH 0 and 14. According to Nernst's equation ($E_{RHE} = E_{SHE} + 0.0591 \times pH$) and theoretical nitrate reduction potential ($E^0 = -0.12$ V vs. RHE), the corresponding reaction potentials for these two pH values are -0.12 V (pH 0) and -0.95 V vs. SHE (pH 14). At pH 0 (Figure R13), the PDS in the reaction pathway of pure Cu_2O is the hydrogenation of $*NO$ into $*NOH$ ($*NO + H^+ + e^- \rightarrow *NOH$) with a high Gibbs free energy change (ΔG) of 0.71 eV. $Pd-Cl/Cu_2O$ shows a significantly decreased PDS energy barrier (0.45 eV). In comparison, the PDS on Pd/Cu_2O is the hydrogenation of $*NO_2$ into $*NO_2H$ ($*NO_2 + H^+ + e^- \rightarrow *NO_2H$), which presents the lowest ΔG of 0.28 eV. When the pH value increases to 14 (Figure R14), the PDS on the three models is the hydrogenation of $*NO_2$

into $^*\text{NO}_2\text{H}$ and the lower energy is required on Pd-Cl/Cu₂O (0.40 eV) compared to Cu₂O (0.68 eV) and Pd/Cu₂O (0.53 eV). These results prove an easier H⁺ transfer of nitrate intermediate occurred on Pd-Cl/Cu₂O at pH 14 (our research system).

Figure R13. Gibbs free energy diagram of various intermediates generated during NO₃RR over catalysts at pH 0.

Figure R14. Gibbs free energy diagram of various intermediates generated during NO₃RR over catalysts at pH 14.

Comment 6: *The model for the Pd-Cl is under described. For example, how many Cl ions were included, why was this chosen, what is the effect of more or fewer Cl ions, how was the negative charge on the Cl accounted for? None of these are described let alone justified. Without these answers, the validity of the results cannot be assessed.*

Response: We have added more details of the model based on the kind suggestion.

1) We have added two models (Figure R15), namely Pd atom coordinated with one Cl or three Cl, and compared them with the previously constructed configurations of Pd coordinated with two Cl.

Figure R15. The models of Pd-Cl/Cu₂O with different Cl numbers.

2) By comparing the calculation results of Gibbs free energy changes for H₂O dissociation and H⁺ release among these three models (Figure R16), we found that the model of Pd combining with three Cl is not conducive to H₂O dissociation (1.22 eV) and H⁺ desorption (0.37 eV). For Pd combining with one Cl, it could promote H₂O dissociation (0.46 eV), but has a relatively high H⁺ desorption energy barrier (0.25 eV). In contrast, Pd coordinated with two Cl is more favorable to release H⁺ (0.09 eV) while maintaining a strong H₂O dissociation ability (0.67 eV), which is beneficial for the hydrogenation of nitrate intermediates to ammonia. Therefore, we choose the model of Pd atoms coordinating with two Cl for NO₃RR process. Besides, through XAFS characterization and NO₃RR performance experiment results (Figures 3f, 5a and Supplementary Table 1), the as-prepared Pd-Cl/Cu₂O catalyst which Pd center is coordinated with two Cl atoms delivers a superior NH₃ yield rate and FE.

Figure R16. Gibbs free energy change of H₂O dissociation and H⁺ supply over Pd-Cl/Cu₂O with different numbers of Cl.

Figure 3. f, The fitting EXAFS spectra of Pd-Cl/Cu₂O. Inset: fitting model.

Supplementary Table 1 | The Pd K-edge EXAFS fitting results for Pd-Cl/Cu₂O.

Sample	Shell	CN	R(Å)	$\sigma^2(\text{Å}^2)$	$\Delta E_0(\text{eV})$	R factor
Pd-Cl/Cu ₂ O	Pd-Cl	1.964	2.34	0.007	22.7	0.0008
	Pd-Cu	3.004	2.52	0.002	7.80	

Figure 5. a, NH₃ yield rate and NH₃ FE of catalysts in a 1 M KOH with 56 mM NO₃⁻ electrolyte for 1 h electrolysis.

3) From the results of charge density difference in Figure R17, the two Cl atoms on Pd-Cl/Cu₂O both obtain 0.47 *e*, which is supplied by the coordinated Pd atom (0.13 *e*) in addition to the contribution from the Cu₂O substrate. Thus, the modification of Cl would seize the electrons of Pd atom due to Cl with the strongest first electron affinity (Nat. Commun. 13, 6875 (2022)). In addition, the binding energy shift between Cl 2*p*

and Pd 3d in XPS experiments (Supplementary Fig. 21c-d and 22) indicates that the electron transferred from Pd to Cl in Pd-Cl/Cu₂O under the Cl mediation.

Figure R17. The charge density difference between *H intermediate and catalysts. The isosurface level was $0.005 e \cdot \text{bohr}^{-3}$. The yellow and cyan colors represented positive and negative charge regions, respectively.

Supplementary Figure 21. c, Pd 3d and d, Cl 2p XPS spectra of samples.

Supplementary Figure 22. Cl 2p XPS spectra of catalysts.

Corresponding revision:

➤ Figures R15-16 have been added to **Supplementary Figures 2-3** (Pages 6-7, Revised supplementary information).

➤ The content “By comparing the calculation results of Gibbs free energy changes for H₂O dissociation and H⁺ release among the models of Pd-Cl/Cu₂O with different Cl numbers (Supplementary Fig. 3), we found that the model of Pd combining with three Cl is not conducive to H₂O dissociation (1.22 eV) and H⁺ desorption (0.37 eV) due to the coordination saturation. For Pd combining with one Cl, it could promote H₂O dissociation (0.46 eV), but has a relatively high H⁺ desorption energy barrier (0.25 eV). In contrast, Pd coordinated with two Cl is more favorable to release H⁺ (0.09 eV) while maintaining a strong H₂O dissociation ability (0.67 eV), which is beneficial for the hydrogenation of nitrate intermediates to ammonia.” has been added to **Supplementary Figure 3** (Page 7, Revised supplementary information).

➤ The corresponding content has been revised to “To test the halogen mediating effect, density functional theory (DFT) calculations were conducted on the models of Cu₂O, Pd-dispersed Cu₂O (Pd/Cu₂O) and Pd-Cl/Cu₂O with different Cl numbers (Supplementary Figs. 1-2). Cu₂O showed a high Gibbs free energy change for H₂O dissociation (ΔG_{*H_2O} of 0.98 eV, Fig. 2a), while the corresponding value on Pd/Cu₂O (0.71 eV) decreased, indicating the accelerated H₂O dissociation under the assistance of Pd SACs. However, the relatively strong *H adsorption on Pd/Cu₂O limited its desorption (ΔG_{H^+} of 0.17 eV). In contrast (Supplementary Fig. 3), Pd-Cl/Cu₂O which Pd atoms coordinated with two Cl was more favorable to generate H⁺ (ΔG_{H^+} of 0.09 eV) while maintaining a strong H₂O dissociation ability (ΔG_{*H_2O} of 0.67 eV).” (Page 7, Lines 110-119, Revised manuscript).

➤ Figure R17 has been added to **Figure 2b** (Page 8, Line 141, Revised manuscript).

➤ The corresponding content has been revised to “From the charge density difference (Fig. 2b), the two Cl atoms on Pd-Cl/Cu₂O both obtained 0.47 e, which is supplied by the coordinated Pd atom (0.13 e) in addition to the contribution from the Cu₂O substrate.” (Page 7, Lines 121-124, Revised manuscript).

Comment 7: It is unclear what the reference states of the molecules are, for example,

are the energetics references to H₂O and NO₃⁻ in the gas phase or some sort of aqueous reference.

Response: We use H₂O and NO₃⁻ in the gas phase as the energetics references.

Corresponding revision:

➤ The content “H₂O and NO₃⁻ in the gas phase are as the energetics references.” has been added (Page 28, Lines 548-549, Revised manuscript).

Comment 8: In terms of the results themselves, there are some questions. The authors state that the pure Pd is limited by the large *H dissociation free energy (0.69 eV). This is confusing as once formed and on the surface the *H should be able to react (at least thermodynamically) with the nitrate intermediate. Therefore the H desorption barrier seems irrelevant.

Response: We have introduced additional explanations to clarify the question as the following.

1) We have supplemented the calculations about the Gibbs free energy changes of H₂O dissociation to release H⁺ and nitrate reduction process on Pd-Cl/Cu₂O model at pH 14 conditions (-0.95 V vs. SHE). As shown in Figure R18, the energy barrier of H⁺ release (0.26 eV) is significantly lower than that of *NO₂ hydrogenation (PDS of nitrate reduction pathway, 0.40 eV). This result indicates that H⁺ release is more easily to occur on the Pd-Cl/Cu₂O. Therefore, it is necessary to consider H⁺ desorption in the reaction system.

Figure R18. The Gibbs free energy change of H⁺ supply and *NO₂ hydrogenation over Pd-Cl/Cu₂O at the potential of -0.95 V vs. SHE.

2) In the nitrate reaction pathway of Pd-Cl/Cu₂O at the potential of -0.95 V vs. SHE at alkaline conditions, nitrate intermediates are adsorbed on the Cu site, while *H is on the Pd site. At this potential, *H and *H coupling may occur at the Pd site, resulting in the generation of by-product H₂. Therefore, to achieve the nitrate reduction into ammonia, it is necessary to consider the desorption of *H from Pd site and thus the formed H⁺ could subsequently participate in the hydrogenation of nitrate intermediates adsorbed on the Cu site ($\text{NO}_3^- + 9\text{H}^+ + 8e^- \rightleftharpoons \text{NH}_3 + 3\text{H}_2\text{O}$). Furthermore, the *in situ* Raman, *in situ* ATR-IR and NO₃RR performance experiments (Figures 4 and 5a) demonstrate that Pd-Cl/Cu₂O catalysts could promote dangling O-H water dissociation and fast *H desorption. The constructed local H⁺-abundant environments effectively support the free H⁺ feeding to nitrate intermediate hydrogenation and correspondingly Pd-Cl/Cu₂O delivers a superior NH₃ yield rate and FE in alkaline NO₃RR. For the Pd/Cu₂O catalysts, the difficulty in donating H⁺ lead to the lower NO₃RR performance due to the strong binding of *H on Pd without Cl mediation. Hence, it is important to consider the desorption of *H in our reaction system.

Figure 4. *NO intermediates hydrogenation. **a**, *In situ* Raman spectra of Pd-Cl/Cu₂O. **b**, Schematic diagram of the local H⁺-abundant environment construction over Pd-Cl/Cu₂O. **c-e**, *In situ* Raman spectra of Pd-Cl/Cu₂O (**c**), corresponding peak area (**d**) and Raman shift (**e**) of various interfacial H₂O structures. **f,g**, *In situ* ATR-IR spectra of Pd-Cl/Cu₂O (**f**) and Pd/Cu₂O (**g**) catalysts. Si-O signal was derived from the reduction of surface SiO₂ on the Si semi-cylindrical prism substrate under the applied potentials.

Figure 5. a, NH₃ yield rate and NH₃ FE of catalysts in a 1 M KOH with 56 mM NO₃⁻ electrolyte for 1 h electrolysis.

***Comment 9:** Similarly, examining the dissociation of H-OH on the Pd-Cl, the entire process is only downhill from solution phase H₂O by 0.1 eV. This is very small, especially if the H₂O is referenced to the gas phases, which would significantly overestimate the binding compared to a proper solution phase reference point.*

Response: We implemented additional calculations to clarify the kind question raised.

We recalculated the Gibbs free energy changes of H₂O dissociation and H⁺ release on Cu₂O, Pd/Cu₂O and Pd-Cl/Cu₂O models (Figure R19). This trend is similar to the previous result, where the dissociation of gas phases H₂O into *H+*OH on Pd-Cl/Cu₂O model is only downhill by 0.08 eV. Although this value is small, we consider it is also possible to occur, as follows:

1) The impact of catalysts. The energy barrier of 0.08 eV is not only related to the energy change of gas phase H₂O dissociation, but also affected by the interaction between the catalyst and different reaction intermediate species. That is, catalyst also plays a role in the energy change of H₂O dissociation.

2) The reliability of the methodology. We have referred to the relevant papers on the H₂O dissociation. For instance, the energy changes between *+H₂O and *H+*OH are only 0.051, 0.04, and 0.02 eV on IrMo (001)-OH sample (Nat. Commun. 13, 5497, (2022)), Pt-Co catalyst (Nano Res. 15, 6, 4958–4964, (2022)), and S-NiFe₂O₄ (Angew. Chem. 133, 14236–14242, (2021)) respectively, which are all relatively small. Thus, the calculated energy barrier of 0.08 eV on the Pd-Cl/Cu₂O is reasonable.

3) The trend of free energy change. In this reaction path, apart from the value of intermediate energy change, we pay more attention to the trend of reaction intermediate energy change on different catalysts. That is, Pd-Cl/Cu₂O is more favorable to promote H₂O dissociation and H⁺ release compared with Cu₂O and Pd/Cu₂O, which is consistent with our experimental results (Figure 4).

Figure R19. Gibbs free energy change of H₂O dissociation and H⁺ supply over catalysts.

Figure 4. **a**, *In situ* Raman spectra of Pd-Cl/Cu₂O. **b**, Schematic diagram of the local H⁺-abundant environment construction over Pd-Cl/Cu₂O. **c-e**, *In situ* Raman spectra of Pd-Cl/Cu₂O (**c**), corresponding peak area (**d**) and Raman shift (**e**) of various interfacial H₂O structures.

Corresponding revision:

➤ Figure R19 has been updated to **Figure 2a** (Page 8, Line 141, Revised manuscript).

Comment 10: *The NO₃ reduction pathway is highly complex with multiple possible branches. The authors need to describe how they choose this particular pathway and if they considered alternative pathways. If they did not, then it is very possible that these results are uninformative as to the chemistry at play.*

Response: We agree with the reviewer's opinion that nitrate reduction pathway is highly complex with multiple possible branches. Correspondingly, we choose the optimal pathway after comparing alternative pathways at different potentials. During the nitrate reduction process of Cu₂O at the potential of -0.12 and -0.95 V vs. SHE (Figure R20), the energy is downhill in *NO₂H → *NO but uphill in *NO₂H → *NO₂H₂. And the energy is considerably downhill in *NOH → *NHOH and *NHOH → *NH₂OH than that of *NOH → *N and *NHOH → *NH, respectively. Therefore, Cu₂O is more likely to undergo the pathway of *NO₃ → *NO₂ → *NO₂H → *NO → *NOH → *NHOH → *NH₂OH → *NH₂ → *NH₃. For Pd/Cu₂O and Pd-Cl/Cu₂O at various potentials (Figures R21-22), the energy is uphill or relatively small downhill in *NO₂H → *NO₂H₂ and *NOH → *NHOH than that of *NO₂H → *NO and *NOH → *N, respectively. Hence, Pd/Cu₂O and Pd-Cl/Cu₂O catalysts are more inclined to follow the pathway of *NO₃ → *NO₂ → *NO₂H → *NO → *NOH → *N → *NH → *NH₂ → *NH₃.

Figure R20. Gibbs free energy diagram of various intermediates generated during NO₃RR over Cu₂O at the potential of -0.12 V and -0.95 V vs. SHE.

Figure R21. Gibbs free energy diagram of various intermediates generated during NO₃RR over Pd/Cu₂O at the potential of -0.12 V and -0.95 V vs. SHE.

Figure R22. Gibbs free energy diagram of various intermediates generated during NO₃RR over Pd-Cl/Cu₂O at the potential of -0.12 V and -0.95 V vs. SHE.

Corresponding revision:

- Figures R20-22 have been added to **Supplementary Figures 6-8** (Pages 10-12, Revised supplementary information).
- The content “Nitrate reduction pathway is highly complex with multiple possible branches. Correspondingly, we choose the optimal pathway after comparing alternative pathways at different potentials. During the nitrate reduction process of Cu₂O at the potential of -0.12 and -0.95 V vs. SHE (Supplementary Fig. 6), the energy is downhill in *NO₂H → *NO but uphill in *NO₂H → *NO₂H₂. And the energy is considerably

downhill in $*NOH \rightarrow *NHOH$ and $*NHOH \rightarrow *NH_2OH$ than that of $*NOH \rightarrow *N$ and $*NHOH \rightarrow *NH$, respectively. Therefore, Cu_2O is more likely to undergo the pathway of $*NO_3 \rightarrow *NO_2 \rightarrow *NO_2H \rightarrow *NO \rightarrow *NOH \rightarrow *NHOH \rightarrow *NH_2OH \rightarrow *NH_2 \rightarrow *NH_3$. For Pd/Cu_2O and $Pd-Cl/Cu_2O$ at various potentials (Supplementary Figs. 7-8), the energy is uphill or relatively small downhill in $*NO_2H \rightarrow *NO_2H_2$ and $*NOH \rightarrow *NHOH$ than that of $*NO_2H \rightarrow *NO$ and $*NOH \rightarrow *N$, respectively. Hence, Pd/Cu_2O and $Pd-Cl/Cu_2O$ catalysts are more inclined to follow the pathway of $*NO_3 \rightarrow *NO_2 \rightarrow *NO_2H \rightarrow *NO \rightarrow *NOH \rightarrow *N \rightarrow *NH \rightarrow *NH_2 \rightarrow *NH_3$.” has been added to Supplementary Figures 6-8 (Page 12, Revised supplementary information).

➤ The corresponding content has been revised to “Next, after considering the effects of potential, pH on NO_3RR pathway with multiple possible branches (Supplementary Figs. 4-8), the optimal pathway on Cu_2O , Pd/Cu_2O and $Pd-Cl/Cu_2O$ models at the potential of -0.95 V vs. SHE was proposed and the corresponding ΔG of each intermediate was calculated (Fig. 2d). In such a sequential electron–proton transfer process (Supplementary Fig. 9)^{41,42}, the hydrogenation of $*NO_2$ into $*NO_2H$ ($*NO_2 + e^- + H^+ \rightarrow *NO_2H$) was the potential-determining step (PDS), which involved a high ΔG of 0.68 eV over pure Cu_2O (Supplementary Fig. 4). Pd/Cu_2O also presented a relatively high ΔG of PDS (0.53 eV). $Pd-Cl/Cu_2O$ showed the lowest ΔG of PDS (0.40 eV) and correspondingly advanced the progress of NO_3RR .” (Page 8, Lines 130-139, Revised manuscript).

Comment 11: *In Figure 2b, the authors need to elaborate what the $0.03 e^-$ is, it is unclear.*

Response: According to reviewer’s suggestion, we have recalculated the charge density difference (Figure R17) and made the clarification.

The previous $0.03 e$ has been updated to $0.13 e$, which represents the electrons transferred from $Pd-Cl/Cu_2O$ catalysts to $*H$ atoms, and $0.09 e$ from Pd/Cu_2O catalysts to $*H$ atoms.

Figure R17. b, The charge density difference between *H intermediate and catalysts. The isosurface level was $0.005 e \cdot \text{bohr}^{-3}$. The yellow and cyan colors represented positive and negative charge regions, respectively. $0.13 e$ was the electrons transferred from Pd-Cl/Cu₂O to *H, and $0.09 e$ was the electrons transferred from Pd/Cu₂O to *H.

Corresponding revision:

- Figure R17 has been added to **Figure 2b** (Page 8, Line 141, Revised manuscript).
- The corresponding content has been revised to “**Thus, under the regulation of Cl ligand, the shifted *d*-band center of Pd enabled *H on the catalyst to obtain more electrons ($0.13 e$) and make *H more unstable to promote the H⁺ release (Fig. 2b).**” (Page 7, Lines 127-129, Revised manuscript).
- The content “ **$0.13 e$ was the electrons transferred from Pd-Cl/Cu₂O to *H, and $0.09 e$ was the electrons transferred from Pd/Cu₂O to *H.**” has been added to the legend of **Figure 2b** (Page 8, Lines 145-146, Revised manuscript).

Comment 12: In general, it would be helpful early on to establish that PdCl samples are generated by having Cl in the solution when synthesizing the material. It is unclear if the Cl effect arises from the mere presence of Cl or from the synthesis. To confirm one way or the other, additional experiments where Cl ions are added post synthesis to the Pd/Cu₂O samples.

Response: We appreciate the reviewer for posing this insightful question. To elucidate the source of Cl that induces enhanced NH₃ production activity, we supplemented two groups of control experiments.

The first group is to dip a certain amount of KCl on Pd/Cu₂O electrode, while the second group is to add KCl to the reaction electrolyte in Pd/Cu₂O systems for testing NO₃RR performance. As shown in **Figure R23**, the NH₃ yield rate and NH₃ FE of the

two control experiments are comparable to the Pd/Cu₂O, and far lower than that of Pd/Cl-Cu₂O. This result proves that introducing Cl species in the sample synthesis to prepare Cl coordinated Pd single atom catalysts can effectively achieve impressive NO₃RR performance, through Cl ligand regulating the electronic structure of Pd atoms to mediate the proton feeding. While simply adding Cl ions to the reaction system would not significantly improve the NO₃RR performance. Hence, the Cl effect arises from the synthesis.

Figure R23. a,c NH₃ yield rate and b,d NH₃ FE of various catalysts in a 1 M KOH with 56 mM NO₃⁻ electrolyte for 1 h electrolysis.

Corresponding revision:

- The content “Control experiments further demonstrated that the mediated effect originated from the Cl ligand of Pd-Cl/Cu₂O rather than the free Cl ions in the system (Supplementary Fig. 43).” has been added (Page 15, Lines 268-270, Revised manuscript).
- Figure R23 has been added to Supplementary Figure 43 (Page 48, Revised supplementary information).
- The content “To elucidate the source of Cl that induces enhanced NH₃ production activity, we supplemented two groups of control experiments. The first group is to dip a certain amount of KCl on Pd/Cu₂O electrode, while the second group is to add KCl

to the reaction electrolyte in Pd/Cu₂O systems for testing NO₃RR performance. As shown in Supplementary Fig. 43, the NH₃ yield rate and NH₃ FE of the two control experiments are comparable to the Pd/Cu₂O, and far lower than that of Pd/Cl-Cu₂O. This result proves that introducing Cl species in the sample synthesis to prepare Cl coordinated Pd single atom catalysts can effectively achieve impressive NO₃RR performance, through Cl ligand regulating the electronic structure of Pd atoms to mediate the proton feeding. While simply adding Cl ions to the reaction system would not significantly improve the NO₃RR performance. Hence, the Cl effect arises from the synthesis.” has been added to Supplementary Figure 43 (Page 48, Revised supplementary information).

Comment 13: The discussion of the effect of other halogen mediation is weak.

Response: We have introduced discussion as suggested. Since Cl has a strong ability to regulate the electronic structure of single atoms due to its strongest first electron affinity among halogen elements (*Nat. Commun.* 13, 6875 (2022)), our manuscript mainly focuses on modulation of Cl ligand as the object of study. While other halogen elements serve as extensions to demonstrate the universality of halogen-mediated effect strategies. We agree with the reviewer's opinion, and we have conducted a more detailed analysis and description of other halogen-mediated alkaline NO₃RR performance to demonstrate the broad applicability of this strategy.

Corresponding revision:

➤ The corresponding content has been revised to “In addition to Cl mediated strategy, this performing principle with high NH₃ performance was available for other halogen ligand systems, such as Pd-F/Cu₂O, Pd-Br/Cu₂O and Pd-I/Cu₂O. These catalysts were successfully synthesized using a similar wet-immersion and H₂ calcination approach, as proved by XRD and XPS characterization (Supplementary Figs. 58-61). Besides, Pd-(F, Br, I)/Cu₂O catalyst exhibited a higher NH₃ activity and selectivity for alkaline NO₃RR than those of Cu₂O matrix and Pd/Cu₂O. Their NH₃ yield rates in a 1 M KOH with 56 mM NO₃⁻ electrolyte at -0.4 V *versus* RHE were ~3.7, ~3.1, ~2.6-fold of Cu₂O, and ~1.7, ~1.4, ~1.2-fold of Pd/Cu₂O, respectively (Supplementary Fig. 62a). The

corresponding NH₃ FE value were ~2.9, ~2.3, ~2.1-fold of Cu₂O, and ~1.3, ~1.2, ~1.1-fold of Pd/Cu₂O, respectively (Supplementary Fig. 62b). Therefore, halogen-mediate strategy has expansive universality in enhancing alkaline NO₃RR performance.” (Page 18, Lines 339-341, Page 19, Lines 342-350, Revised manuscript).

***Comment 14:** Given the use of single-atom catalysis for the work, it is highly surprising that the authors did not reference key works on single atom catalysis for nitrate reduction, specifically by Wu et al¹ and Chen et al.² Here the form examined Pd single atoms in Cu, reflecting the current work, and the latter used Ru in Cu while coupling electrochemical conversion to NH₃ recovery in a production train highly parallel to that in this work. At a minimum, these contributions should be noted.*

1. Wu, X.; Nazemi, M.; Gupta, S.; Chismar, A.; Hong, K.; Jacobs, H.; Zhang, W.; Rigby, K.; Hedtke, T.; Wang, Q.; Stavitski, E.; Wong, M. S.; Muhich, C.; Kim, J.-H., Contrasting Capability of Single Atom Palladium for Thermocatalytic versus Electrocatalytic Nitrate Reduction Reaction. ACS Catalysis 2023, 13, (10), 6804-6812.
2. Chen, F.-Y.; Wu, Z.-Y.; Gupta, S.; Rivera, D. J.; Lambeets, S. V.; Pecaut, S.; Kim, J. Y. T.; Zhu, P.; Finprock, Y. Z.; Meira, D. M., Efficient conversion of low-concentration nitrate sources into ammonia on a Ru-dispersed Cu nanowire electrocatalyst. Nature nanotechnology 2022, 1-9.

Response: Thank you for your comment. The two literatures mentioned by reviewer have a great reference value for our work. To reflect their contributions, we have supplemented and cited the work of Wu et al (ACS Catal. 13, 6804-6812 (2023)). in our manuscript. Besides, we have highlighted the previously cited the work of Chen et al. (Nat. Nanotechnol. 17, 759-767 (2022)) (on page 4, line 76 of the revised manuscript, as well as on page 74, line 1330 of the revised supplementary information).

Corresponding revision:

➤ The references (ACS Catal. 13, 6804-6812 (2023), #16; Nat. Nanotechnol. 17, 759-767 (2022), #18.) have been cited (Page 3, Line 60, Page 3, Line 65, Page 4, Line 76, and Page 17, Line 304, Revised manuscript, Page 74, Line 1330, Revised supplementary information).

REVIEWER COMMENTS

Reviewer #1 (Remarks to the Author):

In the revised version by Liao Fu et al., most of the raised problems have been solved. However, there are some new concerns.

First, it is unusual to choose -0.12 V vs. RHE as the theoretical potential for nitrate reduction. The theoretical potentials that appeared in the reported literature were more positive (Chem. Rev. 2009, 109, 2209-2244.), like 0.69 V vs. RHE at pH=14 (J. Am. Chem. Soc. 2020, 142, 5702-5708.). It would be more persuasive if the authors provided support.

Second, the author stated the H⁺ accumulation on Pd/Cu₂O by introducing Cl under alkaline conditions. However, based on the ion product constant of water, the bulk concentration of H⁺ in the alkaline solution (pH=14) was about 1×10⁻¹⁴ mol/L. Besides, it is widely accepted that the local pH will increase under electroreduction conditions. Is the accumulated H⁺ stable under the reaction condition?

Reviewer #2 (Remarks to the Author):

Publish as is

Reviewer #3 (Remarks to the Author):

Overall the authors have made many clarifications to their work. However, the clarification led to further concerns, particularly again in their computational work. Additionally, these details have not revealed the depth of new insights expected of Nature journals, as single atom catalysts for nitrate reduction and the effects of halides are generally known. As such, I cannot recommend publishing in Nature Communications. Below, please find specific comments on the quality of the revised work.

The authors now present the basic information about their DFT calculations and are referencing their energetics to pH and potential. However, with regards to the latter, the information provided suggests that these energetic pathways have errors. The authors suggest that they account for pH and potential through the Nernst equation. However, they do not comment on the inclusion of pH effects on the free

energies of the reactive species (which is crucial) nor their hydrogen sources. These effects are important and must be considered, as is commonly done in the field.

The authors state that they examined the potentials of -0.12 and -0.95V for pH 0 and 14 respectively. However, these values do not match the experimental conditions. These conditions are used for determining the rate limiting steps. However, it is not necessarily these step energetics which are rate limiting. The rate limiting behavior cannot be extracted without calculation of the reaction barriers.

The authors have indeed included equations as to how entropic effects are included generally; however, they state that they actually neglect these entropy effects on the solid systems in their rebuttal. This statement is not present in the paper. Thus the paper is very misleading. These assumptions must be included in the main text and specifically clarify that entropic effects of the solids were not accounted for.

The authors now justify their selection of two Cl ions by the fact that it gives beneficial results. But the selection of a model based on the desired result is not inappropriate. Rather the mode should be evaluated as to what system is energetically favorable, then the behavior can be extracted.

The authors state that they take gas phase H₂O and NO₃⁻ as the references for their DFT analysis. This is completely unrealistic as the energy of an ion in the gas phase is significantly destabilized compared to one in an aqueous solution. Further water is also stabilized by energetically by condensing. Taken together with their assumptions about entropy of the reactant molecules, this likely leads to massive over prediction of binding. The inclusion of these effects is well reported in the nitrate reduction literature.

The fact that the addition of Cl after synthesis did not change improve catalytic behavior suggests that the Cl possibly acts more as a synthetic directing agent rather than as a co-catalyst. This concept should be explored further and clarified.

Manuscript number: NCOMMS-23-34364A

Title: Sustainable conversion of alkaline nitrate to ammonia at activities greater than 2
A cm⁻²

A Point-to-Point Response to Reviewer's Comments

Dear Editor and Reviewers,

We sincerely appreciate the Editor for giving us the opportunity to revise our manuscript. We highly appreciate Reviewer#1's recognition and Reviewer#2's support and approval of our article, and thank Reviewer#3 for the pertinent comments to improve our work. We have incorporated the corresponding changes into the manuscript (**highlighted in the revised manuscript**) and supporting information to address the editor's concerns and the requests of the reviewers. These changes are specified and discussed in a point-to-point response to the reviewers' comments, as shown below:

Reviewer #1: In the revised version by Liao Fu et al., most of the raised problems have been solved. However, there are some new concerns.

Response: Thank you very much for your recognition and support of our previous revision. Your comments are valuable and very helpful for improving the quality of our paper. In the present revised version, we have acted upon to address the reviewer's new concerns, which were constructive to further improve our work.

Comment 1: First, it is unusual to choose -0.12 V vs. RHE as the theoretical potential for nitrate reduction. The theoretical potentials that appeared in the reported literature were more positive (Chem. Rev. 2009, 109, 2209-2244.), like 0.69 V vs. RHE at pH=14 (J. Am. Chem. Soc. 2020, 142, 5702-5708.). It would be more persuasive if the authors provided support.

Response: We thank the reviewer for the meticulous examination. The relevant references are cited to clarify why we choose -0.12 V vs. RHE as the theoretical potential for nitrate reduction.

As many references introduced, the theoretical potential for nitrate reduction to ammonia is -0.12 V vs. SHE (*Small Methods* **4**, 2000672 (2020), *Chem. Soc. Rev.* **50**, 6720-6733 (2021), *Chem. Eng. J.* **403**, 126269 (2021)). According to Nernst's equation ($E_{\text{RHE}} = E_{\text{SHE}} + 0.0591 \times \text{pH}$), the theoretical potential is -0.12 V vs. RHE at pH=0 or ~-0.69 V vs. RHE at pH=14. In fact, both of them (-0.12 V and 0.69 V vs. RHE) are converted from different pH values under the theoretical potential of nitrate reduction of -0.12 V vs. SHE. Thus, to avoid misunderstanding, we have added the description of pH when the nitrate reduction potential appears in the manuscript, and also cited the references (*Chem. Rev.* **109**, 2209-2244 (2009), #71; *J. Am. Chem. Soc.* **142**, 5702-5708 (2020), #30.) in the revised manuscript.

Corresponding revision:

➤ The corresponding content “The theoretical potential for nitrate reduction to ammonia was 0.69 V versus RHE at pH=14.^{30,71}” has been added (Lines 437-438, Revised manuscript).

➤ The corresponding notes have been revised to “The theoretical potential for nitrate reduction to ammonia was -0.12 V vs. SHE. According to the calculation of Nernst's equation ($E_{\text{RHE}} = E_{\text{SHE}} + 0.0591 \times \text{pH}$), the corresponding theoretical potential was -0.12 V vs. RHE at pH=0.” (Lines 252-254, Revised supplementary information).

Comment 2: Second, the author stated the H⁺ accumulation on Pd/Cu₂O by introducing Cl under alkaline conditions. However, based on the ion product constant of water, the bulk concentration of H⁺ in the alkaline solution (pH=14) was about 1×10⁻¹⁴ mol/L. Besides, it is widely accepted that the local pH will increase under electroreduction conditions. Is the accumulated H⁺ stable under the reaction condition?

Response: Thank the reviewer for raising the question about the stability of accumulated H⁺. We have offered more explanations as the following and added them

to the revised manuscript.

We agree with the reviewer's opinion that the bulk concentration of H^+ in the alkaline electrolyte (pH=14) is about 1×10^{-14} mol/L, and the local pH will increase under electroreduction process for most catalysts. Under this condition, the alkaline NO_3RR process is significantly limited by the insufficient supply of H^+ .

To address this issue, we employ the halogen-mediated H^+ feeding strategy to facilitate the H_2O dissociation and the H^+ release, which could regulate the local pH environment on the catalyst surface, for the effective conversion of NO_3^- to NH_3 . The accumulated H^+ can stably exist at the reaction interface of Pd-Cl/ Cu_2O under the reaction conditions for the following reasons:

(1) According to the *in situ* electrochemical Raman spectra results of Pd-Cl/ Cu_2O , the peak at $\sim 1770\text{ cm}^{-1}$ can be assigned to the H_3O^+ intermediate species (Fig. 4a). As shown in Fig. 4b, the emerged H_3O^+ indicates the high-rate H_2O dissociation on the catalyst surface and thus forming the local acid-like environment (*Nat. Commun.* **10**, 4876 (2019), *Nat. Commun.* **13**, 2024 (2022).). It is worth noting that the detection limit of Raman spectroscopy is approximately ppm level for hydration molecules (*Food Bioprocess. Technol.* **6**, 710–718 (2013), *Nat. Commun.* **13**, 2024 (2022).), which the concentration is higher than that of the content of H^+ ionized from H_2O in 1 M KOH solution. Such a high level of H_3O^+ cannot be ubiquitous in KOH but can accumulate only within a local region around the catalyst surface.

(2) The N-H signals in *in situ* ATR-IR spectra suggest the effective hydrogenation process of NO_3^- intermediate (Fig. 4f), reflecting the stable supply of H^+ within the reaction interface.

(3) From the conversion tests of NO_3^- to NH_3 over Pd-Cl/ Cu_2O (Fig. 5c), the continuously decreased content of NO_3^- and the corresponding increased NH_3 concentration as the increase of reaction time verify the smooth H^+ feeding for the hydrogenation of NO_3^- to NH_3 . These results could reveal the stable characteristic of accumulated H^+ at the reaction interface of catalysts.

(4) Similar researches have also been reported that the H^+ generated from fast H_2O dissociation could stably accumulate on the local cathode surface in alkaline bulk

electrolyte conditions (*Nat. Mater.*, **22**, 1022-1029 (2023), *Adv. Funct. Mater.* **33**, 2304852 (2023)).

Thus, we consider that the accumulated H^+ on the surface of Pd-Cl/Cu₂O catalyst is relatively stable under the reaction condition.

Fig. 4 | NO₃⁻ intermediates hydrogenation. **a**, *In situ* Raman spectra of Pd-Cl/Cu₂O. **b**, Schematic diagram of the local H⁺-abundant environment construction over Pd-Cl/Cu₂O. **c-e**, *In situ* Raman spectra of Pd-Cl/Cu₂O (**c**), corresponding peak area (**d**) and Raman shift (**e**) of various interfacial H₂O structures. **f,g**, *In situ* ATR-IR spectra of Pd-Cl/Cu₂O (**f**) and Pd/Cu₂O (**g**) catalysts. Si-O signal was derived from the reduction of surface SiO₂ on the Si semi-cylindrical prism substrate under the applied potentials.

Figure 5. c, NO₃⁻ removal of catalysts measured in a 1 M KOH with 56 mM NO₃⁻ electrolyte (equals 790.3 µg mL⁻¹ NO₃⁻-N) at -0.4 V *versus* RHE. After 1 h electrolysis, only 7.1 µg mL⁻¹ of NO₃⁻-N

and $0.85 \mu\text{g mL}^{-1}$ of NO_2^- -N remained, both below the WHO regulations for drinking water (NO_3^- -N < $11.3 \mu\text{g mL}^{-1}$ and NO_2^- -N < $0.91 \mu\text{g mL}^{-1}$).

Corresponding revision:

➤ The corresponding content “Notably, the emerging H_3O^+ peak (1770 cm^{-1}) proved that the $^*\text{H}$ generated by H_2O dissociation was immediately desorbed from Pd-Cl/ Cu_2O surface, to construct local H^+ -abundant environments in high-pH conditions (Figs. 4a-b)^{51,52}. The formed H^+ could stably accumulate on the local cathode surface.” has been revised (Lines 205-208, Revised manuscript).

Reviewer #2: Publish as is

Response: Thank you very much for your support and approval of our work.

Reviewer #3: Overall the authors have made many clarifications to their work. However, the clarification led to further concerns, particularly again in their computational work. Additionally, these details have not revealed the depth of new insights expected of Nature journals, as single atom catalysts for nitrate reduction and the effects of halides are generally known. As such, I cannot recommend publishing in Nature Communications. Below, please find specific comments on the quality of the revised work.

Response: Thank you for taking time and effort to carefully examine our manuscript. We apologize most sincerely for the previous revision which did not meet the reviewer's expectations. We carefully studied the reference about the calculation methods commonly recognized in nitrate reduction (*Nat. Nanotechnol.* **17**, 759-767 (2022), *ACS Catal.* **13**, 6804-6812 (2023), *ACS EST Engg.*, DOI: 10.1021/acsestengg.1023c00207 (2023).), and recalculated our computational work. Besides, we have carefully considered each of the reviewer's concerns and made every effort to resolve them that follows:

Concern #1 - The influence of potential and pH on the free energies of the reactive species, and the hydrogen sources in alkaline nitrate reduction pathway should be considered. We have reconsidered the influence of protonation energy (including pH effects) and potential effects using thermodynamic Hess cycles, and H₂O was used as the proton source in alkaline nitrate reaction pathways.

Concern #2 - The rate-limiting step of nitrate reduction pathway should be determined by reaction barrier calculations under experimental conditions. According to previous research (*Energy Environ. Sci.* **3**, 1311-1315 (2010), *Nat. Energy*

5, 605-613 (2020), *Nat. Commun.* **12**, 2870 (2021).), free energy changes of reaction can evaluate the activity of catalysts. Thus, we calculated the free energy changes of nitrate reduction pathway under experimental conditions (-0.6 V vs. RHE, pH=14), and analyzed the potential-determining steps of pathway. We also attempted to calculate the reaction barrier, but we have not made progress, due to the limited computational resources and time-consuming reaction barrier calculations.

Concern #3 - Whether to consider the entropic effects of the solids should be clarified in the paper. We supplemented the description that the entropic effects of the catalyst substrate were not considered in the section of calculation method.

Concern #4 - The model selection should be explained. We supplemented the analysis of choosing models with the energetically favorable system in the revised paper.

Concern #5 - The energetics references of H₂O and NO₃⁻ should be rechecked. We modified the reference phases of H₂O and NO₃⁻, and used the aqueous phases as the references for DFT analysis to reflect a more realistic situation.

Concern #6 - The role of Cl in the Pd-Cl/Cu₂O should be analyzed. We analyzed the XAFS, XPS characterizations, and performance experiments to illuminate that Cl acted as a synthetic directing agent in Pd-Cl/Cu₂O.

We hope that these modifications have improved the strictness and scientificity of this article, meeting the requirements of the reviewer and revealing the depth of new insights expected of the Nature journal. Thanks again for the reviewer's constructive and helpful comments for improving the quality of our work.

Comment 1: The authors now present the basic information about their DFT calculations and are referencing their energetics to pH and potential. However, with regards to the latter, the information provided suggests that these energetic pathways have errors. The authors suggest that they account for pH and potential through the Nernst equation. However, they do not comment on the inclusion of pH effects on the free energies of the reactive species (which is crucial) nor their hydrogen sources. These effects are important and must be considered, as is commonly done in the field.

Response: We have followed the more rigorous and scientific calculation method, from the authoritative references in nitrate reduction (*Nat. Nanotechnol.* **17**, 759-767 (2022), *ACS Catal.* **13**, 6804-6812 (2023), *ACS EST Engg.*, DOI: 10.1021/acsestengg.1023c00207 (2023).), to reconsider the potential and pH effects on the free energies of the reactive species, as well as to reanalyze the hydrogen sources in nitrate reduction. The response consists of the following three aspects:

(1) H₂O as the proton source in alkaline conditions. The pH has an influence on the major proton donor in the research system. For alkaline nitrate reaction pathways (pH=14), we have considered H₂O as the proton source. Under this condition, the H* path will be through the alkaline pathway.

$$G_{\text{OH}^-} - G_{\text{H}_2\text{O}} - G_{e^-} = G_{1/2\text{H}_2} - eU_{\text{RHE}}$$

For electrochemical steps in nitrate reduction, the free energy changes are calculated using the products and reactants of the following reaction equation:

$$\Delta G(U) = G_{\text{XH}^*} - G_{\text{X}^*} + G_{\text{OH}^-} - G_{\text{H}_2\text{O}} - G_{e^-} = G_{\text{XH}^*} - G_{\text{X}^*} + G_{1/2\text{H}_2} - eU_{\text{RHE}}$$

(2) The calculation method for pH and potential effects. The free energies of adsorption of ionic species are calculated using thermodynamic Hess cycles, which include the effects of entropy, solvation energy, protonation energy (including pH effects) and potential effects. The Gibbs energy formulas for nitrate reduction reaction steps are reported by Muhich et. al. (*ACS EST Engg.*, DOI: 10.1021/acsestengg.1023c00207 (2023)). The method is based on that of Calle-Vallejo et. al. (*Phys. Chem. Chem. Phys.* **15**, 3196 (2013)) and Liu et. al. (*ACS Catal.* **9**, 7052-7064 (2019)). The free energy of anion A⁻ is calculated according to:

$$\Delta G_{\text{ads}}(\text{A}^-) = E_{\text{A}^*} + [G_{\text{H}^+} + G_{e^-}] - [G_{\text{HA}} - \Delta G_{\text{sol}} - \Delta G_{\text{protonation}}] - E^*$$

Where E^* and E_{A^*} are the DFT computed enthalpies of bare surface and A* adsorbed to the surface, respectively. G_{H_2} and G_{HA} are the Gibbs free energies of desorbed species H₂ and HA, respectively, in the gas phase at 300 K, as calculated from the following:

$$G_{\text{HA}} = E_{\text{HA}} + E_{\text{ZPE}} - T^*S$$

where E_{HA} is the DFT computed energy of HA in the gas phase, T is the temperature (300 K), E_{ZPE} is the contribution of the zero-point energy, S is the entropic contributions to the free energy obtained using the JANAF database. The solvation energy is described:

$$\Delta G_{\text{sol}} = G_{\text{HA(g)}} - G_{\text{HA(l)}}$$

We agree with the reviewer's opinion that the pH accounts for the effects on the free energies of the species. Free energy modifications due to pH were calculated according to:

$$\Delta G_{\text{protonation}} = G^{\circ} - 2.303kT(\text{p}K_{\text{a}} - \text{pH}) = G_{\text{A}^-} + G_{\text{H}^+} - G_{\text{HA(l)}} - 2.303kT(\text{p}K_{\text{a}} - \text{pH})$$

$G_{\text{HA(g)}}$ and $G_{\text{HA(l)}}$ are the free energies of HA molecule in the gas and liquid phases respectively. k is the Boltzmann constant. K_{a} is the acid dissociation constant for the A^- anion. The standard state (25° C, 100 kPa, 1 mol/kg) energies of ion and neutral species in aqueous solution ($G_{\text{HA(g)}}$, $G_{\text{HA(l)}}$, G_{A^-} , G_{H^+} , K_{a}) are taken from the CRC handbook.

The computational hydrogen electrode (CHE) is used to account for potential effects on reaction energies (*J. Am. Chem. Soc.* **140**, 12256-12262 (2018)):

$$\Delta G = \Delta E + \Delta E_{\text{ZPE}} - T\Delta S + 0.0591*\text{pH} - eU_{\text{RHE}}$$

where ΔE is the DFT computed reaction (electronic) energy, ΔE_{ZPE} and ΔS are the zero-point energy difference and the entropy difference between the adsorbed state and the gas phase, respectively. $0.0591*\text{pH}$ represents the free-energy contribution due to the variations in H concentration. We considered the effect of a potential bias on all states involving one electron or hole in the electrode by shifting the energy of this energy by eU_{RHE} , where U_{RHE} is the electrode potential relative to the reversible hydrogen electrode (RHE).

(3) The calculation results. According to the above equation, we calculated the free energy changes of the NO_3RR over Cu_2O , $\text{Pd}/\text{Cu}_2\text{O}$ and $\text{Pd-Cl}/\text{Cu}_2\text{O}$ at the potential of -0.6 V vs. RHE for $\text{pH}=14$. Since NO_3RR was highly complex with multiple possible branches, we chose the optimal pathway after comparing alternative pathways. For Cu_2O (Fig. R1), the energy is more significantly downhill in $^*\text{NO}_2\text{H} \rightarrow ^*\text{NO}$, $^*\text{NOH} \rightarrow ^*\text{NHOH}$ and $^*\text{NHOH} \rightarrow ^*\text{NH}_2\text{OH}$ steps than that in $^*\text{NO}_2\text{H} \rightarrow ^*\text{NO}_2\text{H}_2$, $^*\text{NOH} \rightarrow ^*\text{N}$ and $^*\text{NHOH} \rightarrow ^*\text{NH}$ steps, respectively. Therefore, Cu_2O is more likely to undergo

the pathway of $*\text{NO}_3 \rightarrow *\text{NO}_2 \rightarrow *\text{NO}_2\text{H} \rightarrow *\text{NO} \rightarrow *\text{NOH} \rightarrow *\text{NHOH} \rightarrow *\text{NH}_2\text{OH} \rightarrow *\text{NH}_2 \rightarrow *\text{NH}_3$. Similarly, Pd/Cu₂O (Fig. R2) is more inclined to follow the pathway of $*\text{NO}_3 \rightarrow *\text{NO}_2 \rightarrow *\text{NO}_2\text{H} \rightarrow *\text{NO} \rightarrow *\text{NOH} \rightarrow *\text{NHOH} \rightarrow *\text{NH} \rightarrow *\text{NH}_2 \rightarrow *\text{NH}_3$, and Pd-Cl/Cu₂O (Fig. R3) tends to follow the pathway of $*\text{NO}_3 \rightarrow *\text{NO}_2 \rightarrow *\text{NO}_2\text{H} \rightarrow *\text{NO} \rightarrow *\text{NOH} \rightarrow *\text{N} \rightarrow *\text{NH} \rightarrow *\text{NH}_2 \rightarrow *\text{NH}_3$.

Figure R1. Gibbs free energy diagram of various intermediates generated during NO₃RR over Cu₂O at the potential of -0.6 V vs. RHE for pH=14.

Figure R2. Gibbs free energy diagram of various intermediates generated during NO₃RR over Pd/Cu₂O at the potential of -0.6 V vs. RHE for pH=14.

Figure R3. Gibbs free energy diagram of various intermediates generated during NO₃RR over Pd-Cl/Cu₂O at the potential of -0.6 V vs. RHE for pH=14.

In the optimal pathway (Figs. R4-6), Cu₂O, Pd/Cu₂O and Pd-Cl/Cu₂O catalysts all encounter the potential-determining step (PDS) of hydrogenation of *NO₂ into *NO₂H (*NO₂ + H₂O + e⁻ → *NO₂H + OH⁻). Pure Cu₂O involves a ΔG of -0.57 eV (Fig. R4), and Pd/Cu₂O presents a relatively lower ΔG of PDS (-0.65 eV, Fig. R5). In comparison, Pd-Cl/Cu₂O shows the lowest ΔG of PDS (-0.76 eV, Fig. R6), and correspondingly advances the progress of NO₃RR. Thus, Pd-Cl/Cu₂O is anticipated as a promising candidate for alkaline NO₃RR towards NH₃ synthesis.

Figure R4. Gibbs free energy diagram of various intermediates generated during NO₃RR over Cu₂O at the potential of -0.6 V vs. RHE for pH=14.

Figure R5. Gibbs free energy diagram of various intermediates generated during NO₃RR over Pd/Cu₂O at the potential of -0.6 V vs. RHE for pH=14.

Figure R6. Gibbs free energy diagram of various intermediates generated during NO₃RR over Pd-Cl/Cu₂O at the potential of -0.6 V vs. RHE for pH=14.

Corresponding revision:

➤ The references (ACS Catal. 13, 6804-6812 (2023), #16; Nat. Nanotechnol. 17, 759-767 (2022), #18; ACS EST Engg., DOI: 10.1021/acsestengg.1023c00207 (2023), #84.) have been cited (Lines 63, 68, 79, 306 and 602, Revised manuscript; Line 1341, Revised supplementary information).

➤ Figures R1-5 have been added to Supplementary Figures 4-8 (Pages 8-12, Revised supplementary information).

➤ Figure R6 has been updated to Figure 2d (Line 143, Revised manuscript).

➤ The corresponding calculation method “H₂O as the proton source in alkaline conditions. The pH has an influence on the major proton donor in the research system.....We considered the effect of a potential bias on all states involving one electron or hole in the electrode by shifting the energy of this energy by eU_{RHE} , where U_{RHE} is the electrode potential relative to the reversible hydrogen electrode (RHE).” has been added to DFT computational details (Lines 588-632, Revised manuscript).

➤ The corresponding content “Next, after considering the effects of potential and pH on NO₃RR pathway with multiple possible branches (Supplementary Figs. 4-6), the optimal pathway on Cu₂O, Pd/Cu₂O and Pd-Cl/Cu₂O models at the potential of -0.6 V versus RHE for pH=14 was proposed and the corresponding ΔG of each intermediate was calculated (Fig. 2d and Supplementary Figs. 7-8). In such a sequential electron-proton transfer process (Supplementary Fig. 9)^{41,42}, the hydrogenation of *NO₂ into *NO₂H (*NO₂ + H₂O + e⁻ → *NO₂H + OH⁻) was the potential-determining step (PDS), which involved a ΔG of -0.57 eV over pure Cu₂O. Pd/Cu₂O also presented a relatively lower ΔG of PDS (-0.65 eV). Pd-Cl/Cu₂O showed the lowest ΔG of PDS (-0.76 eV) and correspondingly advanced the progress of NO₃RR. Thus, Pd-Cl/Cu₂O was anticipated as a promising candidate for alkaline NO₃RR towards NH₃ synthesis.” has been revised (Lines 132-142, Revised manuscript).

➤ The corresponding notes “We calculated the free energy changes of the NO₃RR over Cu₂O, Pd/Cu₂O and Pd-Cl/Cu₂O at the potential of -0.6 V vs. RHE for pH=14. Since NO₃RR is highly complex with multiple possible branches, we chose the optimal pathway after comparing alternative pathways. For Cu₂O (Supplementary Fig. 4), the

energy is more significantly downhill in $*NO_2H \rightarrow *NO$, $*NOH \rightarrow *NHOH$ and $*NHOH \rightarrow *NH_2OH$ steps than that in $*NO_2H \rightarrow *NO_2H_2$, $*NOH \rightarrow *N$ and $*NHOH \rightarrow *NH$ steps, respectively. Therefore, Cu_2O is more likely to undergo the pathway of $*NO_3 \rightarrow *NO_2 \rightarrow *NO_2H \rightarrow *NO \rightarrow *NOH \rightarrow *NHOH \rightarrow *NH_2OH \rightarrow *NH_2 \rightarrow *NH_3$. Similarly, Pd/Cu_2O (Supplementary Fig. 5) is more inclined to follow the pathway of $*NO_3 \rightarrow *NO_2 \rightarrow *NO_2H \rightarrow *NO \rightarrow *NOH \rightarrow *NHOH \rightarrow *NH \rightarrow *NH_2 \rightarrow *NH_3$, and $Pd-Cl/Cu_2O$ (Supplementary Fig. 6) tends to follow the pathway of $*NO_3 \rightarrow *NO_2 \rightarrow *NO_2H \rightarrow *NO \rightarrow *NOH \rightarrow *N \rightarrow *NH \rightarrow *NH_2 \rightarrow *NH_3$.” have been added (Lines 192-201, Revised supplementary information).

Comment 2: *The authors state that they examined the potentials of -0.12 and -0.95V for pH 0 and 14 respectively. However, these values do not match the experimental conditions. These conditions are used for determining the rate limiting steps. However, it is not necessarily these step energetics which are rate limiting. The rate limiting behavior cannot be extracted without calculation of the reaction barriers.*

Response: Following the kind suggestion, we corrected the pH, potential, and hydrogen sources in the nitrate reduction pathway calculation based on experimental conditions (at -0.6 V vs. RHE for pH=14 and H_2O as the proton source).

We agree with the reviewer’s opinion that the reaction rate is relevant to the reaction barriers. We have made attempts to calculate reaction barriers. But the calculation is time-consuming and the computational resources within our group is limited, we apologize for not having made progress on the reaction barriers.

According to previous research (*Energy Environ. Sci.* **3**, 1311-1315 (2010), *Nat. Energy* **5**, 605-613 (2020), *Adv. Mater.* **35**, 2202952 (2023), *J. Am. Chem. Soc.* **145**, 6471–6479 (2023), *Nat. Commun.* **12**, 2870 (2021).), free energy changes of reaction can act as a feasible criterion to evaluate the activity of catalysts.

Based on this, we calculated the free energy changes of the NO_3RR over Cu_2O , Pd/Cu_2O and $Pd-Cl/Cu_2O$ at the potential of -0.6 V vs. RHE for pH=14 (Figs. R4-6). The three catalysts all encounter the potential-determining step (PDS) of hydrogenation of $*NO_2$ into $*NO_2H$ ($*NO_2 + H_2O + e^- \rightarrow *NO_2H + OH^-$). Pure Cu_2O involves a ΔG

of -0.57 eV (Fig. R4), and Pd/Cu₂O presents a relatively lower ΔG of PDS (-0.65 eV, Fig. R5). In comparison, Pd-Cl/Cu₂O shows the lowest ΔG of PDS (-0.76 eV, Fig. R6) and correspondingly advances the progress of NO₃RR. Thus, Pd-Cl/Cu₂O is anticipated as a promising candidate for alkaline NO₃RR towards NH₃ synthesis.

Figure R4. Gibbs free energy diagram of various intermediates generated during NO₃RR over Cu₂O at the potential of -0.6 V vs. RHE for pH=14.

Figure R5. Gibbs free energy diagram of various intermediates generated during NO₃RR over Pd/Cu₂O at the potential of -0.6 V vs. RHE for pH=14.

Figure R6. Gibbs free energy diagram of various intermediates generated during NO₃RR over Pd-Cl/Cu₂O at the potential of -0.6 V vs. RHE for pH=14.

Corresponding revision:

- Figures R4-5 have been added to **Supplementary Figures 7-8** (Pages 11-12, Revised supplementary information).
- Figure R6 has been updated to **Figure 2d** (Line 143, Revised manuscript).
- The corresponding content “**Next, after considering the effects of potential and pH on NO₃RR pathway with multiple possible branches (Supplementary Figs. 4-6), the optimal pathway on Cu₂O, Pd/Cu₂O and Pd-Cl/Cu₂O models at the potential of -0.6 V *versus* RHE for pH=14 was proposed and the corresponding ΔG of each intermediate was calculated (Fig. 2d and Supplementary Figs. 7-8). In such a sequential electron–proton transfer process (Supplementary Fig. 9)^{41,42}, the hydrogenation of *NO₂ into *NO₂H (*NO₂ + H₂O + e⁻ → *NO₂H + OH⁻) was the potential-determining step (PDS), which involved a ΔG of -0.57 eV over pure Cu₂O. Pd/Cu₂O also presented a relatively lower ΔG of PDS (-0.65 eV). Pd-Cl/Cu₂O showed the lowest ΔG of PDS (-0.76 eV) and correspondingly advanced the progress of NO₃RR. Thus, Pd-Cl/Cu₂O was anticipated as a promising candidate for alkaline NO₃RR towards NH₃ synthesis.” has been revised (Lines 132-142, Revised manuscript).**

Comment 3: The authors have indeed included equations as to how entropic effects are included generally; however, they state that they actually neglect these entropy effects on the solid systems in their rebuttal. This statement is not present in the paper. Thus the paper is very misleading. These assumptions must be included in the main text and specifically clarify that entropic effects of the solids were not accounted for.

Response: We thank the reviewer for the meticulous examination. Based on the reviewer’s suggestion, we have supplemented the corresponding descriptions to make them more rigorous and avoid misleading to the readers.

Corresponding revision:

- The corresponding content “**From the above formula, considering that the vibration frequency of the catalyst substrate is small, its corresponding correction is very small and does not affect the calculation results. Thus, we mainly made corrections to gas molecules and adsorbents on the catalyst, and entropic effects of the catalyst substrate**

would not be considered further.” has been added to calculation of thermodynamic corrections. (Lines 583-587, Revised manuscript).

Comment 4: *The authors now justify their selection of two Cl ions by the fact that it gives beneficial results. But the selection of a model based on the desired result is not inappropriate. Rather the mode should be evaluated as to what system is energetically favorable, then the behavior can be extracted.*

Response: We thank the reviewer for raising the concern about the model choice, and we apologize for the misunderstanding that we did not highlight the energy stability of models. The analysis of choosing models has been supplemented in the revised paper.

Choosing a model should indeed be based on its energetically favorable system. Thus, we calculated the adsorbed energy of gradually increased each Cl^- in the $\text{Pd-Cl}_{(n-1)}/\text{Cu}_2\text{O}$ ($n=1, 2, 3$) structure, to select the most stable $\text{Pd-Cl}/\text{Cu}_2\text{O}$ model. The adsorbed energy of Cl^- can be evaluated according to the formula of $\Delta E_{\text{ads}} = E_{\text{Pd-Cl}_n/\text{Cu}_2\text{O}} - E_{\text{Pd-Cl}_{(n-1)}/\text{Cu}_2\text{O}} - E_{\text{Cl}^-}$ ($n=1, 2, 3$). As shown in Fig. R7, the energy of the first, second, and third Cl^- adsorbed on the $\text{Pd-Cl}_{(n-1)}/\text{Cu}_2\text{O}$ configuration are -1.11, -0.87, and 0.11 eV, respectively (Supplementary Fig. 2). This result demonstrates an exothermic process for the adsorption of second Cl^- on $\text{Pd-Cl}_1/\text{Cu}_2\text{O}$ configuration, but an endothermic process for the adsorption of third Cl^- on $\text{Pd-Cl}_2/\text{Cu}_2\text{O}$ configuration. Therefore, the two Cl^- adsorbed on $\text{Pd}/\text{Cu}_2\text{O}$ ($\text{Pd-Cl}_2/\text{Cu}_2\text{O}$) is the most stable, and we chose this model to study its NO_3RR process (Fig. R1).

Figure R7. The adsorbed energy of the first, second, and third Cl^- on the $\text{Pd-Cl}_{(n-1)}/\text{Cu}_2\text{O}$ configuration ($n=1, 2, 3$).

Supplementary Fig. 2. The models of Pd-Cl/Cu₂O with different Cl numbers.

Besides, the quantitative least-squares best-fitting of EXAFS spectra (Fig. 3f and Supplementary Table 1) confirm that Pd center is coordinated with two Cl atoms in the as-prepared Pd-Cl/Cu₂O catalyst. After the electrolysis, the catalysts maintain the original structure (Supplementary Fig. 52c and Supplementary Table 3), suggesting the robust structure of Pd-Cl/Cu₂O with two Cl coordination. The NO₃RR performance tests (Fig. 5a) indicate the as-prepared Pd-Cl/Cu₂O catalyst delivers a superior NH₃ yield rate and FE. These results reveal that the stable configuration of Pd coordinated with two Cl possesses an excellent NH₃ activity.

Figure 3. f, The fitting EXAFS spectra of Pd-Cl/Cu₂O. Inset: fitting model.

Supplementary Table 1. The Pd K-edge EXAFS fitting results for Pd-Cl/Cu₂O.

Sample	Shell	CN	R(Å)	$\sigma^2(\text{Å}^2)$	$\Delta E_0(\text{eV})$	R factor
Pd-Cl/Cu ₂ O	Pd-Cl	1.964	2.34	0.007	22.7	0.0008
	Pd-Cu	3.004	2.52	0.002	7.80	

Supplementary Fig. 52. c, Fitting EXAFS spectra of Pd-Cl/Cu₂O after NO₃RR.

Supplementary Table 3. The Pd K-edge EXAFS fitting results for Pd-Cl/Cu₂O after NO₃RR.

Sample	Shell	CN	R(Å)	$\sigma^2(\text{Å}^2)$	$\Delta E_0(\text{eV})$	R factor
Pd-Cl/Cu ₂ O	Pd-Cl	1.961	2.33	0.0005	20.75	0.0004
	Pd-Cu	3.005	2.58	0.0006	1.68	

Figure 5. a, NH₃ yield rate and NH₃ FE of catalysts in a 1 M KOH with 56 mM NO₃⁻ electrolyte for 1 h electrolysis.

Corresponding revision:

- Figure R7 has been added to **Supplementary Figure 3** (Page 7, Revised supplementary information).
- The corresponding notes “**Choosing a model should indeed be based on its energetically favorable system. Thus, we calculated the adsorbed energy of gradually**

increased each Cl^- in the $\text{Pd-Cl}_{(n-1)}/\text{Cu}_2\text{O}$ ($n=1, 2, 3$) structure, to select the most stable $\text{Pd-Cl}/\text{Cu}_2\text{O}$ model. The adsorbed energy of Cl^- can be evaluated according to the formula of $\Delta E_{\text{ads}} = E_{\text{Pd-Cl}_n/\text{Cu}_2\text{O}} - E_{\text{Pd-Cl}_{(n-1)}/\text{Cu}_2\text{O}} - E_{\text{Cl}^-}$ ($n = 1, 2, 3$). As shown in Supplementary Fig. 3, the energy of the first, second, and third Cl^- adsorbed on the $\text{Pd-Cl}_{(n-1)}/\text{Cu}_2\text{O}$ configuration are -1.11, -0.87, and 0.11 eV, respectively (Supplementary Fig. 2). Therefore, the two Cl^- adsorbed on $\text{Pd}/\text{Cu}_2\text{O}$ ($\text{Pd-Cl}_2/\text{Cu}_2\text{O}$) is the most stable, and we chose this model to study its reaction behavior.” have been added (Lines 134-141, Revised supplementary information).

➤ The corresponding content “The introduction of Cl coordination endowed $\text{Pd-Cl}/\text{Cu}_2\text{O}$ model (optimized model of Pd atoms coordinated with two Cl, Supplementary Fig. 3) with a further decrease of ΔG_{*H_2O} (0.68 eV) and more favorable to generate H^+ (ΔG_{H^+} of -1.34 eV).” has been revised (Lines 118-121, Revised manuscript).

Comment 5: The authors state that they take gas phase H_2O and NO_3^- as the references for their DFT analysis. This is completely unrealistic as the energy of an ion in the gas phase is significantly destabilized compared to one in an aqueous solution. Further water is also stabilized by energetically by condensing. Taken together with their assumptions about entropy of the reactant molecules, this likely leads to massive over prediction of binding. The inclusion of these effects is well reported in the nitrate reduction literature.

Response: Based on the kind suggestion, we have considered aqueous phase H_2O and NO_3^- as the references for DFT analysis to reflect a more realistic reaction situation.

We carefully studied the reference about the commonly recognized calculation methods in nitrate reduction (*Nat. Nanotechnol.* **17**, 759-767 (2022), *ACS Catal.* **13**, 6804-6812 (2023), *ACS EST Engg.*, DOI: 10.1021/acsestengg.1023c00207 (2023).), and recalculated our computational work, including two aspects:

(1) The calculation method.

The free energies of adsorption of ionic species are calculated using thermodynamic Hess cycles (Fig. R8), which include the effects of entropy, solvation energy, protonation energy (including pH effects) and potential effects. The Gibbs

energy formulas for nitrate reduction reaction steps are reported by Muhich et. al. (*ACS EST Engg.*, DOI: 10.1021/acsestengg.1023c00207 (2023)). The method is based on that of Calle-Vallejo et. al. (*Phys. Chem. Chem. Phys.* **15**, 3196 (2013)) and Liu et. al. (*ACS Catal.* **9**, 7052-7064 (2019)). The free energy of anion A^- is calculated according to:

$$\Delta G_{\text{ads}}(A^-) = E_{A^*} + [G_{H^+} + G_{e^-}] - [G_{HA} - \Delta G_{\text{sol}} - \Delta G_{\text{protonation}}] - E^*$$

Where E^* and E_{A^*} are the DFT computed enthalpies of bare surface and A^* adsorbed to the surface, respectively. G_{H_2} and G_{HA} are the Gibbs free energies of desorbed species H_2 and HA , respectively, in the gas phase at 300 K, as calculated from the following:

$$G_{HA} = E_{HA} + E_{\text{ZPE}} - T^*S$$

where E_{HA} is the DFT computed energy of HA in the gas phase, T is the temperature (300 K), E_{ZPE} is the contribution of the zero-point energy, S is the entropic contributions to the free energy obtained using the JANAF database. The solvation energy is described:

$$\Delta G_{\text{sol}} = G_{HA(\text{g})} - G_{HA(\text{l})}$$

The pH accounts for the effects on the free energies of the species. Free energy modifications due to pH were calculated according to:

$$\Delta G_{\text{protonation}} = G^\circ - 2.303kT(\text{p}K_a - \text{pH}) = G_{A^-} + G_{H^+} - G_{HA(\text{l})} - 2.303kT(\text{p}K_a - \text{pH})$$

$G_{HA(\text{g})}$ and $G_{HA(\text{l})}$ are the free energies of HA molecule in the gas and liquid phases respectively. k is the Boltzmann constant. K_a is the acid dissociation constant for the A^- anion. $\text{p}K_a$ of HNO_3 is taken from Lange's handbook of chemistry. The standard state (25° C, 100 kPa, 1 mol/kg) energies of ion and neutral species in aqueous solution ($G_{HA(\text{g})}$, $G_{HA(\text{l})}$, G_{A^-} , G_{H^+} , K_a of H_2O) are taken from the CRC handbook, as shown in Tables R1-2.

Figure R8. Thermodynamic cycle employed to account for free energy of protonation and solvation of NO_3 , which enables calculation of overall free energy of NO_3^- adsorption.

Table R1. Calculation parameters for HNO_3 and H_2O

Name	$G_{(l)}/\text{kJ mol}^{-1}$	$G_{(g)}/\text{kJ mol}^{-1}$	$\text{p}K_a$
HNO_3	-80.7	-73.5	-1.37
H_2O	-237.1	-228.6	13.995

Table R2. Calculation parameters for NO_3^- and H^+

Name	$G/\text{kJ mol}^{-1}$
NO_3^-	-111.3
H^+	0

(2) The calculation results.

According to the above equation, we took aqueous phase H_2O and NO_3^- as the references to investigate reaction behavior over Cu_2O , $\text{Pd}/\text{Cu}_2\text{O}$ and $\text{Pd-Cl}/\text{Cu}_2\text{O}$ (at -0.6 V vs. RHE, $\text{pH}=14$). As shown in Fig. R9, Cu_2O shows a high Gibbs free energy change for H_2O dissociation ($\Delta G_{*\text{H}_2\text{O}}$ of 1.19 eV), while the corresponding value on $\text{Pd}/\text{Cu}_2\text{O}$ (0.70 eV) decreases, indicating the accelerated H_2O dissociation under the assistance of Pd single-atom catalysts. The introduction of Cl coordination endows $\text{Pd-Cl}/\text{Cu}_2\text{O}$ model with a further decrease of $\Delta G_{*\text{H}_2\text{O}}$ (0.68 eV) and more favorable to

generate H^+ (ΔG_{H^+} of -1.34 eV).

Figure R9. Gibbs free energy change of H^+ supply over catalysts.

We also studied the NO_3RR pathway on Cu_2O , $\text{Pd}/\text{Cu}_2\text{O}$ and $\text{Pd-Cl}/\text{Cu}_2\text{O}$ models at the potential of -0.6 V *versus* RHE for $\text{pH}=14$, and the corresponding ΔG of each intermediate was calculated (Figs. R4-6). The three catalysts all encounter the potential-determining step (PDS) of hydrogenation of $^*\text{NO}_2$ into $^*\text{NO}_2\text{H}$ ($^*\text{NO}_2 + \text{H}_2\text{O} + e^- \rightarrow ^*\text{NO}_2\text{H} + \text{OH}^-$). Pure Cu_2O involves a ΔG of -0.57 eV (Fig. R4), and $\text{Pd}/\text{Cu}_2\text{O}$ presents a relatively lower ΔG of PDS (-0.65 eV, Fig. R5). In comparison, $\text{Pd-Cl}/\text{Cu}_2\text{O}$ shows the lowest ΔG of PDS (-0.76 eV, Fig. R6) and correspondingly advances the progress of NO_3RR . Thus, $\text{Pd-Cl}/\text{Cu}_2\text{O}$ is anticipated as a promising candidate for alkaline NO_3RR towards NH_3 synthesis.

Figure R4. Gibbs free energy diagram of various intermediates generated during NO_3RR over Cu_2O at the potential of -0.6 V *vs.* RHE for $\text{pH}=14$.

Figure R5. Gibbs free energy diagram of various intermediates generated during NO₃RR over Pd/Cu₂O at the potential of -0.6 V vs. RHE for pH=14.

Figure R6. Gibbs free energy diagram of various intermediates generated during NO₃RR over Pd-Cl/Cu₂O at the potential of -0.6 V vs. RHE for pH=14.

Corresponding revision:

- Figure R9 has been updated to **Figure 2a** (Line 143, Revised manuscript).
- Figures R4-5 have been added to **Supplementary Figures 7-8** (Pages 11-12, Revised supplementary information).
- Figure R6 has been updated to **Figure 2d** (Line 143, Revised manuscript).
- The corresponding content “**Aqueous phase H₂O and NO₃⁻ were as the energetics references.**” has been revised (Lines 548-549, Revised manuscript).
- The corresponding method “**The free energies of adsorption of ionic species are calculated using thermodynamic Hess cycles, which include the effects of entropy, solvation energy, protonation energy (including pH effects) and potential effects.....The standard state (25° C, 100 kPa, 1 mol/kg) energies of ion and neutral species in aqueous solution ($G_{HA(g)}$, $G_{HA(l)}$, G_{A^-} , G_{H^+} , K_a) are taken from the CRC handbook.**” has been added to DFT computational details (Lines 598-622, Revised

manuscript).

➤ The corresponding content “Cu₂O showed a high Gibbs free energy change for H₂O dissociation (ΔG_{*H_2O} of 1.19 eV, Fig. 2a), while the corresponding value on Pd/Cu₂O (0.70 eV) decreased, indicating the accelerated H₂O dissociation under the assistance of Pd SACs. The introduction of Cl coordination endowed Pd-Cl/Cu₂O model (optimized model of Pd atoms coordinated with two Cl, Supplementary Fig. 3) with a further decrease of ΔG_{*H_2O} (0.68 eV) and more favorable to generate H⁺ (ΔG_{H^+} of -1.34 eV).” has been revised (Lines 115-121, Revised manuscript).

➤ The corresponding content “Next, after considering the effects of potential and pH on NO₃RR pathway with multiple possible branches (Supplementary Figs. 4-6), the optimal pathway on Cu₂O, Pd/Cu₂O and Pd-Cl/Cu₂O models at the potential of -0.6 V *versus* RHE for pH=14 was proposed and the corresponding ΔG of each intermediate was calculated (Fig. 2d and Supplementary Figs. 7-8). In such a sequential electron-proton transfer process (Supplementary Fig. 9)^{41,42}, the hydrogenation of *NO₂ into *NO₂H (*NO₂ + H₂O + e⁻ → *NO₂H + OH⁻) was the potential-determining step (PDS), which involved a ΔG of -0.57 eV over pure Cu₂O. Pd/Cu₂O also presented a relatively lower ΔG of PDS (-0.65 eV). Pd-Cl/Cu₂O showed the lowest ΔG of PDS (-0.76 eV) and correspondingly advanced the progress of NO₃RR. Thus, Pd-Cl/Cu₂O was anticipated as a promising candidate for alkaline NO₃RR towards NH₃ synthesis.” has been revised (Lines 132-142, Revised manuscript).

***Comment 6:** The fact that the addition of Cl after synthesis did not change improve catalytic behavior suggests that the Cl possibly acts more as a synthetic directing agent rather than as a co-catalyst. This concept should be explored further and clarified.*

Response: We agree with the reviewer’s opinion that the Cl acts as a synthetic directing agent. For the structure of Pd-Cl/Cu₂O, Cl plays the vital role in stabilizing Pd single atoms and forming the Pd-Cl coordination. Correspondingly, we implemented a new set of experiments and offered more explanations to clarify it.

(1) In the synthesis process of Pd-Cl/Cu₂O, chloride was utilized as the precursor to bind and stabilize the Pd atoms by impregnation and hydrogen reduction approach.

During the low-temperature treatment procedure, the Cl ligand-mediated effect triggers the formation of Cl coordinated with Pd atom on Cu₂O matrix. According to the quantitative least-squares best-fitting of EXAFS spectra (Fig. 3f and Supplementary Table 1) and XPS spectra (Supplementary Figs. 21c-d and 22), there exists a strong interaction between two Cl and one Pd single atom in the as-prepared Pd-Cl/Cu₂O catalyst. After electrolysis, the unchanged EXAFS results further confirm the stable structure of Cl coordinated Pd (Supplementary Fig. 52c and Supplementary Table 3). Relevant literatures (*Nat. Catal.*, **3**, 376-385 (2020), *Nat. Commun.*, **13**, 6875 (2022); *Electrochem. Energ. Rev.*, **2**, 539-573 (2019).) have reported that Cl ligand can stable single atoms (Au, Pt, and Ru etc.).

(2) The control experiments suggest that simply dipping Cl⁻ on the Pd/Cu₂O or adding Cl⁻ in the electrolyte lead to weak or negligible interaction between Cl and Pd (Fig. R10). The Cl⁻ can easily be eliminated by washing the catalysts with deionized water, due to the weak bond of Pd-Cl. Thus, the two counterparts show comparable NO₃RR activity to pure Pd/Cu₂O, and far lower than that of Pd-Cl/Cu₂O (Supplementary Fig. 43).

(3) In summary, the Cl acts as a synthetic directing agent to stabilize Pd single atoms, which could form strong Pd-Cl bond. Meanwhile, two Cl coordinated with one Pd single atom is the most stable structure.

Figure 3. f, The fitting EXAFS spectra of Pd-Cl/Cu₂O. Inset: fitting model.

Supplementary Table 1. The Pd K-edge EXAFS fitting results for Pd-Cl/Cu₂O.

Sample	Shell	CN	R(Å)	$\sigma^2(\text{Å}^2)$	$\Delta E_0(\text{eV})$	R factor
Pd-Cl/Cu ₂ O	Pd-Cl	1.964	2.34	0.007	22.7	0.0008
	Pd-Cu	3.004	2.52	0.002	7.80	

Supplementary Figure 21. c, Pd 3d and d, Cl 2p XPS spectra of samples.

Supplementary Figure 22. Cl 2p XPS spectra of catalysts.

Supplementary Fig. 52. c, Fitting EXAFS spectra of Pd-Cl/Cu₂O after NO₃RR.

Supplementary Table 3. The Pd K-edge EXAFS fitting results for Pd-Cl/Cu₂O after NO₃RR.

Sample	Shell	CN	R(Å)	$\sigma^2(\text{Å}^2)$	$\Delta E_0(\text{eV})$	R factor
Pd-Cl/Cu ₂ O	Pd-Cl	1.961	2.33	0.0005	20.75	0.0004
	Pd-Cu	3.005	2.58	0.0006	1.68	

Figure R10. a,c Pd 3d and b,d Cl 2p XPS spectra of samples.

Supplementary Figure 43. a,c NH₃ yield rate and b,d NH₃ FE of various catalysts in a 1 M KOH with 56 mM NO₃⁻ electrolyte for 1 h electrolysis.

Corresponding revision:

➤ Figure R10 has been added to **Supplementary Figure 44** (Page 49, Revised supplementary information).

➤ The corresponding notes “**In the synthesis process of Pd-Cl/Cu₂O, chloride was utilized as the precursor to bind and stabilize the Pd atoms by impregnation and hydrogen reduction approach.....In summary, the Cl acts as a synthetic directing agent to stabilize Pd single atoms, which could form strong Pd-Cl bond. Meanwhile, two Cl coordinated with one Pd single atom is the most stable structure.**” have been added (Lines 874-891, Revised supplementary information).

➤ The corresponding content “**The Cl acted as a synthetic directing agent to stabilize Pd single atoms.**” has been added (Lines 409-410, Revised manuscript).

REVIEWERS' COMMENTS

Reviewer #1 (Remarks to the Author):

The authors have resolved my concerns.

Reviewer#3 (Remarks to the Author):

The authors have made major improvements to their manuscript, particularly the computational section. With the corrections and the justifications, this work is now publishable.